# The Expressive Power of Low Precision Softmax Transformers with (Summarized) Chain-of-Thought

**Moritz Brösamle** [1]   **Stephan Eckstein** [1]

## Abstract

Existing expressivity results for transformers typically rely on hardmax attention, high precision, and other architectural modifications that disconnect them from the models used in practice. We bridge this gap by analyzing standard transformer decoders with softmax attention and rounding of activations and attention weights, while allowing depth and width to grow logarithmically with the context length. As an intermediate step, we construct hardmax transformers with ternary activations and well-separated attention scores that simulate Turing machines using Chain-of-Thought (CoT). This lets us convert the constructions to equivalent softmax transformers without the unrealistic parameter magnitudes or activation precision that prior approaches would require. Using the same technique, we analyze a recently proposed summarized CoT paradigm and show that it simulates Turing machines more efficiently, with model size scaling logarithmically in a space bound rather than a time bound. We empirically test predictions made by our results on a Sudoku reasoning task and find better alignment with learnability than for prior high-precision results. Our code is available at https://github.com/moritzbroe/transformer-expressivity.

## 1. Introduction

Theoretical works on the capabilities of transformers typically use models that deviate substantially from those used in practice. In particular, to achieve fixed width and depth, they often require activation precision growing at least logarithmically with the context length, use hardmax attention,

and sometimes add further architectural modifications not seen in practice (Strobl et al., 2024; Pérez et al., 2021; Merrill & Sabharwal, 2024; Yang et al., 2025). By contrast, practical transformers use softmax attention and round activations to 16 or even 8-bit precision, often with little impact on performance (Xiao et al., 2023; Dettmers et al., 2022). This suggests a fundamental mismatch between the models analyzed in expressivity results and those used in practice, necessitating the analysis of the theoretical expressivity of practical models.

We prove such expressivity results for transformers with softmax attention and low-precision activations, while allowing depth and width to grow logarithmically with the context length. This leads to models more in line with practical ones, which typically use large depth and width for increased capacity while activation precision stays small (Brown et al., 2020; Touvron et al., 2023). We focus on showing Turing completeness with Chain-of-Thought (CoT) (Wei et al., 2022b; Nye et al., 2021) and more efficiently with summarized Chain-of-Thought (SCoT) (Yan et al., 2026; Aghajohari et al., 2026; Yang et al., 2025), where the model can write multiple token segments with intermediate summaries before giving its answer. This paradigm has been proposed previously in various forms; see Appendix A.4. We use SCoT as a generic name and do not claim novelty.

**Main results.** We show that a softmax transformer with activation and attention weight rounding can

- (warm-up) recognize a regular language up to input length $\hat{n}$ without CoT (Proposition 3.1 + Theorem 4.2),

- simulate a Turing machine $M$ with CoT for all inputs $w$ on which $M$ requires $t_M(w) \leq \hat{t}$ steps and $|w| \leq \hat{t}$ for a given *time bound* $\hat{t}$, using $\mathcal{O}(t_M(w) + |w|)$ decoding steps (Theorem 3.2 + Theorem 4.2),

- simulate a Turing machine $M$ with SCoT for all inputs $w$ on which $M$ requires $s_M(w) \leq \hat{s}$ space for a given *space bound* $\hat{s}$, with $\mathcal{O}(t_M(w) + |w|)$ decoding steps, and maximum segment length (hence context size) $\mathcal{O}(s_M(w))$ (Theorem 3.3 + Theorem 4.2),

with depth and width logarithmic and parameter magnitudes

---

[1]Department of Mathematics, University of Tübingen, Germany. Correspondence to: Moritz Brösamle <moritzbroesamle@gmail.com>.

*Proceedings of the 43rd International Conference on Machine Learning*, Seoul, South Korea. PMLR 306, 2026. Copyright 2026 by the author(s).

| Work | Depth | Width | Prec. | Steps | Context | Parameters | Att. |
|---|---|---|---|---|---|---|---|
| (Merrill & Sabharwal, 2024) | $\mathcal{O}(1)$ | $\mathcal{O}(1)$ | $\mathcal{O}(\log \hat{t})$ | $\mathcal{O}(t)$ | $\mathcal{O}(t)$ | $\mathcal{O}(1)$ | Hard |
| (Yang et al., 2025) (SCoT) | $\mathcal{O}(1)$ | $\mathcal{O}(1)$ | $\mathcal{O}(\log \hat{t})$ | $\mathcal{O}(t)$ | $\mathcal{O}(s)$ | $\mathcal{O}(1)$ | Hard |
| (Li et al., 2024) | $\mathcal{O}(1)$ | $\mathcal{O}(\log \hat{t})$ | $\mathcal{O}(1)$ | $\mathcal{O}(t \log t)$ | $\mathcal{O}(t \log t)$ | $\mathcal{O}(\hat{t}(\log \hat{t})^2)$ | Soft |
| (Jiang et al., 2026) | $\mathcal{O}(1)$ | $\mathcal{O}(1)$ | $\mathcal{O}(\hat{t})$ | $\mathcal{O}(\exp(t))$ | $\mathcal{O}(\exp(t))$ | $\mathcal{O}(1)$ | Soft |
| Ours (CoT) | $\mathcal{O}(\log \hat{t})$ | $\mathcal{O}(\log \hat{t})$ | $\mathcal{O}(\log \log \hat{t})$ | $\mathcal{O}(t)$ | $\mathcal{O}(t)$ | $\mathcal{O}((\log \hat{t})^3)$ | Soft |
| Ours (SCoT) | $\mathcal{O}(\log \hat{s})$ | $\mathcal{O}(\log \hat{s})$ | $\mathcal{O}(\log \log \hat{s})$ | $\mathcal{O}(t)$ | $\mathcal{O}(s)$ | $\mathcal{O}((\log \hat{s})^3)$ | Soft |

*Table 1.* Comparison of asymptotic scalings of transformer decoder constructions for simulating a Turing-machine running on input $w$ using $t = t_M(w) \leq \hat{t}$ steps and $s = s_M(w) \leq \hat{s}$ space.

sub-logarithmic in $\hat{n}/\hat{t}/\hat{s}$. Constant mantissa bits suffice everywhere, while the required exponent bits are double-logarithmic for attention weights and triple-logarithmic for activations. All results yield explicit bounds on depth, width and precision. We use these in Appendix F.1 to argue that rounding activations and attention weights to the commonly used bfloat16 precision suffices in our constructions up to context sizes of around $10^{38}$, while such rounding can break prior log-precision constructions at small context lengths even at 32-bit precision.

**Construction Outline.** Directly constructing softmax transformers is difficult due to their continuous nature, and the difficulty is worsened by activation and attention-weight rounding. Hence, a natural route to obtain such transformers is by converting hardmax constructions to softmax attention, while keeping the error from softmax attention and rounding small enough to ensure the softmax variant generates the same tokens as its hardmax counterpart. In general, any hardmax transformer can in principle be converted to softmax attention by scaling the query and key projections by a sufficiently large constant $c$ to sharpen the softmax attention weights towards their hardmax counterparts (Liu et al., 2023; Yang et al., 2026; Sanford et al., 2024). While one could use this technique to convert existing hardmax transformer constructions (Merrill & Sabharwal, 2024; Yang et al., 2025) to softmax attention, this would, in addition to their need for logarithmic activation precision, require unrealistic weight magnitudes: the gaps between attention scores in these constructions shrink polynomially with sequence length and hence the scaling factor $c$ required to sufficiently sharpen attention weights would need to grow polynomially with context length.

Hence, in Section 3, we construct hardmax transformers with ternary ($\{-1, 0, 1\}$-valued) activations and well-separated attention scores for recognizing regular languages (Proposition 3.1) and simulating Turing machines with CoT and SCoT (Theorems 3.2 and 3.3). In Section 4, we then show how to convert these constructions to softmax attention with rounding of activations (Theorem 4.1) and attention weights (Theorem 4.2).

**Expressivity and Learnability.** Our results only yield expressivity bounds, not learnability guarantees. We therefore empirically investigate in Section 5 whether expressivity results for softmax transformers with rounding give better qualitative predictions about learnability than results for transformer models that deviate further from those actually trained, e.g. by using hardmax attention and high precision. We find evidence for this in a specific setting by training small transformers to imitate a standard depth-first search algorithm on Sudokus. Concretely, as long as SCoT summaries keep the context length bounded, small models reliably solve very hard Sudokus, while CoT models of the same size fail beyond a certain context length. While this pattern is consistent with both our results and prior log-precision results on CoT and SCoT (Merrill & Sabharwal, 2024; Yang et al., 2025), the two types of results suggest different remedies for improving long-context performance of the CoT model. Our results suggest increasing model size, which cleanly improves performance for longer contexts. Prior log-precision CoT results (Merrill & Sabharwal, 2024), on the other hand, suggest increasing activation precision, but moving from 16-bit to 32-bit has negligible effect.

Unfortunately, even under our more realistic assumptions, expressivity does not generally imply learnability. For instance, transformers are hard to train on automata tasks (Liu et al., 2023), so our expressivity result for regular language recognition still overestimates learnability.

**Prior Work.** A significant number of prior works showed that transformers are capable of simulating Turing machines. The asymptotic scalings of the most directly comparable ones are shown in Table 1, while Appendix A presents an extended version of the table including constructions with non-standard positional encodings and encoder-decoder transformers and discusses each work. Many prior works aim to construct a transformer with fixed depth, width and parameters, a property referred to as *uniformity*, usually at the cost of using hardmax attention and logarithmically growing activation precision. Among prior works, (Merrill & Sabharwal, 2024) and (Yang et al., 2025) are the ones that most directly simulate Turing machines by generating tokens encoding Turing-machine steps. As stated previously,

converting them to softmax attention would require scaling parameters by a context-dependent factor, hence breaking uniformity. (Jiang et al., 2026) offers the only result we are aware of about truly uniform softmax transformers being Turing complete, albeit at the cost of exponential slowdown and precision linear in the simulated Turing-machine time. (Li et al., 2024) (and, in an encoder-decoder setup with hardmax attention, (Wei et al., 2022a)) drop uniformity and use logarithmic width to encode positions, similar to our constructions. However, (Li et al., 2024) require problem-specific positional encodings, hence yielding super-linear growth of the number of parameters.

## 2. Preliminaries

### 2.1. Finite Automata and Turing Machines

A *deterministic finite automaton* (DFA) is a tuple $M = (Q, \Sigma, \delta, q_{\text{init}}, F)$ where $Q$ is a finite state set, $\Sigma$ a finite alphabet, $\delta : Q \times \Sigma \to Q$ the transition function, $q_{\text{init}} \in Q$ the initial state and $F \subseteq Q$ the accepting states. We denote by $L(M)$ the language *recognized by* $M$, i.e., the set of words $w \in \Sigma^*$ such that running $M$ on $w$ ends with a state in $F$. See Definition B.3 for details.

A (multi-tape) *Turing machine* is a tuple $M = (K, Q, \Sigma, \Gamma, \delta, \sqcup, q_{\text{init}}, q_{\text{halt}})$ where $K \in \mathbb{N}$ is the number of tapes, $Q$ is a finite state set with initial state $q_{\text{init}}$ and halting state $q_{\text{halt}}$, $\Sigma$ is the input alphabet, $\Gamma \supseteq \Sigma$ is the tape alphabet with blank symbol $\sqcup \in \Gamma \setminus \Sigma$ and $\delta : Q \times \Gamma^K \to Q \times \Gamma^K \times \{L, S, R\}^K$ is the transition function. Tapes are infinite to the right and the first tape is used for both input and output. For input $w \in \Sigma^*$ on which $M$ halts with output $f_M(w) \in \Sigma^*$, we denote by $t_M(w)$ the number of steps and by $s_M(w)$ the largest number of tape cells accessed on any tape during the computation, including usage by the input (i.e., $s_M(w) \geq |w|$). See Definitions B.4 and B.6 for details. Importantly, single-tape Turing machines can have a quadratic slowdown compared to multi-tape machines (Hennie, 1965). Hence, for proving efficiency results, multi-tape machines should always be considered.

### 2.2. Transformers

Throughout this work we will use two types of transformer decoders, one with hardmax attention[1] and one with softmax attention.

We omit normalization like LayerNorm (Ba et al., 2016) or RMSNorm (Zhang & Sennrich, 2019) but note that the

---

[1]By hardmax attention we always refer to *averaging hardmax attention* defined below as opposed to *unique hard attention* variants treated in some prior works (Hao et al., 2022; Jerad et al., 2025) which don't have a direct connection to softmax attention.

constructions could be modified to ignore such normalizations with the same argument as in (Li et al., 2024). Precise definitions for transformers can be found in Appendix B.4.

Let $\mathcal{V}$ be a finite vocabulary, whose elements are called *tokens*. A transformer decoder with depth $L \in \mathbb{N}$, dimension $d \in \mathbb{N}$, key-query and value dimensions $d_k, d_v \in \mathbb{N}$, feed-forward dimension $d_{\text{ff}} \in \mathbb{N}$, $H$ heads and vocabulary $\mathcal{V}$ consists of parameters $\theta$ as specified below and computes a map

$$T_\theta : \mathcal{V}^+ \to \mathcal{V},$$

where $\mathcal{V}^+ := \bigcup_{n \geq 1} \mathcal{V}^n$. For $\tau_0, \ldots, \tau_{n-1} \in \mathcal{V}$, the map is defined as follows:

**Embeddings.** The initial embeddings are the sum of *token embeddings* and *positional embeddings*

$$x_i^{(0)} := \text{emb}_{\tau_i} + \text{posemb}(i)$$

for token embeddings $\text{emb}_\tau \in \mathbb{R}^d$ and a fixed positional embedding function $\text{posemb} : \mathbb{N}_0 \to \mathbb{R}^d$. As activations (including the initial embeddings) in our constructions have entries in $\{-1, 0, 1\}$, we use *binary positional embeddings*, i.e., $\text{posemb}(i)$ consists of the binary representation of $i$ in some $d_{\text{pos}} \leq d$ coordinates and zeros elsewhere. This can loosely be seen as a discrete analogue of the sinusoidal positional embeddings used in the original transformer (Vaswani et al., 2017), as each bit of the binary representation is periodic with exponentially decreasing frequencies for higher bits. However, as many modern transformers use rotary positional encodings (RoPE) (Su et al., 2024; Touvron et al., 2023), we analyze how the results transfer to using RoPE in Appendix F.2. In particular, we show that RoPE can be used at double-logarithmic activation precision to obtain these binary positional embeddings (Lemma F.2).

**Self-Attention.** A single self-attention head is specified by matrices

$$W_Q, W_K \in \mathbb{R}^{d_k \times d}, \ W_V \in \mathbb{R}^{d_v \times d}, \ W_O \in \mathbb{R}^{d \times d_v}.$$

For inputs $(x_0, \ldots, x_{n-1}) \in (\mathbb{R}^d)^n$, define *queries, keys* and *values* as

$$q_i := W_Q x_i, \quad k_j := W_K x_j, \quad v_j := W_V x_j$$

and *attention scores* as

$$s_{ij} := \frac{1}{\sqrt{d_k}} \langle q_i, k_j \rangle$$

for $0 \leq i \leq n - 1$ and $0 \leq j \leq i$. Define *attention weights* for softmax attention as

$$(\alpha_{ij})_{j=0}^i = \text{softmax}\left((s_{ij})_{j=0}^i\right) = \left(\frac{e^{s_{ij}}}{\sum_{l=0}^i e^{s_{il}}}\right)_{j=0}^i$$

and for hardmax attention (Hao et al., 2022) as

$$(\alpha_{ij})_{j=0}^{i} = \text{hardmax}\left((s_{ij})_{j=0}^{i}\right)$$

with

$$\alpha_{ij} = \begin{cases} \frac{1}{|J_i|}, & j \in J_i, \\ 0, & \text{otherwise,} \end{cases} \quad \text{where} \quad J_i := \arg\max_{0 \le j \le i} s_{ij} .$$

The output of the attention head is then defined as

$$z_i := W_O \sum_{j=0}^{i} \alpha_{ij} v_j .$$

**Feedforward and residual connections.** For every $\ell \in \{1, \ldots, L\}$ the transformer applies a transformer layer as

$$x_i^{(\ell-0.5)} := x_i^{(\ell-1)} + \sum_{h=1}^{H} A_{\ell,h}(x_0^{(\ell-1)}, \ldots, x_i^{(\ell-1)})$$
$$x_i^{(\ell)} := x_i^{(\ell-0.5)} + f_\ell(x_i^{(\ell-0.5)})$$

where $A_{\ell,h}$ are attention heads and $f_\ell$ are standard single hidden layer ReLU MLPs with width $d_{\text{ff}}$.

**Output projection.** Finally the output is defined as

$$T_\theta(\tau_0, \ldots, \tau_{n-1}) := \arg\max_{\tau \in \mathcal{V}} \langle \text{unemb}_\tau, x_{n-1}^{(L)} \rangle$$

for unembedding vectors $\text{unemb}_\tau \in \mathbb{R}^d$.

The parameters $\theta$ then consist of embeddings, attention projections, MLP weights and biases and unembeddings, while the positional embeddings are not viewed as parameters.

### 2.3. (Summarized) Chain-of-Thought Generation

When simulating a Turing machine $M$, we want the transformer $T$ to compute $u = f_M(w) \in \Sigma^m$ from $w \in \Sigma^n$ by receiving $w$ as input, generating intermediate tokens (the Chain-of-Thought), and then generating the output $u$. For standard Chain-of-Thought we simply place the input between special delimiter tokens and expect the output to be placed between other delimiter tokens:

**Definition 2.1** (Chain-of-Thought). Let $T$ be a transformer with vocabulary $\mathcal{V}$ such that

$$\{\texttt{<inp>}, \texttt{</inp>}, \texttt{<outp>}, \texttt{</outp>}\} \cup \Sigma \subset \mathcal{V}$$

and let $w \in \Sigma^n, u \in \Sigma^m$. We say that $T$ computes $u$ from $w$ with CoT if running $T$ autoregressively on input

$$\texttt{<inp>}, w_0, \ldots, w_{n-1}, \texttt{</inp>}$$

generates a token sequence ending in

$$\texttt{<outp>}, u_0, \ldots, u_{m-1}, \texttt{</outp>} .$$

See Definition B.16 for details.

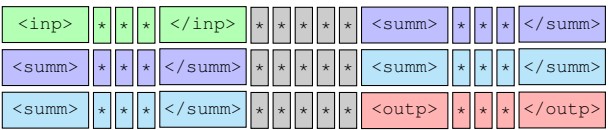

*Figure 1.* SCoT generation schematic: a token segment can be finished by a summary, which is then used to prompt the next segment until an output block is generated.

For summarized Chain-of-Thought, $T$ may generate a summary between other delimiter tokens instead of the output, and can then continue generating from this summary:

**Definition 2.2** (Summarized Chain-of-Thought). Consider the setting of Definition 2.1, where additionally $\texttt{<summ>}, \texttt{</summ>} \in \mathcal{V}$. Let $T$ generate from input

$$\texttt{<inp>}, w_0, \ldots, w_{n-1}, \texttt{</inp>}$$

until writing $\texttt{</summ>}$ or $\texttt{</outp>}$. If the generated sequence ends with $\texttt{<summ>} \ldots \texttt{</summ>}$, $T$ continues generating from just this summary block. This process is repeated until one of the segments ends with $\texttt{</outp>}$. We say that $T$ computes $u$ from $w$ with SCoT if this format is adhered to and the final generated segment ends with

$$\texttt{<outp>}, u_0, \ldots, u_{m-1}, \texttt{</outp>} .$$

See Definition B.17 for details.

The SCoT process is schematically illustrated in Figure 1. CoT simply corresponds to a single input-to-output segment.

### 2.4. Precision and Rounding

We say that a transformer has *ternary weights* if all weight matrices including embeddings and unembeddings have entries in $\{-1, 0, 1\}$. Bias precision is discussed in Remark B.21. We say that the transformer has *ternary activations* on some input if all intermediate representations $(x^{(\ell-0.5)}, x^{(\ell)})$, queries, keys, values, attention head outputs, sum of attention head outputs, MLP hidden activations and MLP outputs have entries in $\{-1, 0, 1\}$. When running a transformer with *activation rounding* to a floating-point format, standard rounding (see Definition B.18) is applied to each of these activations immediately after computation. Similarly, *attention weight rounding* refers to rounding the attention weights $\alpha_{ij}$ after computing them. See Section 4 and Remark D.5 for further discussion on this choice.

## 3. Hardmax Transformer Constructions

We present theorems about the existence of hardmax transformers with ternary weights and activations for recognizing regular languages without CoT and simulating Turing machines with CoT and SCoT, where width and depth grow logarithmically with context size. In the constructive proofs of

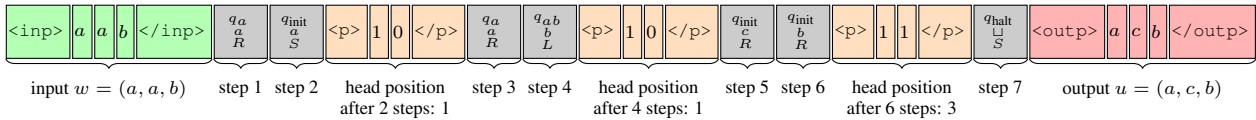

*Figure 2.* Example CoT token sequence for a 1-tape Turing machine that replaces $ab$ with $cb$, run on input $aab$ using $r = 2$ head-position bits. Head positions are written with least significant bit first. Each grey token encodes one Turing machine step consisting of the new state, the written symbol and the head movement (where $L/S/R$ denote left/stay/right).

these statements in Appendix C, the coordinates $\{1, \ldots, d\}$ of the transformer's representations are conceptually divided into disjoint *registers*, holding binary values which encode positions (absolute token positions and Turing machine head positions), states, symbols, etc., and *flags* holding truth values, e.g. for marking different token types. Attention heads are used to extract registers and flags from certain previous tokens, while MLPs are used e.g. for copying, zeroing, incrementing and decrementing binary values stored in registers, often gated by flags. This conceptual abstraction is also used to implement the Turing machine constructions in Python, which allows us to compile Turing machines into PyTorch transformers. The correctness is then checked on random Turing machines as discussed in Appendix C.5.

**Notation.** In the following, $\mathcal{O}_M(\cdot)$ allows the hidden constant to depend on the simulated automaton or Turing machine $M$, while for $\mathcal{O}(\cdot)$ it is uniform in $M$.

### 3.1. Warmup: Regular Language Recognition

Existing results for regular language recognition with transformers (Liu et al., 2023; Merrill & Sabharwal, 2025) already use logarithmic depth and are straightforward to adapt to our needs, i.e., ternary activations and logarithmic width. These constructions use a binary-tree style aggregation to compute the automaton's output and we mainly need to encode positions in binary using logarithmic width instead of using logarithmic precision and constant width. This leads to the following statement:

**Proposition 3.1.** *Let $M$ be a DFA and let $\hat{n} \in \mathbb{N}$. Then there exists a hardmax transformer $T_\theta = T_\theta(M, \hat{n})$ with ternary weights, vocabulary*

$$\mathcal{V} = \Sigma \cup \{\text{<bos>}, \text{True}, \text{False}\},$$

*$H = 1$ head per layer, depth $L = \mathcal{O}(\log \hat{n})$ and dimensions*

$$d, d_{ff} = \mathcal{O}_M(\log \hat{n}), \quad d_k = \mathcal{O}(\log \hat{n}), \quad d_v = \mathcal{O}_M(1),$$

*such that for all $w = (w_0, \ldots, w_{n-1}) \in \Sigma^*$ with $n \leq \hat{n}$,*

$$T_\theta(\text{<bos>}, w_0, \ldots, w_{n-1}) = \begin{cases} \text{True}, & w \in L(M), \\ \text{False}, & w \notin L(M) \end{cases}$$

*and activations are ternary on these inputs.*

The precise statement with explicit constants and its proof are deferred to Appendix C.2.

### 3.2. Simulating Turing Machines with Chain-of-Thought

Next, we show that producing $\mathcal{O}(t)$ intermediate tokens allows transformers in our setting to simulate a Turing machine running for $t$ steps:

**Theorem 3.2** (Turing Machine Simulation with CoT). *Let $M$ be a multi-tape Turing machine with input alphabet $\Sigma$ and let $\hat{t} \in \mathbb{N}$. Then there exists a hardmax transformer $T_\theta = T_\theta(M, \hat{t})$ with ternary weights, vocabulary $\mathcal{V} \supset \Sigma$, $H = \mathcal{O}_M(1)$ heads per layer, depth $L = \mathcal{O}(\log \hat{t})$ and dimensions*

$$d, d_v, d_{ff} = \mathcal{O}_M(\log \hat{t}), \quad d_k = \mathcal{O}(\log \hat{t})$$

*such that the following holds: for all inputs $w \in \Sigma^*$ where $M$ uses at most $\hat{t}$ steps to output $f_M(w) \in \Sigma^*$ and the input has length bounded by $\hat{t}$ as well, the transformer $T$ computes $f_M(w)$ from $w$ with CoT (Definition 2.1) with $\mathcal{O}(t_M(w) + |w|)$ tokens in the generated sequence with ternary activations.*

The precise formulation with explicit constants and the full proof are stated in the appendix as Theorem C.14. The result is directly comparable to (Merrill & Sabharwal, 2024), but replaces the logarithmic precision and constant depth with logarithmic depth/width and ternary activations requiring different construction techniques.

The construction uses a constant $r$ that determines the bit-width of position encodings, hence context lengths and head positions must stay below $2^r$. An example of the token sequence produced by the constructed $T_\theta$ for a single-tape Turing machine is shown in Figure 2, where the <inp>...</inp> and <outp>...</outp> blocks are as required by Definition 2.1. In between, the transformer produces one *run token* for each step of the simulated Turing machine encoding the newly entered state, symbols written at the previous head positions and head movements for all tapes, which is in line with (Merrill & Sabharwal, 2024). However, after each chunk of $r$ such tokens, we additionally insert blocks explicitly writing out the head positions for all tapes in binary to simplify the construction. The main computations the transformer performs to produce this token sequence are:

- The head positions are recovered by collecting the bits

from the `<p>`...`</p>` block into the `</p>` token and then sequentially adding the head movements of the run tokens in sequential layers.

- In order to read the symbol at the current head positions, tokens must search for the closest previous token which wrote a symbol to this position. This is done using a binary search for this latest matching position requiring depth $r$ (i.e., logarithmic in $\hat{t}$).

- After extracting the symbols at the current head positions, the transition function can be applied to compute the next token.

The binary search technique has been used before in (Wei et al., 2022a), while the head position reconstruction technique apparently hasn't been used before—high-precision works like (Merrill & Sabharwal, 2024; Yang et al., 2025) instead use uniform attention over prior tokens to recover the head position in a single layer.

### 3.3. Simulating Turing Machines with Summarized Chain-of-Thought

With summarized CoT, a very similar theorem can be shown. Note that all the scalings now depend on a space bound $\hat{s}$ for the simulated Turing machine instead of a time bound $\hat{t}$, while the segment lengths correspond to the actual space usage of the simulated Turing machine.

**Theorem 3.3** (Turing Machine Simulation with SCoT). *Let $M$ be a Turing machine with input alphabet $\Sigma$ and let $\hat{s} \in \mathbb{N}$. Then there exists a hardmax transformer $T_\theta = T_\theta(M, \hat{s})$ with ternary weights, vocabulary $\mathcal{V} \supset \Sigma$, $H = \mathcal{O}_M(1)$ heads per layer, depth $L = \mathcal{O}(\log \hat{s})$ and dimensions*

$$d, d_v, d_{ff} = \mathcal{O}_M(\log \hat{s}), \quad d_k = \mathcal{O}(\log \hat{s})$$

*such that the following holds: for all inputs $w \in \Sigma^*$ where $M$ uses $t_M(w)$ steps and $s_M(w) \leq \hat{s}$ space to output $f_M(w) \in \Sigma^*$, the transformer $T$ computes $f_M(w)$ from $w$ with SCoT (Definition 2.2) with each produced token segment having length $\mathcal{O}(s_M(w))$ and all segments together having length $\mathcal{O}(t_M(w) + |w|)$ with ternary activations.*

The precise formulation with explicit constants is stated in the appendix as Theorem C.22, where the proof can be found as well. The construction is largely similar to the one without summaries, except for writing summaries at appropriate points. Summaries encode the Turing machine's configuration, i.e., its tape contents, state and head positions, as exemplified in Figure 3. Importantly, the number of tokens in a segment before the final summary is proportional to the length of the initial `<inp>`...`</inp>` or `<summ>`...`</summ>` block in the segment, which ensures that the total length of all segments is $\mathcal{O}(t_M(w))$ and

at the same time each segment length is $\mathcal{O}(s_M(w))$. Writing summaries too frequently would mean losing the former property, while writing them too infrequently means losing the latter property.

## 4. Softmax Transformers and Rounding

Here we present two methods for converting the hardmax transformers from the previous section to softmax transformers with rounding. We use standard floating-point formats with mantissa and exponent bits and rounding to nearest (IEEE, 2019); see Definition B.18 for details. The exact points where rounding is applied can be found in Definition D.4 and correspond closely, to the best of our knowledge, to how rounding is applied in practice as briefly discussed in Remark D.5.

The error made when replacing hardmax by softmax attention can be bounded with the *separation* of attention scores. In particular, for attention scores $s_j$ (corresponding to the scores $s_{ij}$ in an attention head) it holds that (Lemma D.3)

$$\|\operatorname{softmax}((s_j)_{j=0}^{n-1}) - \operatorname{hardmax}((s_j)_{j=0}^{n-1})\|_1 \leq 2ne^{-\beta},$$

where $\beta$ is the *separation* of the scores defined as

$$\beta = s^* - \max\{s_j \mid s_j < s^*\}, \quad s^* = \max_j s_j.$$

Replacing query and key projections $W_Q, W_K$ with $cW_Q, cW_K$ for a constant $c$ scales attention scores and hence $\beta$ by $c^2$. The initial separation in our hardmax constructions is at least $\frac{1}{\sqrt{d_k}}$, as queries and keys are ternary and hence have inner products in $\mathbb{Z}$ which are divided by $\sqrt{d_k}$ to obtain the attention scores. This allows for moderate scalings of $c$ to suffice in order to make softmax and hardmax attention weights and hence head outputs similar. Both settings use this to control the error from replacing hardmax with softmax attention.

### 4.1. Exact Attention

We first consider the simplified setting where activations are rounded, but attention weights are computed in full precision (i.e. no rounding occurs inside attention). All rounded values are ternary in the corresponding hardmax transformers, which allows for a simple way to bound the error from rounding: in particular, if a number is exactly representable in a floating-point format, then perturbing it and rounding it back to the same format can at most double the perturbation (Lemma D.12). Choosing $c$ sufficiently large then allows controlling the error from both softmax conversion and rounding end-to-end as stated in Theorem D.16. Applied to our constructions this results in the following statement:

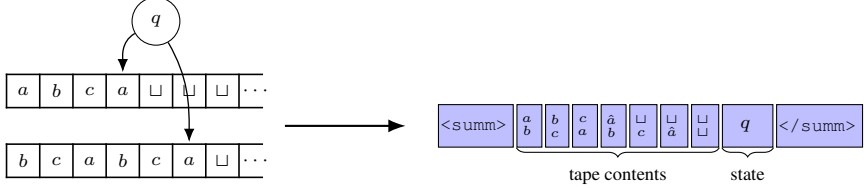

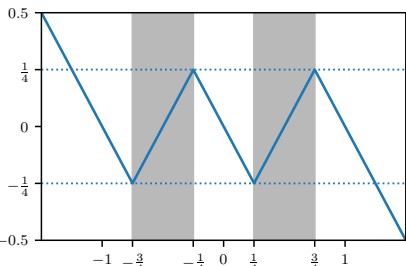

*Figure 3.* Left: Configuration of a 2-tape Turing machine. Right: Encoding of this configuration into a summary for SCoT. Hats over symbols indicate the head position for each tape.

**Theorem 4.1.** *Fix a hardmax transformer $T$ from one of the hardmax constructions from the previous section for an input/time/space bound $\hat{l}$, i.e., $\hat{l} = \hat{n}$ for Proposition 3.1, $\hat{l} = \hat{t}$ for Theorem 3.2 and $\hat{l} = \hat{s}$ for Theorem 3.3. Then there exists*

$$c_0(\hat{l}) = \mathcal{O}\big((\log \hat{l})^{\frac{3}{4}} (\log \log \hat{l})^{\frac{1}{2}}\big)$$

*such that for all $c \geq c_0(\hat{l})$, the softmax transformer $\tilde{T}_c$ obtained from $T$ by scaling all query and key projections by $c$ outputs the same tokens as $T$ for all valid inputs. Further, this still holds when rounding all activations (as in Section 2.4) in $\tilde{T}_c$ to a floating-point format containing $c$.*

The activation precision only needs to be able to represent a sufficiently large $c$, which requires $\mathcal{O}(\log \log \log \hat{l})$ exponent bits. The proof is given in Appendix D.4.1.

### 4.2. Rounding Attention Weights

In real transformers, attention weights $\alpha_{ij}$ are rounded to low precision as well before being multiplied by the values $v_j$. In our hardmax transformer constructions, attention can generally select multiple tokens, leading to attention weights of the form $\frac{1}{k}$. Such weights need not be representable in any floating-point format, hence rounding them introduces an error that cannot be controlled by increasing $c$ and applying Lemma D.12. One way to control the error after all without having to use unrealistically large attention weight precision is to use an additional MLP after each attention layer, which implements coordinatewise the function shown in Figure 4. When added to the residual stream, this MLP maps activation entries in $(-\frac{5}{4}, -\frac{3}{4}), (-\frac{1}{4}, \frac{1}{4}), (\frac{3}{4}, \frac{5}{4})$ to the ternary activations $-1, 0, 1$ respectively. Hence, if the error introduced in the attention layer is coordinatewise bounded by $\frac{1}{4}$, the MLP removes this error by mapping activations back to the correct ternary values. This technique has been used before, e.g. in (Wei et al., 2022a; Yang et al., 2026). The existence of this MLP and its correctness when rounding hidden activations and outputs to a floating-point format with at least 3 exponent bits and 1 mantissa bit is shown in Lemma D.18.

To add these MLPs in a standard architecture requires each layer of the original hardmax transformer to be replaced by 2 layers in its rounded softmax variant: the first one performs

*Figure 4.* Coordinatewise denoising function $f$ implemented by the additional MLPs. For inputs inside the three white blocks, adding this function onto the residual stream (i.e., $x \mapsto x + f(x)$) maps values deviating from $-1, 0, 1$ by at most $\frac{1}{4}$ back to $-1, 0, 1$.

the attention and removes the error with the MLP, while the second one contains the MLP from the original layer and does not use the attention. The more general statement is Theorem D.17, applied to our constructions, it yields the following statement:

**Theorem 4.2.** *Consider one of the hardmax results from the previous section and denote $\hat{l} = \hat{n}$ for Proposition 3.1, $\hat{l} = \hat{t}$ for Theorem 3.2 and $\hat{l} = \hat{s}$ for Theorem 3.3. Then there exists*

$$c_0(\hat{l}) = \mathcal{O}\big((\log \hat{l})^{\frac{3}{4}}\big)$$

*such that for each $c \geq c_0(\hat{l})$ the following holds: in the respective result, the hardmax transformer can be replaced by a softmax transformer with $\{0, \pm 1, \pm 2, \pm c\}$-valued weights satisfying the same asymptotic bounds as the hardmax transformer. Further, this still holds when rounding all activations to a floating-point format containing $c$ and having at least 1 mantissa bit and 3 exponent bits, and rounding attention weights to a floating-point format with at least 4 mantissa and at least $b_0(\hat{l}) = \mathcal{O}(\log \log \hat{l})$ exponent bits.*

Again, the activation precision requires $\mathcal{O}(\log \log \log \hat{l})$ exponent bits to represent a sufficiently large $c$. The proof is given in Appendix D.5.1.

Note that using MLPs for removing errors relies heavily on the ternary activations: if instead activations have entries in some arbitrary set, the size of the MLP mapping perturbed activations back to this set would need to grow proportionally with the size of this set. This prevents such techniques

from being properly applicable e.g. in (Merrill & Sabharwal, 2024; Yang et al., 2025) where activations have entries in a set growing linearly in the context size (Yang et al., 2026).

Combining Theorem 4.2 with Theorems 3.2 and 3.3 and Proposition 3.1 yields the statements from the introduction, concluding the theoretical part of this work.

### 4.3. Sparse Attention

An interesting property of our constructions is an *effective* attention sparsity: for every query position $i$, either the position $j$ maximizing the attention score $s_{ij}$ is unique, or all maximizers carry the same value $v_j$. Therefore, the head output is unchanged if we replace hardmax attention by a 1-sparse variant that selects any one maximizer $j^\star$ and returns $v_{j^\star}$. Computing the attention output for a given position is then a maximum-inner-product search (MIPS) problem. This connects our setting to practical sparse-attention approximations, which speed up attention by solving these MIPS problems approximately (Liu et al., 2025; Kitaev et al., 2020; Bertsch et al., 2023). Our results show that restricting to 1 retrieved token per query does not weaken expressivity for these constructions. See Appendix F.3 for further discussion.

## 5. Sudoku Experiments

To investigate the extent to which the predictions of our expressivity results align with learnability, we train small transformers to imitate a standard depth-first search algorithm on Sudokus, a common task for analyzing algorithmic reasoning capabilities of neural networks (Palm et al., 2018; Shah et al., 2024; Giannoulis et al., 2025; Ye et al., 2025; Wang et al., 2025a). We first identify a small model size for which SCoT learns the algorithm well and observe that CoT fails at long contexts with this same model size (Section 5.2), then show that scaling the model size cleanly extends the CoT context length while scaling activation precision does not (Section 5.3). As a bonus, we observe that SCoT models generalize from easy to hard Sudokus that require significantly longer reasoning (Section 5.4).

### 5.1. Method

We use the sudoku-extreme dataset introduced in (Wang et al., 2025a) consisting of around 4 million particularly difficult Sudokus and generate training data with a deterministic depth-first MRV backtracking solver with lexicographic tie-breaks. For SCoT, we split token sequences into segments with 512 non-summary tokens per segment, while summaries encode the solver's state. For training we use only puzzles where the solver CoT length (i.e., the number of tokens in the solver's CoT token sequence) is at most $N$, to keep CoT training computationally feasible. Throughout

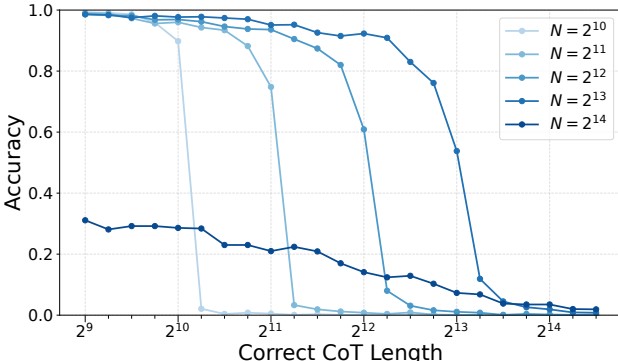

*Figure 5.* Accuracy of $L = 6$ CoT models with dimension 512 and 8 heads trained on puzzles requiring at most $N$ CoT tokens and evaluated on puzzles of varying CoT length.

this section, models are trained for 20 billion tokens (excluding padding) with mixed-precision bfloat16 activations unless stated otherwise. For evaluation, models generate deterministically (temperature 0) with CoT or SCoT and the generation is seen as correct if the final output is the correct solution of the input Sudoku. See Appendix E for details.

### 5.2. Small Models with SCoT and CoT

To compare the capabilities of small models with CoT and SCoT, we determine a minimal model size for which SCoT learns the algorithm. In particular, for dimension 512 and 8 heads, SCoT models with depths $L \in \{4, 5, 6, 7, 8\}$ achieve the following accuracies on a random subsample of 10000 test Sudokus:

| $L$ | 4 | 5 | 6 | 7 | 8 |
|---|---|---|---|---|---|
| Acc(%) | 16.3 | 97.2 | 99.5 | 99.7 | 99.6 |

The performance mostly saturates at depth $L = 6$ and this is robust across runs (Appendix E.4). Next, we train CoT models with this size on Sudokus with CoT length at most $N$ for $N \in \{2^{10}, 2^{11}, 2^{12}, 2^{13}, 2^{14}\}$. The models are evaluated on test puzzles grouped by solver CoT length, as detailed in Appendix E.3. The resulting CoT length vs. accuracy curves are shown in Figure 5. As is to be expected without context extension methods, CoT models trained on puzzles requiring at most $N$ tokens fail on puzzles requiring significantly more tokens. However, the observation to make here is that they already underperform badly well before that boundary for $N = 2^{14}$. This indicates that this model size, which suffices for SCoT, is insufficient for learning to perform the algorithm at such long context lengths with CoT. Longer training of the $N = 2^{14}$ run only partially mitigates this failure (see Figure 8 in the Appendix).

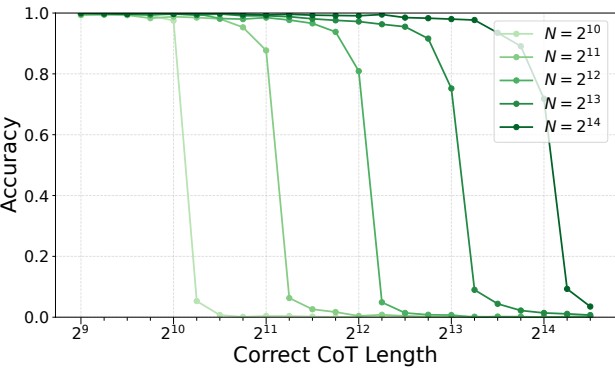

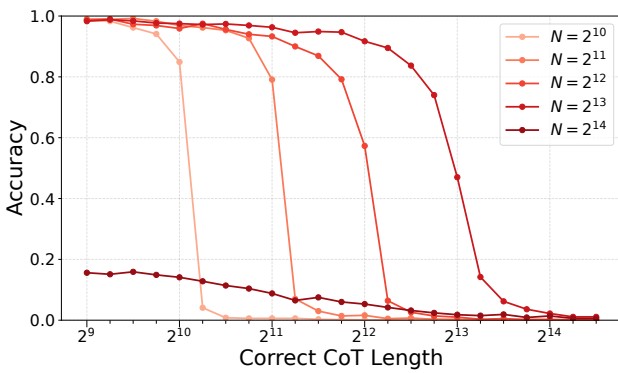

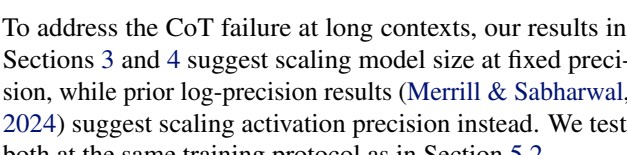

*(a)* Same as Figure 5 but with depth $L = 8$.      *(b)* Same as Figure 5 but with fp32 instead of bfloat16 activations.

### 5.3. Increasing Model Size vs. Activation Precision

To address the CoT failure at long contexts, our results in Sections 3 and 4 suggest scaling model size at fixed precision, while prior log-precision results (Merrill & Sabharwal, 2024) suggest scaling activation precision instead. We test both at the same training protocol as in Section 5.2.

**Model size.** Repeating the CoT training and evaluation runs with the depth increased from $L = 6$ to $L = 8$ yields the accuracy curves shown in Figure 6a. The depth-8 CoT models clearly perform well and do not fail at the longest context length $N = 2^{14}$, hence the increased capacity actually leads to improved learnability in this case.

**Activation precision.** Training and evaluating the depth $L = 6$ CoT runs with full fp32 activations instead of mixed-precision bfloat16 yields the accuracy curves shown in Figure 6b. These are qualitatively identical to the bfloat16 ones; in particular, the model trained on $N = 2^{14}$ still fails to reach good accuracy. This is consistent with the common modern opinion that mixed precision mostly does not impact performance (Micikevicius et al., 2018).

### 5.4. Generalization and Ultra-Long Reasoning

As a bonus, we observe extreme generalization from easy to hard Sudokus for SCoT in this setting: a larger SCoT model ($d = 768, L = 12, H = 12$) trained on Sudokus with CoT length at most $N = 2^{12}$ achieves near-perfect 99.96% accuracy on a random 10k subsample of the test set and solves 92 out of the 100 hardest (by CoT trace length) test puzzles, each of which requires between 1 and 9 million CoT tokens. See Appendix E.5 for details.

This generalization ability relies on the fact that the algorithm solves Sudokus of different difficulty with the same space requirements, so the same segment length can be used regardless of the total reasoning length required. We leave it to future work to further investigate under which circumstances SCoT enables this *constant-space generalization*.

## 6. Conclusion and Future Work

We show that standard transformer decoders with softmax attention, rounding of activations and attention weights and moderate parameter magnitudes can simulate Turing machines using CoT and more efficiently using SCoT, thereby narrowing the gap between the models used in theoretical expressivity results and practical ones. An obvious future avenue is to investigate, both theoretically and empirically, whether the logarithmic scaling of model size in these results is actually necessary.

Apart from that, our theoretical results only investigated *expressivity* and not *learnability*, i.e., whether such transformers can be found through training methods based on gradient descent. In this regard, our experiments merely provide a first step by showing that, in the setting of Sudoku reasoning, our expressivity results make better predictions about learnability than prior high-precision ones.

## Acknowledgements

We thank the reviewers for their constructive feedback, which in particular helped us improve the general clarity of our presentation as well as the framing of the experiments with respect to the theory. Further, the authors are grateful for support by the German Research Foundation through Project 553088969 as well as the Cluster of Excellence "Machine Learning — New Perspectives for Science" (EXC 2064/1 number 390727645).

## Impact Statement

This paper presents work whose goal is to advance the field of machine learning. There are many potential societal consequences of our work, none of which we feel must be specifically highlighted here.

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

# A. Related Work

Here we discuss related works including but not limited to the ones mentioned in the main part already.

## A.1. Transformer Turing Completeness

| Work | Depth | Width | Prec. | Steps | Context | Parameters | Arch. |
|---|---|---|---|---|---|---|---|
| (Pérez et al., 2021) | $\mathcal{O}(1)$ | $\mathcal{O}(1)$ | $\mathcal{O}(\log \hat{t})$ | $\mathcal{O}(t)$ | $\mathcal{O}(t)$ | $\mathcal{O}(1)$ | ED-CoT-H |
| (Bhattamishra et al., 2020) | $\mathcal{O}(1)$ | $\mathcal{O}(1)$ | $\mathcal{O}(\hat{s})$ | $\mathcal{O}(t)$ | $\mathcal{O}(t)$ | $\mathcal{O}(1)$ | ED-Con-H |
| (Merrill & Sabharwal, 2024) | $\mathcal{O}(1)$ | $\mathcal{O}(1)$ | $\mathcal{O}(\log \hat{t})$ | $\mathcal{O}(t)$ | $\mathcal{O}(t)$ | $\mathcal{O}(1)$ | D-CoT-H |
| (Yang et al., 2025) | $\mathcal{O}(1)$ | $\mathcal{O}(1)$ | $\mathcal{O}(\log \hat{t})$ | $\mathcal{O}(t)$ | $\mathcal{O}(s)$ | $\mathcal{O}(1)$ | D-SCoT-H |
| (Wei et al., 2022a) | $\mathcal{O}(\log \hat{t})$ | $\mathcal{O}(\log \hat{t})$ | $\mathcal{O}(\log \log \hat{t})$ | $\mathcal{O}(\hat{t})$ | $\mathcal{O}(\hat{t})$ | $\mathcal{O}((\log \hat{t})^3)$ | ED-Con-H |
| (Li et al., 2024) | $\mathcal{O}(1)$ | $\mathcal{O}(\log \hat{t})$ | $\mathcal{O}(1)$ | $\mathcal{O}(t \log t)$ | $\mathcal{O}(t \log t)$ | $\mathcal{O}(\hat{t}(\log \hat{t})^2)$ | D-CoT-Sfp |
| (Li & Wang, 2026) | $\mathcal{O}(1)$ | $\mathcal{O}(1)$ | $\mathcal{O}(1)$ | $\mathcal{O}(t \hat{s}^\varepsilon)$ | $\mathcal{O}(\hat{s})$ | $\mathcal{O}(1)$ | D-CoT-H |
| (Jiang et al., 2026) | $\mathcal{O}(1)$ | $\mathcal{O}(1)$ | $\mathcal{O}(\hat{t})$ | $\mathcal{O}(\exp(t))$ | $\mathcal{O}(\exp(t))$ | $\mathcal{O}(1)$ | D-CoT-S |
| Ours (CoT) | $\mathcal{O}(\log \hat{t})$ | $\mathcal{O}(\log \hat{t})$ | $\mathcal{O}(\log \log \hat{t})$ | $\mathcal{O}(t)$ | $\mathcal{O}(t)$ | $\mathcal{O}((\log \hat{t})^3)$ | D-CoT-S |
| Ours (SCoT) | $\mathcal{O}(\log \hat{s})$ | $\mathcal{O}(\log \hat{s})$ | $\mathcal{O}(\log \log \hat{s})$ | $\mathcal{O}(t)$ | $\mathcal{O}(s)$ | $\mathcal{O}((\log \hat{s})^3)$ | D-SCoT-S |

*Table 2.* Asymptotic scalings of prior results showing existence of a transformer simulating a given Turing machine $M$ on all inputs $w$ with $t = t_M(w) \leq \hat{t}$ and $s = s_M(w) \leq \hat{s}$ for time bound $\hat{t}$ and space bound $\hat{s}$. Arch. describes the transformer architecture as D/ED–CoT/SCoT/Con–H/S/Sfp (decoder-only / encoder–decoder; token CoT / summarized token CoT / continuous CoT; hardmax / softmax / softmax relying on finite-precision).

Many prior works show Turing completeness in some sense for some type of transformer, often but not always by simulating Turing machines directly. While these works differ substantially in the exact kind of result, we try to frame them to be comparable to our results. In particular, we try to phrase them as constructing a transformer simulating a Turing machine $M$ for all inputs $w$ with $t_M(w) \leq \hat{t}$ and $s_M(w) \leq \hat{s}$. Model depth, width, precision, decoding steps and context size for a given input $w$ are then bounded by $t_M(w), s_M(w), \hat{t}$ and $\hat{s}$. This distinction between *known* upper bounds on space and time usage of $M$ and the input-dependent actual space and time usages $s_M(w), t_M(w)$ has sometimes been overlooked in prior works, but leads to important differences when trying to choose $\hat{t}$ and $\hat{s}$ very large in order to cover many inputs with the same model. The results are summarized in Table 2. For simplicity we assume $t_M(w) \geq |w|$ so that e.g. $\mathcal{O}(t_M(w) + |w|) = \mathcal{O}(t_M(w))$, which is not a real restriction as it is already satisfied if the Turing machine always reads its input fully. Note that most works don't explicitly round activations, and we made our best effort to determine the precision required by them. In particular, if a construction uses activations of the form $\frac{1}{i}$ at position $i$, we write its precision as logarithmic in the context size even if they don't explicitly bound the error from rounding such numbers to floating-point formats. Note that our constructions require the $\mathcal{O}(\log \log \hat{t})/\mathcal{O}(\log \log \hat{s})$ precision only for attention weights, while the number of exponent bits for activations is triple logarithmic and the number of mantissa bits is constant for both.

(Pérez et al., 2021) is the fundamental work on Turing completeness of transformers, and many later works reuse techniques from it. They use an encoder-decoder transformer to simulate a Turing machine with one decoding step per Turing machine step encoding state, written symbols and head movements.

(Bhattamishra et al., 2020) prove Turing completeness by showing that a transformer can simulate an RNN and then invoking the Turing completeness of RNNs (Siegelmann & Sontag, 1992). Since the RNN construction encodes the Turing machine configuration in the precision of its hidden state, this requires $\mathcal{O}(\hat{s})$ precision bits (and additionally $\mathcal{O}(\log \hat{t})$ bits for positional information, typically negligible), and hence the same is true for this result.

(Merrill & Sabharwal, 2024) adapts techniques from (Pérez et al., 2021) to prove Turing completeness for decoder-only transformers, using some architectural modifications like strict causal masking (instead of the standard non-strict masking) and projected pre-norm. As they use a decoder with CoT and produce a comparable token sequence to our CoT construction, we chose them as the main prior comparison point for our CoT results in the main part.

(Yang et al., 2025) introduce a memory-efficient alternative Chain-of-Thought strategy which has Summarized CoT as a special case. For this special case they prove memory-efficient Turing completeness by using similar construction methods as (Merrill & Sabharwal, 2024) to generate token sequences alternating between chunks encoding Turing machine steps and chunks encoding the Turing machine's configuration, much like our SCoT construction (although the exact encodings differ in some non-important ways). In particular, they use the same idea of aligning the number of simulated steps to the size of the initial summary in each segment in order to have decoding steps and context size aligned with the Turing machine's time

and space usage respectively. Hence, we chose them as the main prior comparison for our SCoT results.

(Wei et al., 2022a) has the most technical overlap with our hardmax CoT constructions, even though their main goal in the paper is to prove finite sample bounds. In particular, their construction also uses ternary activations[2] with logarithmic width to represent positions in binary and logarithmic depth for performing a binary search for the latest write at a head position. Also they use the same kind of denoising MLP, but do this for proving finite sample bounds instead of removing actual errors from softmax and rounding. Nonetheless, their work would be very much suited for softmax attention and rounding to low-precision for the same reasons that motivated our hardmax constructions, namely separation of attention scores and ternary activations. However, their work also differs in several key aspects from our hardmax constructions. On the one hand, they use an encoder-decoder architecture as (Pérez et al., 2021). On the other hand, they do not generate a token sequence, but instead feed the final hidden representation at each position in the decoder directly into the next position skipping unembedding and embedding, i.e. $x_i^{(0)} = x_{i-1}^{(L)} + \text{posemb}_i$ . This is akin to modern *latent reasoning* or *continuous Chain-of-Thought* methods (Hao et al., 2025) and allows for dropping a lot of the complexity in our constructions, as the prior token's head position can just be fed through to the next token instead of having to reconstruct it at every step. However, this also forces them to use a fixed number of decoding steps until stopping. Note that one could add unembeddings and embeddings to map the hidden representations to a finite vocabulary, but this vocabulary would have size proportional to $\hat{t}$ instead of being fixed for $M$. The continuous CoT method also makes it difficult to write actual outputs in $\Sigma^*$ (or summaries for SCoT) and hence they only treat sequence classification, i.e. simulating Turing machines for language recognition.

(Li et al., 2024) simulate boolean circuits with transformers, where the number of decoding steps corresponds to the size of the boolean circuit. Boolean circuits can simulate Turing machines with a logarithmic overhead as noted by (Li & Wang, 2026). (Li et al., 2024) also use a logarithmic embedding dimension and low precision activations and consider softmax attention with an explicit rounding model. They however use rounding explicitly to remove errors from softmax conversion, i.e. for a fixed chosen precision they choose the scaling $c$ sufficiently large so that softmax weights are rounded back to the correct hardmax weights (which are always $0$ or $1$ in their construction). Oddly, when using this technique, one fixed transformer can fail when the activation precision is too high. Furthermore, the boolean circuit is directly encoded in the positional encodings, which requires $\mathcal{O}(nd) = \mathcal{O}(n \log n)$ parameters for context size $n$.

(Li & Wang, 2026), building on (Li & Wang, 2025), show space-efficient Turing completeness via simulating *multi-queue machines*. Notably, they appear to be the only work which achieves constant width, depth and precision at the same time. However, their construction is also not fully uniform: they rely on non-standard positional encodings to select tokens at specific offsets depending on the known space bound $\mathcal{O}(\hat{s})$. Unfortunately this raises the number of generated tokens per Turing machine step to $\mathcal{O}(\hat{s}^\varepsilon)$, where $\varepsilon$ can be made arbitrarily small by increasing the number of attention heads. Furthermore, context size is not adaptive as in (Yang et al., 2025) and our SCoT results, but fixed to $\hat{s}$.

(Jiang et al., 2026) show Turing completeness for softmax transformers with fixed model size and precision logarithmic in context size. Interestingly they do not approximate hardmax transformers with softmax transformers as we do, but use softmax transformers directly to simulate counter machines, which are Turing complete (Minsky, 1967). Unfortunately, counter machines can have exponential overhead when simulating Turing machines (Fischer et al., 1968), hence the number of steps and context size scale poorly when simulating Turing machines. This also implies that the required precision in their construction is actually linear in the number of simulated Turing machine steps as $\mathcal{O}(\log(\exp(\hat{t}))) = \mathcal{O}(\hat{t})$.

Two other works on universal computation with transformers and CoT that don't quite fit into the comparison above are (Qiu et al., 2025) and (Nowak et al., 2024). The former one analyzes the aspect of uniformity over Turing machines $M$, i.e. constructing one transformer which can perform any task when given the right prompt. The latter one extends the relationship between standard (i.e. deterministic) Turing machines and transformers evaluated with temperature $0$ to a relationship between probabilistic Turing machines and transformers where the next token is actually sampled from the predicted next-token distribution.

### A.2. Regular Language Recognition with Transformers

Our approach to recognizing regular languages with log-depth transformers closely follows that of (Liu et al., 2023) except for using logarithmic width instead of logarithmic precision to represent positions. Interestingly, they even convert their hardmax transformer to softmax as well by scaling query and key projections. However, due to constant embedding size, separation shrinks polynomially with input length and $W_Q, W_K$ need to grow as $\mathcal{O}(\hat{n} \log \hat{n})$ in their construction for a

---

[2]They frame it as binary activations as hidden representations are binary, but queries and keys are ternary in their construction.

length bound $\hat{n}$. Moreover, they neither control the error from using softmax attention end-to-end like in Theorem 4.1 nor by inserting additional MLPs as in Theorem 4.2, but instead construct their MLPs to be invariant under small perturbations and hence remove the errors in parallel to their computation. This elegantly avoids the doubling of depth incurred by our MLP technique, but is unfortunately not directly applicable to our constructions without raising the MLP width to at least linear in context size.

Another work on regular language recognition with log-depth transformers is (Merrill & Sabharwal, 2025). While their theoretical advantage over (Liu et al., 2023) lies mainly in *looping* the same fixed transformer layers instead of each layer being different, which our results do not consider, they offer interesting empirical results. In particular, their experiments indicate that logarithmic scaling of depth might actually be necessary for training transformers to simulate certain automata.

### A.3. Other Transformers Expressivity Works

Two other works on transformers have significant technical overlap with our softmax and low-precision analyses and hence should be discussed. (Sanford et al., 2023; 2024) analyze standard softmax self-attention and use scaling to sharpen scores, and (Sanford et al., 2024) additionally relies on finite-precision rounding to exactly recover hardmax outputs (similar to (Li et al., 2024)). Importantly, as pointed out by (Yang et al., 2026), due to the logarithmic activation precision used in (Sanford et al., 2024), replacing the reliance on rounding with a denoising MLP similar to our constructions would require MLP width linear in the context size. This is in contrast to (Li et al., 2024), where the denoising MLP technique could be applied to replace the reliance on rounding without large MLP widths.

Furthermore, there is a substantial number of works showing relatively poor representational ability of transformers under different assumptions when fixing model size (Hahn, 2020; Hao et al., 2022; Merrill et al., 2022; Yang et al., 2024; Chiang et al., 2023; Merrill & Sabharwal, 2023). Relatedly, (Amiri et al., 2025) proves lower bounds on the required CoT length for certain problems. These are orthogonal to our results, as we don't investigate the ability of a fixed size transformer to handle arbitrarily long contexts and instead let depth and width grow logarithmically with context size.

### A.4. Memory-Efficient Long Reasoning

The idea of decoupling total reasoning length and context size in CoT transformers has appeared in several forms recently.

(Yan et al., 2026) (InftyThink) is almost the same as our SCoT, as stated in the introduction. The main difference is that for InftyThink, the input always remains in context, i.e. each segment is prompted with the concatenation of the input and the previous summary instead of just the previous summary in our considerations. This is however not a significant difference, as with our SCoT the model could carry along the input in each summary if needed, while for InftyThink the input could simply be ignored. We made this choice for simplicity and better compatibility with Turing machine simulation.

(Aghajohari et al., 2026) (MarkovianThinker) is very close in spirit, but with a subtle difference: there are no explicit summary delimiters. Instead, decoding proceeds in chunks of at most $N$ tokens and each segment is prompted with the original input together with the final chunk of the previous segment consisting of the last $n < N$ tokens. Since $N, n$ are fixed hyperparameters for a run, the effective context size is not input-adaptive, which would make the context scale as $\mathcal{O}(\hat{s})$ rather than $\mathcal{O}(s_M(w))$ when aiming for an analogue of Theorem 3.3.

(Yang et al., 2025) (PENCIL) introduce a different memory-efficient Chain-of-Thought variant which has SCoT as a special case. Out of these three works, (Yang et al., 2025) is the only one showing theoretical results and they show them for the SCoT special case as discussed above.

Lastly, (Xiao et al., 2024) do not quite fit into this list, but are still worth mentioning. They propose fixing a context size and dropping the oldest token for each newly generated one. While their goal is not to extend reasoning length with limited context size, their method and findings could still be applied to decouple reasoning length and context size by writing summaries frequently. Similar to (Aghajohari et al., 2026), however, this does not directly allow the model to adapt the context size to the problem in the same way that the summary trigger tokens do.

### A.5. Generalization

Length generalization, i.e. training on shorter/smaller inputs and generalizing to longer/larger ones, is a well-studied challenge for transformers and other architectures (Jelassi et al., 2023; Zhou et al., 2024; Schwarzschild et al., 2021). Even with careful tuning of positional encodings and training procedures, large generalization factors remain difficult to achieve

reliably, although apparently possible in some cases with recurrent networks (Bansal et al., 2022). The constant-space generalization observed in our Sudoku experiments (Section 5) is a fundamentally different notion: the segment lengths remain similar between training and test while the total reasoning length varies by orders of magnitude. While this is significantly simpler than length generalization, the large factors achieved nonetheless do not seem to be found in prior works for tasks of comparable complexity. Note that constant-space generalization only becomes meaningful with SCoT or similar schemes that decouple reasoning length from context size.

## B. Definitions

Here we aim to refine the definitions from Section 2, so that the statements and proofs in the rest of the appendix leave no room for interpretation.

### B.1. General Notation

First we define the binary representation of a number using $r$ bits. We always write the least significant bit first and use the alphabet $\{-1, 1\}$ instead of the standard $\{0, 1\}$ to encode bits. The reason for this becomes clear in Lemma B.2 below.

**Definition B.1** (Binary Representation). Let $r \in \mathbb{N}$ and $i \in \mathbb{N}_0$ with $0 \leq i < 2^r$. Define

$$\mathrm{bin}_r^s(i) := \begin{cases} 1, & \left\lfloor \dfrac{i}{2^s} \right\rfloor \equiv 1 \pmod 2, \\ -1, & \left\lfloor \dfrac{i}{2^s} \right\rfloor \equiv 0 \pmod 2. \end{cases} \qquad \text{for } s \in \{0, 1, \ldots, r - 1\}$$

and

$$\mathrm{bin}_r(i) = (\mathrm{bin}_r^0(i), \ldots, \mathrm{bin}_r^{r-1}(i)) \, .$$

For example, $\mathrm{bin}_4(11) = (1, 1, -1, 1)$, which is obtained by taking the standard binary representation of 11, i.e. $(1, 0, 1, 1)$, reversing it and replacing 0's with $-1$'s.

**Encodings.** For a finite set $S$ we write $\mathrm{enc}_S : S \to \{-1, 1\}^{d_S}$ for an arbitrary fixed injective encoding (e.g. $\mathrm{enc}_Q$, $\mathrm{enc}_\Sigma$, $\mathrm{enc}_\Gamma$), the exact choice does not matter. For numerical values $i \in \{0, \ldots, 2^r - 1\}$, however, we always use the standard binary encoding $\mathrm{bin}_r(i)$ (instead of an arbitrary encoding $\mathrm{enc}_{\{0, \ldots, 2^r - 1\}}$), since MLPs need to efficiently implement arithmetic operations such as increments and decrements on these representations.

The reason we use $\{-1, 1\}$ rather than $\{0, 1\}$ is that it works better as keys/queries in attention via the following simple separation property.

**Lemma B.2.** *Let* $q, k \in \{-1, 1\}^r$ *for some* $r \in \mathbb{N}$ *with* $q \neq k$. *Then*

$$\langle q, q \rangle \geq \langle q, k \rangle + 2$$

*In particular, this holds for* $q = \mathrm{bin}_r(i)$ *and* $k = \mathrm{bin}_r(j)$ *with* $i \neq j$.

*Proof.* Straightforward: each coordinate contributes $+1$ if the bits match and $-1$ if they differ, so a single differing bit reduces the inner product by 2. □

### B.2. Deterministic Finite Automata

This is standard and repeated from the preliminaries section in the main part for completeness here:

**Definition B.3** (Deterministic Finite Automaton (DFA)). A deterministic finite automaton (DFA) $M$ is a 5-tuple

$$M = (Q, \Sigma, \delta, q_{\mathrm{init}}, F)$$

where $Q$ is a finite set of *states*, $\Sigma$ a finite *alphabet*, $\delta : Q \times \Sigma \to Q$ the *transition function*, $q_{\mathrm{init}} \in Q$ the *initial state* and $F$ a set of *accepting states*. For $w \in \Sigma^*$ with $|w| = n$ we write $w = (w_0, \ldots, w_{n-1})$ and define the sequence of states of $M$ on $w$ recursively as $q_0 = q_{\mathrm{init}}$ and $q_{t+1} = \delta(q_t, w_t)$ for $t = 0, \ldots, n - 1$. We define $L(M)$ as the set of $w \in \Sigma^*$ such that the final state $q_{|w|}$ is in $F$.

## B.3. Turing Machines

We formally fix the conventions and notation for the Turing machines being simulated by transformers in our constructions. We work with multi-tape Turing machines, each infinite to the right. On input $w = w_0 \dots w_{n-1} \in \Sigma^*$, the first tape contains $w_0, \dots, w_{n-1}$ in cells $0, \dots, n-1$ and $\sqcup$ elsewhere, all other tapes contain only $\sqcup$, and all heads start on cell 0.

**Definition B.4** (Multi-tape Turing Machine). A multi-tape Turing machine $M$ is an 8-tuple

$$M = (K, Q, \Sigma, \Gamma, \delta, \sqcup, q_{\text{init}}, q_{\text{halt}})$$

where $K \in \mathbb{N}$ is the number of tapes, $Q$ is a finite set of *states*, $\Sigma$ is a finite *input alphabet*, $\Gamma \supset \Sigma$ is a finite *tape alphabet*, $\sqcup \in \Gamma \setminus \Sigma$ is the *blank symbol*, $q_{\text{init}} \in Q$ is the *initial state*, $q_{\text{halt}} \in Q \setminus \{q_{\text{init}}\}$ is the *halting state*, and

$$\delta : Q \times \Gamma^K \to Q \times \Gamma^K \times \{L, S, R\}^K$$

is the *transition function*.

A configuration of a Turing machine $M$ consists of the state, the tape contents and the head positions.

**Definition B.5** (Configuration space). Let $M = (K, Q, \Sigma, \Gamma, \delta, \sqcup, q_{\text{init}}, q_{\text{halt}})$ be a multi-tape Turing machine. Then we define

$$\text{Config}(M) := Q \times (\Gamma^{\mathbb{N}_0})^K \times \mathbb{N}_0^K .$$

**Definition B.6** (Run, time, space usage and output). Let $M$ be a Turing machine as defined above and let $w = w_0 \dots w_{n-1} \in \Sigma^*$.

**Run on input.** The *run* of $M$ on input $w$ is the sequence of configurations

$$C_t = \left( q_t, (x_t^k)_{k=1}^K, (n_t^k)_{k=1}^K \right) \in \text{Config}(M) \qquad (t = 0, 1, 2, \dots)$$

where $q_t$ is the state after $t$ steps, $x_t^k \in \Gamma^{\mathbb{N}_0}$ the contents of the $k$-th tape and $n_t^k \in \mathbb{N}_0$ the position of the head on tape $k$. The initial configuration is

$$q_0 = q_{\text{init}}, \qquad x_0^1 = (w_0, \dots, w_{n-1}, \sqcup, \sqcup, \dots), \qquad x_0^k = \sqcup^{\mathbb{N}_0} \text{ for } k \geq 2,$$

and

$$n_0^k = 0 \quad \text{for all } k = 1, \dots, K.$$

For $t = 1, 2, \dots$ we set

$$\left( q_t, (y_t^k)_{k=1}^K, (\Delta_t^k)_{k=1}^K \right) = \delta \left( q_{t-1}, \left( (x_{t-1}^k)_{n_{t-1}^k} \right)_{k=1}^K \right),$$

and define

$$(x_t^k)_i = \begin{cases} y_t^k, & \text{if } i = n_{t-1}^k, \\ (x_{t-1}^k)_i, & \text{otherwise,} \end{cases} \qquad n_t^k = \begin{cases} n_{t-1}^k + 1, & \text{if } \Delta_t^k = R, \\ \max(0, n_{t-1}^k - 1), & \text{if } \Delta_t^k = L, \\ n_{t-1}^k, & \text{if } \Delta_t^k = S. \end{cases}$$

**Halting and running time.** We say that $M$ *halts* on input $w$ if there exists $t \geq 0$ such that $q_t = q_{\text{halt}}$. In this case we set

$$t_M(w) := \min\{t \geq 0 \mid q_t = q_{\text{halt}}\},$$

and if no such $t$ exists we set $t_M(w) := \infty$.

**Output.** If $M$ halts on $w$ and at time $t_M(w)$ the contents of tape 1 are of the form

$$x_{t_M(w)}^1 = (y_0, \dots, y_{m-1}, \sqcup, \sqcup, \dots)$$

for some $y = (y_0, \dots, y_{m-1}) \in \Sigma^*$, we define the *output* of $M$ on $w$ as $f_M(w) := y$.

**Space usage.** The *space usage* of $M$ on input $w$ is defined as

$$s_M(w) := \begin{cases} \max\left( \{|w|\} \cup \{1 + n_t^k \mid 0 \leq t \leq t_M(w), \, k \in \{1, \dots, K\}\} \right), & \text{if } t_M(w) < \infty, \\ \infty, & \text{if } t_M(w) = \infty. \end{cases}$$

## B.4. Transformers

The aim here is to define precise function classes for the MLPs, Attention layers, transformer layers, transformer stacks (sequence of transformer layers) and full transformers (transformer stack with added embedding and unembeddings) for both hardmax and softmax attention. We only define and use standard transformer decoders as used in LLMs, hence attention is always causal (i.e. position $i$ can attend to positions $j \leq i$).

For a set $X$ we write $X^n$ for the set of length-$n$ sequences $(x_0, \ldots, x_{n-1})$ with $x_i \in X$ and

$$X^+ := \bigcup_{n \geq 1} X^n$$

for the set of all non-empty finite sequences over $X$. We write

$$f : X^+ \overset{\text{lp}}{\to} X^+$$

if $f$ is *length-preserving*, i.e., if $f(x_0, \ldots, x_{n-1}) \in X^n$ for all $n \geq 1$.

**Definition B.7** (MLP). Let $d, d_{\text{ff}} \in \mathbb{N}$ (*input* and *hidden dimension*). Define the parameter space

$$\Theta^{\text{MLP}}(d, d_{\text{ff}}) := \mathbb{R}^{d_{\text{ff}} \times d} \times \mathbb{R}^{d_{\text{ff}}} \times \mathbb{R}^{d \times d_{\text{ff}}}.$$

For $\theta = (W_1, b, W_2) \in \Theta^{\text{MLP}}(d, d_{\text{ff}})$ referred to as *up projection, bias* and *down projection*, we define $F_\theta : \mathbb{R}^d \to \mathbb{R}^d$ as

$$F_\theta(x) = W_2 \operatorname{ReLU}(W_1 x + b),$$

where $\operatorname{ReLU}(t) = \max(0, t)$ is applied component-wise. We define the function class of all such MLPs as

$$\mathcal{F}(d, d_{\text{ff}}) := \{F_\theta \mid \theta \in \Theta^{\text{MLP}}(d, d_{\text{ff}})\}.$$

**Definition B.8** (Softmax). Let $(s_0, \ldots, s_{n-1}) \in \mathbb{R}^n$. Define

$$\operatorname{softmax}\big((s_j)_{j=0}^{n-1}\big) := \left( \frac{e^{s_0}}{\sum_{j=0}^{n-1} e^{s_j}}, \ldots, \frac{e^{s_{n-1}}}{\sum_{j=0}^{n-1} e^{s_j}} \right).$$

Note that $\sum_{j=0}^{n-1} \operatorname{softmax}\big((s_j)_{j=0}^{n-1}\big)_j = 1$.

**Definition B.9** (Hardmax). Let $s_0, \ldots, s_{n-1} \in \mathbb{R}$. Define

$$J := \arg\max_j s_j = \{j \in \{0, \ldots, n-1\} \mid s_j = \max\{s_0, \ldots, s_{n-1}\}\}$$

and define $\operatorname{hardmax}(s_0, \ldots, s_{n-1}) = (y_0, \ldots, y_{n-1})$ where

$$y_j = \begin{cases} \dfrac{1}{|J|} & , \ j \in J \\ 0 & , \ j \notin J \end{cases}$$

Again note that $\sum_j y_j = 1$.

**Definition B.10** (Single Attention Head). Let $d, d_k, d_v \in \mathbb{N}$ (*input dimension, key-query dimension* and *value dimension*). Define the parameter space

$$\Theta^{\text{att}}(d, d_k, d_v) := \mathbb{R}^{d_k \times d} \times \mathbb{R}^{d_k \times d} \times \mathbb{R}^{d_v \times d} \times \mathbb{R}^{d \times d_v}.$$

For $\theta = (W_Q, W_K, W_V, W_O) \in \Theta^{\text{att}}(d, d_k, d_v)$ referred to as *query, key, value* and *output projection*, we define

$$A_\theta^{\text{hard}} : (\mathbb{R}^d)^+ \overset{\text{lp}}{\to} (\mathbb{R}^d)^+$$

as mapping $(x_0, \ldots, x_{n-1}) \in (\mathbb{R}^d)^n$ to $(y_0, \ldots, y_{n-1}) \in (\mathbb{R}^d)^n$ defined as follows. Define *queries, keys* and *values*

$$q_i = W_Q x_i, \qquad k_i = W_K x_i, \qquad v_i = W_V x_i,$$

and for $j \leq i$ define *attention scores*

$$s_{ij} = \frac{1}{\sqrt{d_k}} \langle q_i, k_j \rangle$$

and *attention weights*

$$(\alpha_{ij})_{j=0}^i = \mathrm{hardmax}((s_{ij})_{j=0}^i) \ .$$

Then the output at position $i$ is defined as

$$y_i = W_O \sum_{j=0}^i \alpha_{ij} v_j \ .$$

Analogously we define $A_\theta^{\mathrm{soft}}$ by replacing hardmax with softmax in the definition of the attention weights. The corresponding function classes are defined as

$$\mathcal{A}^{\mathrm{hard}}(d, d_k, d_v) := \{A_\theta^{\mathrm{hard}} \mid \theta \in \Theta^{\mathrm{att}}(d, d_k, d_v)\}$$

and analogously for $\mathcal{A}^{\mathrm{soft}}(d, d_k, d_v)$. Note that the scaling by $\frac{1}{\sqrt{d_k}}$ in the definition of the scores $s_{ij}$ does not influence the output of the hardmax (as hardmax is invariant under scaling scores by a positive constant) and does not influence the set of representable functions in both cases (hardmax and softmax) as it can be absorbed into the query or key projections. However, it is standard practice and influences the considerations in Appendix D, in particular the parameter scales for softmax conversion.

**Definition B.11** (Transformer Layer). Let $d, d_{\mathrm{ff}}, d_k, d_v \in \mathbb{N}$ and $H \in \mathbb{N}$ (the *number of attention heads*). Define the parameter space

$$\Theta^{\mathrm{TL}}(d, d_{\mathrm{ff}}, d_k, d_v, H) := \left( \Theta^{\mathrm{att}}(d, d_k, d_v) \right)^H \times \Theta^{\mathrm{MLP}}(d, d_{\mathrm{ff}}).$$

For

$$\theta = \left( (\theta^h)_{h=1}^H, \theta^{\mathrm{MLP}} \right) \in \Theta^{\mathrm{TL}}(d, d_{\mathrm{ff}}, d_k, d_v, H),$$

where $\theta^{\mathrm{MLP}} = (W_1, b, W_2) \in \Theta^{\mathrm{MLP}}(d, d_{\mathrm{ff}})$ and $\theta^h \in \Theta^{\mathrm{att}}(d, d_k, d_v)$ for $h = 1, \ldots, H$, we define the transformer layer

$$TL_\theta^{\mathrm{hard}} : (\mathbb{R}^d)^+ \xrightarrow{\mathrm{lp}} (\mathbb{R}^d)^+$$

as mapping $(x_0, \ldots, x_{n-1}) \in (\mathbb{R}^d)^n$ to $(y_0, \ldots, y_{n-1}) \in (\mathbb{R}^d)^n$ where

$$y_i = \tilde{y}_i + F_{\theta^{\mathrm{MLP}}}(\tilde{y}_i)$$

and

$$\tilde{y}_i = x_i + \sum_{h=1}^H A_{\theta^h}^{\mathrm{hard}}(x_0, \ldots, x_{n-1})_i \ .$$

The class of such transformer layers is defined as

$$\mathcal{TL}^{\mathrm{hard}}(d, d_{\mathrm{ff}}, d_k, d_v, H) := \{TL_\theta^{\mathrm{hard}} \mid \theta \in \Theta^{\mathrm{TL}}(d, d_{\mathrm{ff}}, d_k, d_v, H)\}.$$

Everything is defined analogously for softmax.

**Definition B.12** (Transformer Stack). Let $d, d_{\mathrm{ff}}, d_k, d_v, H \in \mathbb{N}$ and $L \in \mathbb{N}$ (the *depth*). Define the parameter space

$$\Theta^{\mathrm{TS}}(d, d_{\mathrm{ff}}, d_k, d_v, H, L) := \left( \Theta^{\mathrm{TL}}(d, d_{\mathrm{ff}}, d_k, d_v, H) \right)^L .$$

For

$$\theta = (\theta^1, \ldots, \theta^L) \in \Theta^{\mathrm{TS}}(d, d_{\mathrm{ff}}, d_k, d_v, H, L)$$

we define the transformer stack

$$TS_\theta^{\mathrm{hard}} : (\mathbb{R}^d)^+ \xrightarrow{\mathrm{lp}} (\mathbb{R}^d)^+$$

as follows. For $x = (x_0, \ldots, x_{n-1}) \in (\mathbb{R}^d)^n$ set $x^{(0)} := x$ and recursively

$$x^{(\ell)} := TL_{\theta^\ell}^{\mathrm{hard}}(x^{(\ell-1)}) \qquad \text{for } \ell = 1, \ldots, L.$$

Then $TS_\theta^{\text{hard}}(x) := x^{(L)}$. The corresponding function class is

$$\mathcal{TS}^{\text{hard}}(d, d_{\text{ff}}, d_k, d_v, H, L) := \{TS_\theta^{\text{hard}} \mid \theta \in \Theta^{\text{TS}}(d, d_{\text{ff}}, d_k, d_v, H, L)\}.$$

Everything is defined analogously for softmax.

For later reference, given an input sequence $x = (x_0, \ldots, x_{n-1})$ we write $x_i^{(0)}$ for the input embedding (including the positional embedding) at position $i$ and $x_i^{(\ell)}$ for the representation at position $i$ after the $\ell$-th layer. Within a layer, we additionally use

$$x_i^{(\ell-0.5)}$$

to denote the representation at position $i$ after applying attention (including the residual addition) but before the MLP block of layer $\ell$.

**Definition B.13** (Transformer). Let $d, d_{\text{ff}}, d_k, d_v, H, L \in \mathbb{N}$ and let $\mathcal{V}$ be a finite set (the *vocabulary*). Define the parameter space

$$\Theta^{\text{T}}(d, d_{\text{ff}}, d_k, d_v, H, L, \mathcal{V}) := \Theta^{\text{TS}}(d, d_{\text{ff}}, d_k, d_v, H, L) \times (\mathbb{R}^d)^{\mathcal{V}} \times (\mathbb{R}^d)^{\mathcal{V}}.$$

where $(\mathbb{R}^d)^{\mathcal{V}}$ denotes the set of families $(e_v)_{v \in \mathcal{V}}$ with $e_v \in \mathbb{R}^d$. For

$$\theta = \left(\theta^{\text{TS}}, (\text{emb}_\tau)_{\tau \in \mathcal{V}}, (\text{unemb}_\tau)_{\tau \in \mathcal{V}}\right) \in \Theta^{\text{T}}(d, d_{\text{ff}}, d_k, d_v, H, L, \mathcal{V})$$

referred to as *transformer stack parameters, embedding vectors* and *unembedding vectors* and a *positional embedding function*[3]

$$\text{posemb} : \mathbb{N}_0 \to \mathbb{R}^d$$

we define the (hardmax) transformer

$$T_\theta^{\text{hard}} : \mathcal{V}^+ \to \mathcal{V}$$

as follows. For a sequence $(\tau_0, \ldots, \tau_{n-1}) \in \mathcal{V}^n$ with $n \geq 1$ we index tokens from 0 to $n-1$ and set

$$x_i^{(0)} := \text{emb}_{\tau_i} + \text{posemb}(i) \in \mathbb{R}^d \qquad (i = 0, \ldots, n-1),$$

and write $x^{(0)} := (x_0^{(0)}, \ldots, x_{n-1}^{(0)}) \in (\mathbb{R}^d)^n$. Define

$$x^{(L)} := TS_{\theta^{\text{TS}}}^{\text{hard}}(x^{(0)}) = (x_0^{(L)}, \ldots, x_{n-1}^{(L)}) \in (\mathbb{R}^d)^n.$$

Using only the last representation $x_{n-1}^{(L)}$, define scores

$$s(\tau) := \langle \text{unemb}_\tau, x_{n-1}^{(L)} \rangle \qquad \text{for } \tau \in \mathcal{V}$$

and choose

$$T_\theta^{\text{hard}}(\tau_0, \ldots, \tau_{n-1}) := \arg\max_{\tau \in \mathcal{V}} s(\tau),$$

where ties may be broken arbitrarily (and do not occur in our constructions). The corresponding function class is

$$\mathcal{T}^{\text{hard}}(d, d_{\text{ff}}, d_k, d_v, H, L, \mathcal{V}) := \{T_\theta^{\text{hard}} \mid \theta \in \Theta^{\text{T}}(d, d_{\text{ff}}, d_k, d_v, H, L, \mathcal{V})\}.$$

Everything is defined analogously for softmax.

*Remark* B.14 (Positional Embeddings). The positional embedding function $\text{posemb}$ belongs to the transformer, but this is implicit in the definition above. We will always use a fixed, highly regular positional embedding which consists of the binary representation $\text{bin}_{d_{\text{pos}}}(i)$ of the position $i$ being written to some $d_{\text{pos}} < d$ coordinates; see Definition B.1. Importantly, the positional embeddings are not treated as parameters, otherwise the positional embeddings would add $Nd$ parameters for a context size limit $N$, which is much larger than the polylogarithmic (in $N$) number of parameters in our constructions.

---

[3]The positional embedding function is not listed in the parameters, as we only consider highly regular positional embeddings. See the remark below for details.

## B.5. Autoregressive Generation and (Summarized) Chain-of-Thought

**Definition B.15** (Autoregressive generation). Let $T$ be a transformer (hardmax or softmax) with vocabulary $\mathcal{V}$ and let $F \subset \mathcal{V}$ be a non-empty set of stopping symbols. For a sequence $(\tau_0, \dots, \tau_{n-1}) \in \mathcal{V}^n$ with $n \geq 1$ we define the generation

$$\mathrm{gen}(T, (\tau_0, \dots, \tau_{n-1}), F) \in \mathcal{V}^+$$

as follows. For $k = n, n+1, \dots$ set

$$\tau_k := T(\tau_0, \dots, \tau_{k-1}).$$

If the set

$$\{k \geq n \mid \tau_k \in F\}$$

is non-empty, let $m := \min\{k \geq n \mid \tau_k \in F\}$ and define

$$\mathrm{gen}(T, (\tau_0, \dots, \tau_{n-1}), F) := (\tau_0, \dots, \tau_m).$$

If this set is empty, the generation is undefined.

We will use transformers to compute functions $\Sigma^* \to \Sigma^*$ using (Summarized) Chain-of-Thought. We formalize how a transformer $T$ with vocabulary $\mathcal{V}$ maps an input word $w = (w_0, \dots, w_{n-1}) \in \Sigma^*$ to an output word in $\Sigma^*$ via special delimiter tokens.

**Definition B.16** (Chain-of-Thought). Let $T$ be a transformer with vocabulary $\mathcal{V}$ where $\{\texttt{<inp>}, \texttt{</inp>}, \texttt{<outp>}, \texttt{</outp>}\} \subset \mathcal{V}$, $\Sigma \subset \mathcal{V}$, and these sets are disjoint. For $w = (w_0, \dots, w_{n-1}) \in \Sigma^*$ consider

$$\tau := \mathrm{gen}\big(T, (\texttt{<inp>}, w_0, \dots, w_{n-1}, \texttt{</inp>}), \{\texttt{</outp>}\}\big)$$

whenever the generation is defined. If $\tau$ is of the form

$$\tau = (\texttt{<inp>}, w_0, \dots, w_{n-1}, \texttt{</inp>}, *, \texttt{<outp>}, u_0, \dots, u_{m-1}, \texttt{</outp>})$$

where $*$ denotes an arbitrary (possibly empty) sequence of tokens from $\mathcal{V}$, $m \geq 0$, $u_0, \dots, u_{m-1} \in \Sigma$ and $\texttt{<outp>}$ occurs exactly once in $\tau$, then we define

$$T_{\mathrm{CoT}}(w_0, \dots, w_{n-1}) := (u_0, \dots, u_{m-1}) \in \Sigma^*.$$

and

$$t_T(w) = |\tau|$$

is the total number of tokens in the generated sequence (including the input tokens). Otherwise (in particular if the generation does not finish or does not have this form), $T_{\mathrm{CoT}}(w_0, \dots, w_{n-1})$ and $t_T(w)$ are undefined.

**Definition B.17** (Summarized Chain-of-Thought). Let $T$ be a transformer with vocabulary $\mathcal{V}$ where $\{\texttt{<inp>}, \texttt{</inp>}, \texttt{<summ>}, \texttt{</summ>}, \texttt{<outp>}, \texttt{</outp>}\} \subset \mathcal{V}$, $\Sigma \subset \mathcal{V}$, and these sets are disjoint. For $w = (w_0, \dots, w_{n-1}) \in \Sigma^*$ set

$$\mathrm{prompt}_0 := (\texttt{<inp>}, w_0, \dots, w_{n-1}, \texttt{</inp>}).$$

Then for $t = 0, 1, 2, \dots$:

- Define

$$\mathrm{gen}_t := \mathrm{gen}\big(T, \mathrm{prompt}_t, \{\texttt{</outp>}, \texttt{</summ>}\}\big)$$

whenever this generation is defined; if it is undefined for some $t$, we say $T_{\mathrm{SCoT}}(w_0, \dots, w_{n-1})$ is undefined.

- If $\mathrm{gen}_t$ ends in $\texttt{</outp>}$ and can be written as

$$\mathrm{gen}_t = (\mathrm{prompt}_t, *, \texttt{<outp>}, u_0, \dots, u_{m-1}, \texttt{</outp>})$$

with $m \geq 0$, $u_0, \dots, u_{m-1} \in \Sigma$ and $\texttt{<outp>}$ occurring exactly once after $\mathrm{prompt}_t$, then we set

$$T_{\mathrm{SCoT}}(w_0, \dots, w_{n-1}) := (u_0, \dots, u_{m-1}) \in \Sigma^*$$

and stop the process. We then define the total generation length as

$$t_T(w) = \sum_t |\operatorname{gen}_t|$$

i.e. the total number of tokens produced across all segments (counting the prompt tokens repeated in each segment), and maximum segment length as

$$s_T(w) = \max_t |\operatorname{gen}_t|.$$

- If $\operatorname{gen}_t$ ends in `</summ>` and can be written as

$$\operatorname{gen}_t = (\operatorname{prompt}_t, *, \texttt{<summ>}, s_0, \ldots, s_{r-1}, \texttt{</summ>})$$

with $r \geq 1$, $s_0, \ldots, s_{r-1} \in \mathcal{V}$ and `<summ>` occurring exactly once after $\operatorname{prompt}_t$, then we set

$$\operatorname{prompt}_{t+1} := (\texttt{<summ>}, s_0, \ldots, s_{r-1}, \texttt{</summ>})$$

and continue with the next iteration.

If in some iteration neither of the above cases applies, or if the process never reaches the first case (termination with `</outp>` and a valid output as above), then $T_{\mathrm{SCoT}}(w_0, \ldots, w_{n-1})$ is undefined.

### B.6. Precision and Rounding

Here we introduce some notation for finite precision numbers and rounding. The definitions for floating point numbers and rounding are standard (IEEE, 2019) except that we ignore infinite and NaN values. We will often work with very low precision values in $\{-1, 0, 1\}$, and we also consider standard floating point numbers.

**Definition B.18** (Floating Point Numbers). We define the set of floating point numbers with $b_m \in \mathbb{N}$ *mantissa bits* and $b_e \in \mathbb{N}_{\geq 2}$ *exponent bits* as

$$\mathbb{F}_{b_m, b_e} = \underbrace{\{\pm(1 + t2^{-b_m})2^{\kappa} \mid t \in \{0, \ldots, 2^{b_m} - 1\}, \kappa \in \{E_{\min}, \ldots, E_{\max}\}\}}_{\text{normals}}$$
$$\cup \underbrace{\{\pm(t2^{-b_m})2^{E_{\min}} \mid t \in \{0, \ldots, 2^{b_m} - 1\}\}}_{\text{subnormals and } 0},$$

where $E_{\min} = 2 - 2^{b_e - 1}$ and $E_{\max} = 2^{b_e - 1} - 1$ are the smallest and largest exponent values.

Hence the smallest positive subnormal and smallest positive normal numbers are $2^{E_{\min} - b_m}$ and $2^{E_{\min}}$, respectively, and the largest positive number is $(2 - 2^{-b_m})2^{E_{\max}}$. We refer to numbers $x$ with

$$2^{E_{\min}} \leq |x| \leq (2 - 2^{-b_m})2^{E_{\max}}$$

as being in the *normal range*.

Next, we define for a finite set $F \subset \mathbb{R}$ the rounding operation $[\cdot]_F : \mathbb{R} \to F$ as

$$[x]_F := \arg\min_{y \in F} |x - y|,$$

i.e. rounding $x$ to the nearest number in $F$, where ties can be broken arbitrarily and don't occur in our considerations. The relative rounding error for floating point numbers in the normal range can be bounded as follows:

**Lemma B.19.** *Let $b_m, b_e \in \mathbb{N}$ with $b_e \geq 2$ and let $\mathbb{F}_{b_m, b_e}$ be as above. For $x \in \mathbb{R}$ in the normal range, i.e.*

$$2^{E_{\min}} \leq |x| \leq (2 - 2^{-b_m})2^{E_{\max}},$$

*set $y := [x]_{\mathbb{F}_{b_m, b_e}}$. Then*

$$|y - x| \leq 2^{-b_m - 1}|x|.$$

*Proof.* By symmetry it suffices to consider $x \geq 0$. Choose $k \in \mathbb{Z}$ such that $2^k \leq x < 2^{k+1}$. As $x$ is in the normal range, we have $k \in \{E_{\min}, \ldots, E_{\max}\}$. In the interval $[2^k, 2^{k+1}]$, the representable positive floats form a grid with step size at most $2^{k-b_m}$. Therefore rounding to the nearest representable float yields an absolute error at most $\frac{1}{2} \cdot 2^{k-b_m} = 2^{k-b_m-1}$. Since $x \geq 2^k$, we obtain $|y - x| \leq 2^{k-b_m-1} \leq 2^{-b_m-1}x$. □

**Definition B.20** (Ternary Weights and Activations). We say that a transformer $T = T_\theta$ (or transformer layer, MLP, etc.) specified by parameters $\theta$ has *ternary weights* if all weights in $\theta$ (i.e. all parameters except for MLP biases) have entries in $\{-1, 0, 1\}$. For a set of input sequences $\mathcal{X} \subset \mathcal{V}^+$ we say that a (hardmax) transformer $T$ has *ternary activations* on $\mathcal{X}$ if for every $(\tau_0, \ldots, \tau_{n-1}) \in \mathcal{X}$ and for all layers $\ell$, heads $h$, and positions $i$, all hidden representations $x_i^{(\ell)}$ and $x_i^{(\ell-0.5)}$, all queries $q_i^{(\ell,h)}$, keys $k_i^{(\ell,h)}$ and values $v_i^{(\ell,h)}$, and all hidden MLP activations after applying ReLU have entries in $\{-1, 0, 1\}$. Note that this makes no assumption on attention scores $s_{ij}^{(\ell,h)}$ and weights $\alpha_{ij}^{(\ell,h)}$, i.e. those are not meant when we use the term *activations*.

*Remark* B.21 (Biases). Biases occur only in the hidden layer of the MLP in our constructions. In the hardmax constructions, biases always lie in $\{-d, -d+1, \ldots, d-1, d\}$, while for the denoising MLP (see Appendix D.5) they can additionally be $\pm\frac{1}{4}$ and $\pm\frac{3}{4}$. Representing them would hence require $\mathcal{O}(\log d)$ mantissa (and $\mathcal{O}(\log\log d)$ exponent) bits, i.e. double-logarithmic in the context length. Alternatively, one could get completely rid of biases without changing any of the theorems in the main part by adding $d$ constant 1 dimensions (set at embedding) to the representations connected with weights 0 or $\pm 1$ to each MLP neuron to replace biases in $\{-d, \ldots, d\}$. For the denoising MLP technique (Theorem 4.2) this would increase the set of possible weights to $\{0, \pm\frac{1}{4}, \pm\frac{3}{4}, \pm 1, \pm 2, \pm c\}$. In both cases, activation precision stays exactly the same.

# C. Additional Material for Section 3

## C.1. General Construction Methods for Hardmax Transformers

Here we introduce the general methods we use to construct hardmax transformers without having to manually specify parameters. Many of these considerations are inspired by (Elhage et al., 2021). We also fix some notation used throughout the section and prove some basic lemmas.

**Flags and Registers.** We split the $d$ coordinates of the residual stream into disjoint *registers* and *flags*. A flag is an index $F \in \{1, \ldots, d\}$. A register is a tuple

$$I = (i_0, \ldots, i_{m-1}) \qquad \text{with } i_0, \ldots, i_{m-1} \in \{1, \ldots, d\} \text{ pairwise distinct,}$$

and we write $|I| := m$. For $x \in \mathbb{R}^d$ we define

$$x[I] := (x_{i_0}, \ldots, x_{i_{m-1}}) \in \mathbb{R}^{|I|}, \qquad x[F] := x_F \in \mathbb{R}.$$

For registers and flags $R_1, \ldots, R_k$ we use the shorthand $x[R_1, \ldots, R_k] := (x[R_1], \ldots, x[R_k])$. For each token and layer, each register stores either zeros or a $\{-1, 1\}$-binary number[4] in the sense of Definition B.1, while flags store bits in $\{0, 1\}$. Unless specified otherwise, registers and flags introduced later in a construction are zero-initialized in the embedding for all tokens, and we always take them to be pairwise disjoint, often without further mention. We use 0-based indexing and half-open slices for registers: $I[k] := i_k$ and $I[a:b] := (i_a, \ldots, i_{b-1})$. Lastly, for a register $I$ we write $\bar{I}$ for the indices in $\{1, \ldots, d\}$ not occurring in $I$.

**Building a transformer.** Note that if an attention/MLP module in a transformer does not write anything to a coordinate (i.e. outputs 0 at this coordinate), the coordinate retains its value from before the attention/MLP module due to the residual connection. To construct a hardmax transformer $T \in \mathcal{T}^{\mathrm{hard}}(d, d_{\mathrm{ff}}, d_k, d_v, H, L, \mathcal{V})$ we conceptually start with an empty transformer with embeddings all zero and no attention heads or mlp neurons to modify the residual stream. Some registers of the embedding or positional embedding are defined, and then gradually attention heads and mlp neurons modifying different parts of the residual stream are added. Equivalently, one can start with all attention and MLP parameters being zero, which means all multihead attention and MLP modules output all zeros, hence not modifying the residual stream. "Adding" an attention head or MLP neurons to a layer $\ell$ then corresponds to setting the parameters of the next all-zero attention head or the next all-zero MLP neurons to perform some operation. All such updates are additive. If we say that some module "writes" its output to a register $I$, then either the register will be zero before the operation or explicitly zeroed out.

---

[4]Note that to encode the number 0, the register will have all entries $-1$. If all entries are zero, it does not encode anything.

**Parallelism.** In the constructions for Theorems C.14 and C.22, some multi-layer operations overlap. Importantly, such overlapping operations and in general different operations scheduled in the same layer always either modify disjoint registers and flags or are gated by different token types and hence do not interfere with each other.

**Positional Embeddings.** In all of our hardmax transformer constructions we will use the following positional embeddings. For a constant $R \in \mathbb{N}$ determining the maximum context size $N = 2^R$, we choose a $R$-dimensional register $I_{\text{pos}}$ and set

$$\text{posemb}(i)[I_{\text{pos}}] = \text{bin}_R(i)$$

with other coordinates (those not in $I_{\text{pos}}$) initialized to zero. In words, the positional embedding simply writes the binary encoding of $i$ to some $R$ coordinates.

**Attention Heads.** When we say that we add an attention head (Definition B.10) to layer $\ell$ which attends with queries $I_Q$, keys $I_K$, values $I_V$ and writes to $I_O$, where $I_Q, I_K, I_V, I_O$ are (concatenations of) registers and flags with $|I_Q| = |I_K|$ and $|I_V| = |I_O|$, we mean: take the next unused head index $h \in \{1, \ldots, H\}$ in that layer and set its projections so that

$$q_i^{(\ell,h)} = x_i^{(\ell-1)}[I_Q], \qquad k_j^{(\ell,h)} = x_j^{(\ell-1)}[I_K], \qquad v_j^{(\ell,h)} = x_j^{(\ell-1)}[I_V],$$

and we often suppress the layer/head superscripts when clear. The output projection injects the head output into coordinates $I_O$ (so it is added to the residual stream on $I_O$). Concretely, we choose $W_Q, W_K, W_V, W_O$ as selector matrices: writing $I_Q = (i_1, \ldots, i_{|I_Q|})$, we set $(W_Q)_{j,i_j} = 1$ for $j = 1, \ldots, |I_Q|$ and zero elsewhere (and analogously for $W_K$ and $W_V$). Writing $I_O = (o_1, \ldots, o_{|I_O|})$, we set $(W_O)_{o_j,j} = 1$ for $j = 1, \ldots, |I_O|$ and zero elsewhere. Note that for these attention heads, projections always have ternary entries and the same activation precision as the $x_i^{(l-1)}$. If $|I_Q| = |I_K|$ is smaller than the key-query dimension $d_k$, we can simply set the remaining coordinates of queries and keys to zero without mentioning, similarly for the values. If not all heads in a layer are used, we set the remaining heads to zero (e.g. by setting $W_V = 0$).

**MLPs.** The MLP in layer $\ell$ is applied pointwise and added to the residual stream. Since its output is added, we sometimes say that an MLP *writes* a value into a register when that register is zero before the MLP, so that the post-MLP contents equal the written value. We describe MLPs in terms of register operations (e.g. zeroing a register, decrementing a binary representation, setting flags), often conditional on certain flags. Each operation uses a certain number of hidden neurons; if the total number of used neurons in a layer is smaller than $d_{\text{ff}}$, the remaining neurons simply do nothing (e.g. have zero output weights). Multiple operations can be implemented in parallel by combining their hidden neurons, as formalized by Lemma C.2.

The following two generic lemmas for MLPs will be used repeatedly. The first constructs a single-neuron MLP that fires precisely when certain registers are set to certain binary numbers and certain flags are set to certain truth values and outputs some ternary vector in this case and zero otherwise.

**Lemma C.1** (Single Neuron MLP). *Let $d \in \mathbb{N}$ and let $I_1, \ldots, I_n$ be registers and $F_1, \ldots, F_m$ be flags, all disjoint. Let $x_i^* \in \{-1, 1\}^{|I_i|}$ for $i = 1, \ldots, n$ and let $b_j^* \in \{0, 1\}$ for $j = 1, \ldots, m$. Let $y \in \{-1, 0, 1\}^d$. Define the admissible inputs as*

$$\mathcal{X} = \{x \in \{-1, 0, 1\}^d \mid x[F_j] \in \{0, 1\} \text{ for } j \in \{1, \ldots, m\}\} .$$

*Then there exists a single-neuron MLP $f \in \mathcal{F}(d, 1)$ such that*

$$f(x) = \begin{cases} y, & \text{if } x[I_i] = x_i^* \text{ for } i \in \{1, \ldots, n\} \text{ and } x[F_j] = b_j^* \text{ for } j \in \{1, \ldots, m\}, \\ 0, & \text{if } x \in \mathcal{X} \text{ but the above condition is not met}. \end{cases}$$

*Furthermore, $f$ has ternary weights and activations in the sense of Definition B.20 on these inputs.*

*Proof.* Set $F^+ := \{F_j \mid b_j^* = 1\}$. We define the input weight matrix $W_1 \in \mathbb{R}^{1 \times d}$ by

$$(W_1)_{1,s} := \begin{cases} (x_i^*)_k, & s = I_i[k] \text{ for some } i \in \{1, \ldots, n\} \text{ and } k \in \{0, \ldots, |I_i| - 1\}, \\ 1, & s \in F^+, \\ -1, & s \in \{F_1, \ldots, F_m\} \setminus F^+, \\ 0, & \text{otherwise}, \end{cases}$$

and set $b := -(\sum_i |I_i| + |F^+|) + 1$. For $x \in \mathcal{X}$ with the target pattern $x[I_i] = x_i^*, x[F_j] = b_j^*$ we obtain $W_1 x = \sum_i |I_i| + |F^+|$ and hence $\mathrm{ReLU}(W_1 x + b) = 1$. If $x \in \mathcal{X}$ differs from the target pattern in at least one coordinate in one of the registers or flags, then at that coordinate the contribution drops by at least 1, so $W_1 x \leq \sum_i |I_i| + |F^+| - 1$ and $\mathrm{ReLU}(W_1 x + b) = 0$.

Now choose $W_2 \in \mathbb{R}^{d \times 1}$ as $W_2 = y$ and the MLP implements the desired function. All entries of $W_1$ and $W_2$ lie in $\{-1, 0, 1\}$, so $f$ has ternary weights. On admissible inputs, the single hidden activation is always in $\{0, 1\}$, and the output is in $\{-1, 0, 1\}^d$, so $f$ has ternary activations in the sense of Definition B.20. □

The next lemma shows how to merge two MLPs into one that computes their sum by simply combining their hidden neurons.

**Lemma C.2.** *Let $f_1 \in \mathcal{F}(d, d_1)$ and $f_2 \in \mathcal{F}(d, d_2)$. Then there exists $f \in \mathcal{F}(d, d_1 + d_2)$ such that*

$$f(x) = f_1(x) + f_2(x) \qquad \text{for all } x \in \mathbb{R}^d.$$

*Furthermore, if $f_1$ and $f_2$ have ternary weights, then so does $f$. If, in addition, both $f_1$ and $f_2$ have ternary activations on admissible inputs in the sense of Definition B.20, then $f$ has ternary hidden activations on admissible inputs as well. Hence, if the output of $f$ is ternary on admissible inputs, $f$ has ternary activations too.*

*Proof.* Writing $f_i(x) = W_2^{(i)} \mathrm{ReLU}(W_1^{(i)} x + b^{(i)})$ for $i = 1, 2$, define

$$W_1 := \begin{pmatrix} W_1^{(1)} \\ W_1^{(2)} \end{pmatrix}, \quad b := \begin{pmatrix} b^{(1)} \\ b^{(2)} \end{pmatrix}, \quad W_2 := \begin{pmatrix} W_2^{(1)} & W_2^{(2)} \end{pmatrix}.$$

Then $f(x) := W_2 \mathrm{ReLU}(W_1 x + b) = f_1(x) + f_2(x)$ for all $x$. Concatenation preserves ternary weights, and on admissible inputs the hidden activations of $f$ are just the concatenation of those of $f_1$ and $f_2$, hence ternary as well. □

Next, we want to introduce some basic MLP operations used in the hardmax transformer constructions in this section. First, we want to construct an MLP which zeros a register $I$ if some flags have certain values. Since MLP outputs are added to the residual stream, to zero a register $I$ we need to construct an MLP that outputs $-x[I]$ on these coordinates and 0 elsewhere, so that $x[I]$ is zeroed out and other coordinates remain unchanged when added to the residual stream.

**Lemma C.3** (Zeroing). *Let $d \in \mathbb{N}$ and consider a register $I$, flags $F_1, \ldots, F_m$ and values $b_1^*, \ldots, b_m^* \in \{0, 1\}$ where $I, F_1, \ldots, F_m$ are pairwise disjoint (i.e. no flag $F_i$ appears as an index in the register $I$ or equals another flag). Then there exists a ternary MLP $f \in \mathcal{F}(d, d_{ff})$ with $d_{ff} = 2|I|$ such that for all $x \in \{-1, 0, 1\}^d$ with $x[F_j] \in \{0, 1\}$ for $j = 1, \ldots, m$,*

$$f(x)[I] = \begin{cases} -x[I], & \text{if } x[F_j] = b_j^* \text{ for } j \in \{1, \ldots, m\} \\ 0, & \text{otherwise.} \end{cases}$$

*and other output components (indices not in $I$) being zero.*

*Proof.* For each $i \in I$ we add two hidden neurons, one activates if all flags are set correctly (i.e. $x[F_j] = b_j^*$ for $j \in \{1, \ldots, m\}$) and $x[i] = 1$ and outputs $-e_i$, i.e. it outputs $-1$ at coordinate $i$ and 0 elsewhere. The other neuron activates if all flags are set correctly and $x[i] = -1$ and outputs $e_i$. By Lemma C.1 these $2|I|$ neurons can be implemented as single-neuron MLPs and combined with Lemma C.2 into one MLP $f$ with $2|I|$ hidden neurons. □

Next we need to be able to copy one representation block into another one, again conditional on certain flags. This works very similarly to the zeroing operation. Note that with residual connections, the first register is added onto the second one, hence it is only a copying operation if the second register starts out empty.

**Lemma C.4** (Copying). *Let $d \in \mathbb{N}$ with registers $I_1, I_2$ with $|I_1| = |I_2| = d'$ and flags $F_1, \ldots, F_m$ and values $b_1^*, \ldots, b_m^* \in \{0, 1\}$ where $I_1, I_2, F_1, \ldots, F_m$ are pairwise disjoint. Then there exists a ternary MLP $f \in \mathcal{F}(d, d_{ff})$ where $d_{ff} = 2d'$ such that for all $x \in \{-1, 0, 1\}^d$ with $x[F_j] \in \{0, 1\}$ for $j = 1, \ldots, m$,*

$$f(x)[I_2] = \begin{cases} x[I_1], & \text{if } x[F_j] = b_j^* \text{ for } j \in \{1, \ldots, m\} \\ 0, & \text{otherwise} \end{cases}$$

*and other output components (indices not in $I_2$) being zero.*

*Proof.* For each $j \in \{0, \ldots, d' - 1\}$ we add a neuron outputting $e_{I_2[j]}$ if all flags are set correctly and $x[I_1[j]] = 1$, and one neuron outputting $-e_{I_2[j]}$ if all flags are set correctly and $x[I_1[j]] = -1$. By Lemma C.1 these $2d'$ neurons can be implemented as single-neuron MLPs and combined with Lemma C.2 into one MLP. □

Next, we need an MLP to subtract $2^k$ from a number stored in $\pm 1$-binary form and saturate at $0$. This is used to compute position offsets so that an attention head can attend $2^k$ positions back.

**Lemma C.5** (Power of 2 Subtraction). *Let $d \in \mathbb{N}$, registers $I_1, I_2$ with $|I_1| = |I_2| = d' \in \mathbb{N}$, flags $F_1, \ldots, F_m$ and target bits $b_1^*, \ldots, b_m^* \in \{0, 1\}$ where $I_1, I_2, F_1, \ldots, F_m$ are pairwise disjoint, and let $k \in \{0, \ldots, d' - 1\}$. Then there exists a ternary MLP $f \in \mathcal{F}(d, d_{ff})$ where $d_{ff} = 4d'$ such that for all $x \in \{-1, 0, 1\}^d$ with $x[F_j] \in \{0, 1\}$ it holds that*

$$f(x)[I_2] = \begin{cases} \mathrm{bin}_{d'}\big(\max(0, p - 2^k)\big), & \text{if } x[I_1] = \mathrm{bin}_{d'}(p) \text{ and } x[F_j] = b_j^* \text{ for all } j \in \{1, \ldots, m\} \\ 0, & \text{otherwise,} \end{cases}$$

*and other output components (indices not in $I_2$) being zero.*

*Proof.* Recalling Definition B.1, for inputs with $x[I_1] = \mathrm{bin}_{d'}(p)$ the coordinate $x[I_1[m]] \in \{-1, 1\}$ is the $2^m$-bit of $p$ in $\{\pm 1\}$ encoding.

We now describe how to build $f$, using Lemma C.2 and Lemma C.1 repeatedly.

**Step 1: Copy $I_1$ to $I_2$.** By Lemma C.4 we can add $2d'$ neurons to the MLP which copy $x[I_1]$ to the output dimensions $I_2$ conditioned on $x[F_j] = b_j^*$ for all $j$. In particular, for $x[I_1] = \mathrm{bin}_{d'}(p)$ and correctly set flags, this writes $\mathrm{bin}_{d'}(p)$ to $f(x)[I_2]$, while incorrect flags keep the output zero.

**Step 2: Saturating case $p < 2^k$.** The condition $p < 2^k$ is equivalent to

$$x[I_1[t]] = -1 \quad \text{for all } t \in \{k, \ldots, d' - 1\}.$$

In this case we want the output on $I_2$ to be $\mathrm{bin}_{d'}(0)$, i.e. all $I_2$-coordinates equal to $-1$. For $m \geq k$ this is already true. For each $m \in \{0, \ldots, k - 1\}$ we add two identical neurons which write $-1$ to output coordinate $I_2[m]$ if $x[F_j] = b_j^*$ for all $j$, $x[I_1[m]] = 1$ and $x[I_1[t]] = -1$ for all $t \in \{k, \ldots, d' - 1\}$. Together they flip the output at coordinate $I_2[m]$ from $1$ to $-1$ if it was $1$ before. This adds $2k$ neurons to the MLP.

**Step 3: Borrow case $p \geq 2^k$.** Next, if $p \geq 2^k$, we do standard subtraction. In particular, we need to flip the lowest bit $m \geq k$ which is $1$ to $-1$ and all bits in $\{k, \ldots, m - 1\}$ (which are hence $-1$) to $1$. Thus for each $m \in \{k, \ldots, d' - 1\}$ we add two identical neurons which activate precisely if $x[F_j] = b_j^*$ for all $j$, $x[I_1[m]] = 1$ and $x[I_1[s]] = -1$ for $s \in \{k, \ldots, m - 1\}$ and add $-1$ to position $I_2[m]$ of the output and $1$ to positions $I_2[s]$ for $s \in \{k, \ldots, m - 1\}$. This adds $2(d' - k)$ neurons to the MLP.

Together these $4d'$ neurons implement the desired behavior. All building blocks are ternary MLPs (copying and single-neuron gadgets) conditioned on the flags, and repeated use of Lemma C.2 preserves ternary weights and activations. □

We will also need an in-place variant (i.e. where the output is added to the same register we subtract from).

**Lemma C.6** (In-place Power of 2 Subtraction). *Let $d \in \mathbb{N}$, let $I$ be a register with $|I| = d' \in \mathbb{N}$, let flags $F_1, \ldots, F_m$ and target bits $b_1^*, \ldots, b_m^* \in \{0, 1\}$ where $I, F_1, \ldots, F_m$ are pairwise disjoint, and let $k \in \{0, \ldots, d' - 1\}$. Then there exists a ternary MLP $f \in \mathcal{F}(d, d_{ff})$ with $d_{ff} = 2d'$ such that for all $x \in \{-1, 0, 1\}^d$ with $x[I] = \mathrm{bin}_{d'}(p)$ and $x[F_j] \in \{0, 1\}$ it holds that*

$$(\mathrm{id} + f)(x)[I] = \begin{cases} \mathrm{bin}_{d'}\big(\max(0, p - 2^k)\big), & \text{if } x[F_j] = b_j^* \text{ for all } j \in \{1, \ldots, m\}, \\ x[I], & \text{otherwise,} \end{cases}$$

*while other coordinates are always unmodified.*

*Proof.* This is the same bit-flipping construction as in the proof of Lemma C.5, but omitting the initial copying step. Each bit flip is implemented by two identical neurons so that, when added to the residual stream, it changes a coordinate in $\{-1, 1\}$ to the desired value. This uses $2d'$ neurons in total. □

We will also need to add $2^k$ to a number stored in $\pm 1$-binary form. We use the overflow behavior of saturating at the maximum value $2^{d'} - 1$, but this does not occur in our constructions and is only done for analogy to the previous lemma. The proof is then fully analogous to that of the previous lemma and hence omitted.

**Lemma C.7** (Power of 2 Addition). *Let $d \in \mathbb{N}$, registers $I_1, I_2$ with $|I_1| = |I_2| = d' \in \mathbb{N}$, flags $F_1, \ldots, F_m$ and target bits $b_1^*, \ldots, b_m^* \in \{0, 1\}$ where $I_1, I_2, F_1, \ldots, F_m$ are pairwise disjoint, and let $k \in \{0, \ldots, d' - 1\}$. Then there exists a ternary MLP $f \in \mathcal{F}(d, d_{ff})$ where $d_{ff} = 4d'$ such that for all $x \in \{-1, 0, 1\}^d$ with $x[F_j] \in \{0, 1\}$ it holds that*

$$f(x)[I_2] = \begin{cases} \text{bin}_{d'}\left(\min(2^{d'} - 1, p + 2^k)\right), & \text{if } x[I_1] = \text{bin}_{d'}(p) \text{ and } x[F_j] = b_j^* \text{ for all } j \in \{1, \ldots, m\} \\ 0, & \text{otherwise,} \end{cases}$$

*and other output components (indices not in $I_2$) being zero.*

## C.2. Proof of Proposition 3.1

We restate Proposition 3.1 in a more explicit way, briefly say how the statement of Proposition 3.1 follows from this restatement, and then provide a constructive proof using the techniques introduced in Appendix C.1. We use $r$ bits to encode absolute positions in the token sequence, in particular for the positional embedding. Hence, the transformer can handle at most $2^r$ tokens and as we prepend a `<bos>` token, handles input words $w \in \Sigma^*$ of length at most $2^r - 1$.

**Theorem C.8.** *Let $M = (Q, \Sigma, \delta, q_{init}, F)$ be a DFA and let $r \in \mathbb{N}$. Set $d_Q = \lceil \log_2 |Q| \rceil, d_\Sigma = \lceil \log_2 |\Sigma| \rceil$. Then there exists a hardmax transformer*

$$T_M^r \in \mathcal{T}^{\text{hard}}(d, d_{ff}, d_k, d_v, H, L, \mathcal{V})$$

*with vocabulary $\mathcal{V} = \Sigma \cup \{$`<bos>`$, \text{True}, \text{False}\}$, depth $L = r + 2$, $H = 1$ head per layer using dimensions $d_k = r, d_v = |Q|d_Q, d = 2r + 2|Q|d_Q + d_\Sigma + 1$ and $d_{ff} = 4|Q|d_Q + 2|Q|^2 d_Q + 6r$ and ternary weights such that for all $w = (w_0, \ldots, w_{n-1}) \in \Sigma^*$ with $n \leq 2^r - 1$,*

$$T(\text{<bos>}, w_0, \ldots, w_{n-1}) = \begin{cases} \text{True}, & w \in L(M), \\ \text{False}, & w \notin L(M). \end{cases}$$

*Moreover, $T$ has ternary activations on all such inputs.*

*Remark* C.9. To see how Proposition 3.1 follows from the above theorem statement, for a length bound $\hat{n} \in \mathbb{N}$ we simply set $r = \lceil \log_2(\hat{n} + 1) \rceil$. Then we have $r = \mathcal{O}(\log \hat{n})$ and hence also obtain the asymptotic scalings for depth and the dimensions claimed in Proposition 3.1.

The general idea for the proof of Theorem C.8 is as follows. For a given input $(\text{<bos>}, w_0, \ldots, w_{n-1})$, the goal is to output at the last token (position $n$, which is $w_{n-1}$ if $n \geq 1$ and `<bos>` if $n = 0$) whether or not the final state after reading $w$ is in $F$ or not. This final state can be written as

$$q_{\text{fin}} = \begin{cases} q_{\text{init}}, & n = 0, \\ (\delta(\cdot, w_{n-1}) \circ \delta(\cdot, w_{n-2}) \circ \cdots \circ \delta(\cdot, w_0))(q_{\text{init}}), & n \geq 1. \end{cases}$$

We encode functions $f : Q \to Q$ in a register which is initialized to $\delta(\cdot, w_i)$ at token $w_i$. In the first step, each token extracts the encoding $\delta(\cdot, w_{i-1})$ from the previous token and computes the encoding of $\delta(\cdot, w_i) \circ \delta(\cdot, w_{i-1})$. In the next step, the encoding stored 2 tokens back is extracted and the encoding of $\delta(\cdot, w_i) \circ \cdots \circ \delta(\cdot, w_{i-3})$ is computed. This continues, in each step doubling the number of recent tokens included in the embedding, until after $r$ steps the full encoding $\delta(\cdot, w_i) \circ \cdots \circ \delta(\cdot, w_0)$ is stored at position $i$ and we can easily compute $q_{\text{fin}}$ at the last token (position $n$).

To encode functions $f : Q \to Q$ we choose an enumeration $Q = \{q_0, \ldots, q_{|Q|-1}\}$ and an encoding $\text{enc}_Q : Q \to \{-1, 1\}^{d_Q}$ where $d_Q = \lceil \log_2 |Q| \rceil$ as in the theorem statement. Note that as we already enumerate $Q$, we could use the corresponding encoding $\text{enc}_Q(q_i) = \text{bin}_{d_Q}(i)$, but any other encoding will do as well. Now for each function $f : Q \to Q$ define

$$\text{enc}_{Q \to Q}(f) := (\text{enc}_Q(f(q_0)), \ldots, \text{enc}_Q(f(q_{|Q|-1}))),$$

which is a $d_Q|Q|$ dimensional encoding. As we need to compose these encodings, i.e. compute the encoding of $f_1 \circ f_2$ from the encodings of $f_1$ and $f_2$, we state one lemma for achieving this with an MLP before we prove Theorem C.8.

**Lemma C.10.** *With the notation above, consider some $d \geq 2d_Q|Q|$ and disjoint registers $I_1, I_2$. Then there exists an MLP $f \in \mathcal{F}(d, d_{ff})$ where $d_{ff} = 2d_Q|Q|^2$ such that for an input $x \in \mathbb{R}^d$ with $x[I_1] = \text{enc}_{Q \to Q}(f_1)$ and $x[I_2] = \text{enc}_{Q \to Q}(f_2)$ for some functions $f_1, f_2 : Q \to Q$, it holds that*

$$f(x)[I_1] = \text{enc}_{Q \to Q}(f_1 \circ f_2) .$$

*Proof.* We need to output the encodings $\text{enc}_Q(f_1(f_2(q_i)))$ for $i = 0, 1, \ldots, |Q| - 1$. Note that

$$\text{enc}_Q(f_1(f_2(q_i))) = \sum_{j=0}^{|Q|-1} \mathbf{1}_{f_2(q_i)=q_j} \text{ enc}_Q(f_1(q_j)) .$$

For $i, j \in \{0, \ldots, |Q| - 1\}$ and $k \in \{0, \ldots, d_Q - 1\}$ we add a hidden neuron which activates precisely if $x[I_2[d_Q i : d_Q(i + 1)]] = \text{enc}_Q(q_j)$ and $x[I_1[d_Q j + k]] = 1$ and outputs $e_{I_1[d_Q i+k]}$. Analogously we add a neuron outputting $-e_{I_1[d_Q i+k]}$ if $x[I_2[d_Q i : d_Q(i + 1)]] = \text{enc}_Q(q_j)$ and $x[I_1[d_Q j + k]] = -1$. For fixed $i, j$, these $2d_Q$ neurons write $\text{enc}_Q(f_1(q_j))$ to the slice $I_1[d_Q i : d_Q(i + 1)]$ whenever $f_2(q_i) = q_j$, and for each $i$ exactly one $j$ contributes. Hence together these $2d_Q|Q|^2$ neurons produce the desired output. □

*Proof of Theorem C.8.* The representation dimensions are split into $I_{\text{pos}}$ ($r$ dimensions, encodes position), $I_{\text{enc}}$ and $I_{\text{enc,ex}}$ (each $|Q|d_Q$ dimensions, roles will become clear in the proof), $I_{\text{sym}}$ ($\lceil \log_2 |\Sigma| \rceil$ dimensions, used to initially encode the input symbol at each position), $I_{\text{bos}}$ (1 dimension, used to mark the `<bos>` symbol), $I_{\text{pos,ex}}$ ($r$ dimensions, used to store the decremented position for attending backwards).

We write $x_0, x_1, \ldots, x_n \in \mathcal{V}$ for the input tokens, i.e. for an input word $w = (w_0, \ldots, w_{n-1})$ we have $x_0 = $ `<bos>` and $x_i = w_{i-1}$ for $i \in \{1, \ldots, n\}$. We write $x_i^{(\ell)}$ for the hidden representation at position $i$ after layer $\ell$ and $x_i^{(\ell-0.5)}$ for the representation after the attention but before the MLP in layer $\ell$, following the notation introduced in Section 2.

**Embeddings.** We define the token embedding as

$$\text{emb}(v)[I_{\text{sym}}] = \begin{cases} \text{enc}_\Sigma(v), & v \in \Sigma \\ 0, & \text{otherwise} \end{cases} \quad \text{and} \quad \text{emb}(v)[I_{\text{bos}}] = \begin{cases} 1, & v = \text{`<bos>'} \\ 0, & \text{otherwise} \end{cases}$$

and the positional embedding as

$$\text{posemb}(i)[I_{\text{pos}}] = \text{bin}_r(i)$$

with other registers initialized to zero.

**Initializing $I_{\text{enc}}$.** The attention in layer 1 is not used, while the MLP in layer 1 is used to write $\text{enc}_{Q \to Q}(\delta(\cdot, x_i))$ into $x_i^{(1)}[I_{\text{enc}}]$. This is done by having one neuron for each $\sigma \in \Sigma$ which activates to 1 precisely if $x_i^{(0.5)}[I_{\text{sym}}] = \text{enc}_\Sigma(\sigma)$ and writes $\text{enc}_{Q \to Q}(\delta(\cdot, x_i))$ to coordinates $I_{\text{enc}}$ (i.e. this encoding is hardcoded into the output weights of this neuron). For the `<bos>` symbol, we write $\text{enc}_{Q \to Q}(\text{id}_Q)$ into $I_{\text{enc}}$ instead.

**Binary Tree Accumulation.** For $k = 0, 1, \ldots, r - 1$ we do the following. The MLP in layer $k + 1$ zeros the register $I_{\text{pos,ex}}$ and writes $\text{bin}_r((i - 2^k)^+)$ into it. This can be done in parallel (using Lemma C.2) and uses Lemma C.5 for the subtraction. Hence, it holds that $x_i^{(k+1)}[I_{\text{pos,ex}}] = \text{bin}_r((i - 2^k)^+)$. Next, in layer $k + 2$ we add an attention head attending with queries $I_{\text{pos,ex}}$, keys $I_{\text{pos}}$, values $I_{\text{enc}}$ and writing to $I_{\text{enc,ex}}$. By Lemma B.2, the dot product $\langle q_i, k_j \rangle$ is uniquely maximized at $j = (i - 2^k)^+$ (with a gap of at least 2 to all $j \neq (i - 2^k)^+$), hence position $i$ attends precisely to position $(i - 2^k)^+$. Let

$$x_i^{(k+1.5)}[I_{\text{enc}}] =: \text{enc}_{Q \to Q}(f_1) \quad \text{and} \quad x_i^{(k+1.5)}[I_{\text{enc,ex}}] =: \text{enc}_{Q \to Q}(f_2),$$

where the fact that these two registers each encode a function on $Q$ follows inductively. Then the MLP in layer $k + 2$ adds $\text{enc}_{Q \to Q}(f_1 \circ f_2)$ to $I_{\text{enc}}$, while at the same time zeroing its previous contents and zeroing $I_{\text{enc,ex}}$ as well, using Lemma C.10 for computing the composition, Lemma C.3 for zeroing and Lemma C.2 for doing them in parallel (and in parallel to the computation of $\text{bin}_r((i - 2^{k+1})^+)$ for the next layer if $k < r - 1$).

After layer 1, the register $I_{\text{enc}}$ contains $x_i^{(1)}[I_{\text{enc}}] = \text{enc}_{Q \to Q}(\delta(\cdot, x_i))$. The attention in layer 2 then extracts $\text{enc}_{Q \to Q}(\delta(\cdot, x_{i-1}))$ and the MLP composes it with $\text{enc}_{Q \to Q}(\delta(\cdot, x_i))$, hence obtaining

$$x_i^{(2)}[I_{\text{enc}}] = \text{enc}_{Q \to Q}(\delta(\cdot, x_i) \circ \delta(\cdot, x_{i-1}))$$

where the $\delta(\cdot, x_{i-1})$ is replaced with $\text{id}_Q$ for $i = 1$ as this token attends to <bos> instead. After one more layer, we obtain

$$x_i^{(3)}[I_{\text{enc}}] = \text{enc}_{Q \to Q}(\delta(\cdot, x_i) \circ \cdots \circ \delta(\cdot, x_{\max(1,i-3)})) \, .$$

This scheme continues until after layer $r + 1$ we have

$$x_i^{(r+1)}[I_{\text{enc}}] = \text{enc}_{Q \to Q}(\delta(\cdot, x_i) \circ \cdots \circ \delta(\cdot, x_{\max(1,i-(2^r-1))})) = \text{enc}_{Q \to Q}(\delta(\cdot, x_i) \circ \cdots \circ \delta(\cdot, x_1))$$

where we used $i \leq n < 2^r$. In particular, by the definition of $\text{enc}_{Q \to Q}$ and our choice $q_0 = q_{\text{init}}$, the first $d_Q$ coordinates of this register, i.e. the slice $I_{\text{enc}}[0 : d_Q]$, encode

$$(\delta(\cdot, x_n) \circ \cdots \circ \delta(\cdot, x_1))(q_{\text{init}}) =: q_{\text{fin}},$$

i.e. the state of $M$ after reading the whole input.

**Answer Computation and Unembedding.** Finally we use the MLP in layer $r + 2$ to map the encoding of the final state, $\text{enc}_Q(q_{\text{fin}})$, to 1 if $q_{\text{fin}} \in F$ and to $-1$ otherwise, which can simply be written into the first coordinate while at the same time zeroing its contents. At the last token (position $n$, which equals <bos> when $n = 0$) this yields

$$x_n^{(r+2)}[1] = \begin{cases} 1, & q_{\text{fin}} \in F \\ -1, & q_{\text{fin}} \notin F \end{cases} .$$

Lastly we set the unembedding vectors to $\text{unemb}_{\text{True}} = e_1, \text{unemb}_{\text{False}} = -e_1$ and zero for other tokens, hence correctly outputting True if $q_{\text{fin}} \in F$ and False otherwise.

**Dimensions.** Note that the MLP in layer $k + 2$ does several things at once:

- It zeros out $I_{\text{enc}}$ and $I_{\text{enc,ex}}$, using $2d_Q|Q|$ neurons for each part.

- It computes the composition $\text{enc}_{Q \to Q}(f_1 \circ f_2)$, using $2d_Q|Q|^2$ neurons.

- It zeros out $I_{\text{pos,ex}}$, requiring $2r$ neurons.

- It computes $\text{bin}_r((i - 2^{k+1})^+)$ for the next layer (unless $k = r - 1$), using $4r$ neurons.

hence these MLPs each use $4d_Q|Q| + 2d_Q|Q|^2 + 6r$ neurons, while the initial and final MLP use fewer. Queries and keys are always position encodings using $r$ bits, hence $d_k = r$ suffices. The extracted values are encodings of functions $Q \to Q$, hence we need $d_v = d_Q|Q|$. □

## C.3. Proof of Theorem 3.2

To prove Theorem 3.2, we first define the vocabulary the constructed transformer uses and the token sequence it will produce, restate the theorem in a more explicit way, see how Theorem 3.2 follows from that restatement and lastly prove the restated theorem. The construction will depend on a constant $r \in \mathbb{N}$, the number of bits used to encode both absolute positions in the token sequence (e.g. the positional embedding) and tape indices in the Turing machine. While one could slightly improve constants for Turing machines with small space usage by using fewer bits to encode tape positions, this would not change the asymptotic scaling and would complicate the considerations.

### C.3.1. VOCABULARY

The vocabulary for the transformer is defined as follows:

**Definition C.11.** Let $M = (K, Q, \Sigma, \Gamma, \delta, \sqcup, q_{\text{init}}, q_{\text{halt}})$ be a Turing machine. We define the vocabulary used by our CoT construction as

$$\mathcal{V} = \mathcal{V}_M^{\text{CoT}} = \mathcal{V}_{\text{delim}} \cup \mathcal{V}_{\text{run}} \cup \mathcal{V}_{\text{pos}} \cup \Sigma$$

with *delimiter tokens*

$$\mathcal{V}_{\text{delim}} = \{\texttt{<inp>}, \texttt{</inp>}, \texttt{<outp>}, \texttt{</outp>}, \texttt{<p>}, \texttt{</p>}\} \,,$$

*run tokens*

$$\mathcal{V}_{\text{run}} = Q \times \Gamma^K \times \{L, S, R\}^K \,,$$

and *position* tokens

$$\mathcal{V}_{\text{pos}} = \{-1, 1\}^K$$

It should be noted that in Figure 2, the head positions are written with 0/1 tokens instead of $-1$/1 used here for consistency with the actual encodings. Clearly this does not matter, as the tokens in a transformer's vocabulary can simply be renamed.

### C.3.2. TOKEN SEQUENCE

We now define the token sequence the transformer will produce when running with CoT on some input $w$ and bound its length.

**Definition C.12.** Consider a Turing machine $M$, a constant $r \in \mathbb{N}$ with $r \geq 2$ and an input $w = (w_0, \ldots, w_{n-1}) \in \Sigma^*$ such that $f_M(w) = u = (u_0, \ldots, u_{m-1}) \in \Sigma^*$. Define *run tokens* for $t \in \{1, \ldots, t_M(w)\}$ as

$$\text{run}_t = (q_t, (y_t^k)_{k=1}^K, (\Delta_t^k)_{k=1}^K) \,,$$

using the notation from Definition B.6, where $q_t \in Q$ encodes the state entered, $y_t^k \in \Gamma$ the symbol written at the old head position $n_{t-1}^k$ on tape $k$ and $\Delta_t^k \in \{L, S, R\}$ the head movement on tape $k$ in step $t$. Then we define $\text{toks}(M, w, r) \in \mathcal{V}^+$ with the vocabulary $\mathcal{V}$ from Definition C.11 as the concatenation of three parts:

- The encoded input, i.e. the $n + 2$ tokens

$$\texttt{<inp>}, w_0, \ldots, w_{n-1}, \texttt{</inp>} \,.$$

- Alternation between the next $r$ run tokens and the encoded head positions afterwards, encoded as

$$\texttt{<p>}, (\text{bin}_r^0(n_t^k))_{k=1}^K, \ldots, (\text{bin}_r^{r-1}(n_t^k))_{k=1}^K, \texttt{</p>}$$

where $t$ is the index of the last run token. This continues until a run token has a halting state, i.e. this part always ends with a run token.

- The encoded output, i.e. the $m + 2$ tokens

$$\texttt{<outp>}, u_0, \ldots, u_{m-1}, \texttt{</outp>} \,.$$

We now bound the length of this token sequence.

**Lemma C.13.** *Consider the setting from Definition C.12. Then*

$$|toks(M, w, r)| \leq 4 + 2|w| + 4t_M(w) \,.$$

*Proof.* Let $n := |w|$ and let the three parts be input, trace, output $\in \mathcal{V}^+$. For the trace, $r$ out of every $2r + 2$ tokens and some additional tokens at the end are run tokens, and there are $t_M(w)$ such run tokens in total. Hence,

$$|\text{trace}| \leq \frac{2r + 2}{r} t_M(w) \leq 3t_M(w)$$

using that $r \geq 2$. Furthermore, as the head on tape 1 starts at 0 and moves by at most 1 in each step, the output length is bounded by $n + t_M(w)$. Hence,

$$|\text{toks}(M, w, r)| \leq (n + 2) + 3t_M(w) + (n + t_M(w) + 2) = 4 + 2|w| + 4t_M(w) \,.$$

$\square$

C.3.3. MAIN CONSTRUCTION

Now we restate Theorem 3.2 in a more explicit way:

**Theorem C.14.** *Let* $M = (K, Q, \Sigma, \Gamma, \delta, \sqcup, q_{init}, q_{halt})$ *be a Turing machine and let* $r \in \mathbb{N}$ *be even. Set* $d_Q = \lceil \log_2 |Q| \rceil$ *and* $d_\Gamma = \lceil \log_2 |\Gamma| \rceil$. *Then there exists a transformer* $T = T_M^r \in \mathcal{T}^{\mathrm{hard}}(d, d_{ff}, d_k, d_v, H, L, \mathcal{V})$ *with depth* $L = \frac{5r}{2} + 8$, $H = 3K$ *heads per layer using dimensions* $d_k = 4r - 1$, $d_v = \max\{r, d_Q, d_\Gamma\}$,

$$d_{ff} = \max\{18r + 2, \ 14Kr + 2r, \ |Q||\Gamma|^K + 1\} \tag{1}$$

*and*

$$d = 6Kr + 6r + 3d_Q + (3K + 1)d_\Gamma + 10K + 21 \tag{2}$$

*and the vocabulary defined in Definition C.11 such that for all* $w \in \Sigma^*$ *the following holds. If* $M$ *halts on* $w$ *after* $t_M(w)$ *steps with output* $f_M(w) \in \Sigma^*$, *then running* $T$ *on input* $w$ *using CoT (see Definition 2.1) produces the token sequence* $\mathrm{toks}(M, w, r)$ *defined in Definition C.12 if this token sequence has length at most* $2^r$.

*Remark* C.15. To see how Theorem 3.2 follows from Theorem C.14, let $\hat{t} \in \mathbb{N}$. Set

$$r = 2\lceil \frac{1}{2} \log_2(4 + 6\hat{t}) \rceil .$$

Then for an input $w \in \Sigma^*$ with $|w|, t_M(w) \leq \hat{t}$ it holds that

$$|\mathrm{toks}(M, w, r)| \leq 4 + 2|w| + 4t_M(w) \leq 4 + 6\hat{t} \leq 2^r$$

and hence Theorem C.14 applies and bounds the length of the token sequence by

$$4 + 2|w| + 4t_M(w) = \mathcal{O}(t_M(w) + |w|) .$$

Moreover, the transformer $T$ has ternary weights and ternary activations (in the sense of Definition B.20) on the full set of sequences arising during CoT generation; see Lemma D.15 (Items 4 and 6).

In the construction we will use the following lemma to add head movements (encoded with 2 bits) onto a binary representation of a head position. If the head movement is $L$, the head position gets decremented while saturating at zero (corresponding to the Turing machine behavior; see Definition B.6), while the head movement $R$ increments the position (here overflow will never occur). Other cases leave the head position unchanged.

**Lemma C.16.** *Let* $d, r \in \mathbb{N}$, *let* $I_1, I_2$ *be* $r$-*dimensional registers, let* $I_3$ *a 2-dimensional register and let* $F$ *be a flag. Consider an encoding* $\mathrm{enc}_\Delta : \{L, S, R\} \to \{-1, 1\}^2$. *Then there exists an MLP* $f \in \mathcal{F}(d, d_{ff})$ *with width* $d_{ff} = 6r$ *such that for all* $x \in \{-1, 0, 1\}^d$ *where* $x[I_1] = \mathrm{bin}_r(s)$ *for some* $s \in \{0, 1, \dots, 2^r - 2\}$ *and* $x[F] = 1$, *it holds that*

$$f(x)[I_2] = \begin{cases} \mathrm{bin}_r(\max(0, s - 1)) & \text{if } x[I_3] = \mathrm{enc}_\Delta(L) \\ \mathrm{bin}_r(s + 1) & \text{if } x[I_3] = \mathrm{enc}_\Delta(R) \\ \mathrm{bin}_r(s) & \text{if } x[I_3] = \mathrm{enc}_\Delta(S) \end{cases}$$

*and* $f(x)[I_2] = 0$ *if* $x[F] = 0$, *while other coordinates are always zero. Note that the case where* $x[F] = 1$ *but* $x[I_1]$ *is different is undefined as it never occurs when using this lemma.*

*Proof.* This is very similar to Lemma C.5. In particular, the MLP uses

- $2r$ neurons to copy $I_1$ to $I_2$ conditional on $F$ being set to 1.

- For each $j \in \{0, \dots, r - 1\}$ two identical neurons which activate to 1 if $F$ is set to 1, $I_3$ is set to $\mathrm{enc}_\Delta(L)$, $I_1[j]$ is set to 1 and $I_1[j']$ is set to $-1$ for $j' < j$. These neurons then flip bits $j' < j$ of $I_2$ from $-1$ to 1 and flip bit $j$ from 1 to $-1$, implementing the decrement.

- For each $j \in \{0, \dots, r - 1\}$ two identical neurons which activate to 1 if $F$ is set to 1, $I_3$ is set to $\mathrm{enc}_\Delta(R)$, $I_1[j]$ is set to $-1$ and $I_1[j']$ is set to 1 for $j' < j$. These neurons then flip bits $j' < j$ of $I_2$ from 1 to $-1$ and flip bit $j$ from $-1$ to 1, implementing the increment.

$\square$

Furthermore we will need to subtract ($r$-bit binary representations of) numbers. While there are several ways to achieve this with MLPs trading off depth and width differently, we will use a simple sequential method making use of Lemma C.6:

**Lemma C.17.** *Let $d, r \in \mathbb{N}$, let $I_1, I_2$ be $r$-dimensional registers and let $F$ be a flag. Then there exist $r$ MLPs $f_0, f_1, \ldots, f_{r-1} \in \mathcal{F}(d, d_{ff})$ where $d_{ff} = 2r$ such that the following holds for*

$$\bar{f} = (\mathrm{id} + f_{r-1}) \circ \cdots \circ (\mathrm{id} + f_0) .$$

*For any $x \in \{-1, 0, 1\}^d$ with $x[I_1] = \mathrm{bin}_r(s_1), x[I_2] = \mathrm{bin}_r(s_2)$ where $0 \le s_1 \le s_2 < 2^r - 1$ and $x[F] = 1$,*

$$\bar{f}(x)[I_2] = \mathrm{bin}_r(s_2 - s_1)$$

*and $\bar{f}(x)[I_2] = x[I_2]$ if $x[F] = 0$, while other coordinates always stay unmodified. Other cases again do not occur.*

*Proof.* This is just a repeated application of the in-place subtraction gadget from Lemma C.6, conditioning on a $-1/1$-valued register bit instead of a $0/1$-valued flag. In particular, each $f_i$ (for $i \in \{0, \ldots, r-1\}$) consists of 2 identical neurons for each $j \in \{i, \ldots, r-1\}$ flipping bit $j$ of $I_2$ from 1 to $-1$ and bits $j' < j$ from $-1$ to 1 if $I_2[j]$ is set to 1, $I_2[j']$ is set to $-1$ for $i \le j' < j$, $F$ is set to 1 and $I_1[i]$ is set to 1. Hence, $f_i$ decrements $I_2$ by $2^i$ if $I_1[i]$ is set to 1 and $F$ is set to 1, which together subtracts the contents of $I_1$ from those of $I_2$. $\square$

*Proof of Theorem C.14.* As the construction is quite a bit more complex than the previous one, we will not explicitly list all the registers here and instead define new ones during the construction. Unless specified otherwise, these registers are initialized to zero. Furthermore all registers are taken to be disjoint without explicit mentioning. Several $r$-dimensional registers are used to encode tape positions inside the Turing machine. As head positions satisfy $n_t^k \le t$, the assumption in Theorem C.14 implies that

$$\begin{aligned} s_M(w) &= \max(\{|w|\} \cup \{1 + n_t^k \mid k \in \{1, \ldots, K\}, t \in \{1, \ldots, t_M(w)\}\}) \\ &\le \max(|w|, 1 + t_M(w)) \\ &< |\mathrm{toks}(M, w, r)| - 2 \\ &\le 2^r - 2 \end{aligned}$$

which ensures that tape positions where symbols are written or a head moves are always representable with the $r$ bits used and can be incremented without overflow.

**Embeddings.** We use $r$-dimensional binary positional embeddings and encode the tokens with flags for the token types and binary representations of states and symbols, as detailed below. The positional embeddings will again be $r$-bit binary representations of the position, i.e.

$$\mathrm{posemb}(i)[I_{\mathrm{pos}}] = \mathrm{bin}_r(i)$$

where $I_{\mathrm{pos}}$ is an $r$-dimensional register and the positional embedding is zero for other coordinates.

The token embedding will consist of several parts. First, we add flags $F_{\texttt{<inp>}}, F_{\texttt{</inp>}}, F_{\texttt{<outp>}}, F_{\texttt{</outp>}}, F_{\texttt{<p>}}, F_{\texttt{</p>}}$ marking the delimiter tokens, e.g.

$$\mathrm{emb}(v)[F_{\texttt{<inp>}}] = \begin{cases} 1, & v = \texttt{<inp>} \\ 0, & \text{else} \end{cases}$$

and analogously for the other flags. Furthermore, we add flags $F_{\mathrm{run}}, F_{\Sigma}, F_{\mathrm{pos}}$ which are set to 1 for tokens $x_i \in \mathcal{V}_{\mathrm{run}}, x_i \in \Sigma$ and $x_i \in \mathcal{V}_{\mathrm{pos}}$ respectively.

Next, we need the token embeddings to encode the actual information in the tokens. For position tokens this is done by using $K$ 1-dimensional registers $I_{\mathrm{posbit}}^k$ which are initialized to

$$\mathrm{emb}(v)[I_{\mathrm{posbit}}^k] = \begin{cases} v_k, & v \in \mathcal{V}_{\mathrm{pos}} \\ 0, & \text{else} \end{cases} .$$

As previously, we fix injective encodings $\text{enc}_Q : Q \to \{-1, 1\}^{\lceil \log_2 |Q| \rceil}$ and $\text{enc}_\Gamma : \Gamma \to \{-1, 1\}^{\lceil \log_2 |\Gamma| \rceil}$. For tokens encoding a state, which are precisely the run tokens, we initialize $I_{\text{state}}$ as the encoding of the state. Also, we initialize the state to $\text{enc}_Q(q_{\text{init}})$ for the `</inp>` token:

$$\text{emb}(v)[I_{\text{state}}] = \begin{cases} \text{enc}_Q(q), & v = (q, *, *) \in \mathcal{V}_{\text{run}} \\ \text{enc}_Q(q_{\text{init}}), & v = \texttt{</inp>} \\ 0, & \text{else} \end{cases}$$

For tokens encoding a symbol on tape $k$, i.e. run tokens and for $k = 1$ additionally input-output tokens (i.e. those in $\Sigma$), we add a $\lceil \log_2 \Gamma \rceil$-dimensional register $I_{\text{sym}}^k$ per tape encoding the symbol:

$$\text{emb}(v)[I_{\text{sym}}^k] = \begin{cases} \text{enc}_\Gamma(x_k), & v = (*, (x_j)_{j=1}^K, *) \in \mathcal{V}_{\text{run}} \\ \text{enc}_\Gamma(x), & v = x \in \Sigma \text{ and } k = 1 \\ 0, & \text{else} \end{cases}$$

Next, we encode the head movements for run tokens with a 2-dimensional register $I_{\text{move}}^k$ using some injective encoding $\text{enc}_\Delta : \{L, S, R\} \to \{-1, 1\}^2$, initializing it for each tape $k$ as

$$\text{emb}(v)[I_{\text{move}}^k] = \begin{cases} \text{enc}_\Delta(\Delta^k), & v = (*, *, (\Delta^j)_{j=1}^K) \in \mathcal{V}_{\text{run}} \\ 0, & \text{else} \end{cases}$$

Next, we add a 1-dimensional register $I_{\text{const}}$ which carries a constant 1, i.e.

$$\text{emb}(v)[I_{\text{const}}] = 1 , \ \forall v \in \mathcal{V}$$

All registers introduced later are implicitly assumed to be zero-initialized unless specified otherwise.

**Distinguishing Input and Output tokens.** Tokens between `<inp>` and `</inp>` (from now on referred to as *input tokens*) and those between `<outp>` and `</outp>` (referred to as *output tokens*) are not distinguished at embedding, as they both come from $\Sigma \subset \mathcal{V}$. As they need to be treated differently, we want to set flags $F_{\text{input}}, F_{\text{output}}$ marking them. This is done by adding an attention head to the first layer which attends with queries $I_{\text{const}}$, keys and values $F_{\texttt{<outp>}}$ and writing the output to a flag $F_{\exists\texttt{<outp>}}$. Hence, tokens after and including the `<outp>` token attend to `<outp>` and write a 1 into this flag, while others attend uniformly over previous tokens (which are not `<outp>`) and write 0 to the flag. In particular,

$$x_i^{(0.5)}[F_{\exists\texttt{<outp>}}] = \begin{cases} 1, & \exists_{j \leq i} : x_j = \texttt{<outp>} \\ 0, & \text{otherwise} \end{cases}$$

Next, we use the subsequent MLP to mark input tokens and output tokens by writing 1 to $F_{\text{output}}$ if $F_\Sigma$ and $F_{\exists\texttt{<outp>}}$ are both 1, and writing 1 to $F_{\text{input}}$ if $F_\Sigma$ is 1 and $F_{\exists\texttt{<outp>}}$ is 0. Both can be achieved with 1 neuron each using Lemma C.1.

**Head and Symbol Positions 1: Input Tokens, `</inp>`, `<outp>` and Output Tokens.** Next, the transformer needs to reconstruct the head positions for all tokens which need to later extract a symbol from some tape position (run tokens, output tokens and `</inp>`, `</p>` and `<outp>`) and the position of the symbol for tokens encoding a symbol which can be extracted (run tokens and for $k = 1$ input tokens, while output tokens are never attended to).

We introduce $r$-dimensional registers $I_{\text{searchpos}}^k$ and $I_{\text{spos}}^k$ for $k \in \{1, \ldots, K\}$. The goal of the Head and Symbol position acquisition is then stated as follows:

- For run tokens, `</inp>`, `<p>` and `</p>` write the head positions $n_t^k$ (in $\pm 1$-binary format) into $I_{\text{searchpos}}^k$ for all $k$, where head positions for `</inp>` are $n_0^k = 0$ and for `<p>` and `</p>` they are the values encoded in binary between them.

- For `<outp>` and output tokens, we write the position on tape 1 of the next output token we need to output. Note that at `<outp>` we need to output $f(w)_0$, at $f(w)_0$ we need to output $f(w)_1$ etc. In general, for `<outp>` and output tokens we write the distance to the `<outp>` token to $I^1_{\text{searchpos}}$ to extract the correct symbol later on.

In this part, we will handle input tokens, `</inp>`, `<outp>` and output tokens.

For `</inp>` and `<outp>`, we can simply set the embeddings to the correct values:

$$\text{emb}(v)[I^k_{\text{searchpos}}] = \begin{cases} \text{bin}_r(0), & v = \text{</inp>} \\ \text{bin}_r(0), & v = \text{<outp> and } k = 1 \\ 0, & \text{otherwise} \end{cases}.$$

An input token $w_i$ (which is written on position $i$ on tape 1) has absolute position $i + 1$ in our token sequence as $(x_0, x_1, x_2, \dots) = (\text{<inp>}, w_0, w_1, \dots)$. Hence, we use the MLP in layer 1 to write $\text{bin}_r(i - 1)$ to $I^1_{\text{spos}}$ for input tokens by applying Lemma C.5 with $k = 0$, $I_1 = I_{\text{pos}}$, $I_2 = I^1_{\text{spos}}$ obtaining

$$x_i^{(1)}[I^1_{\text{spos}}] = x_i^{(0.5)}[I^1_{\text{spos}}] + \begin{cases} \text{bin}_r(i - 1), & x_i^{(0.5)}[F_\Sigma] = 1 \text{ and } x_i^{(0.5)}[F_{\exists\text{<outp>}}] = 0 \\ 0, & \text{otherwise} \end{cases}.$$

Next, we need to handle output tokens. First, we use the MLP in layer 1 to copy $I_{\text{pos}}$ into a register $I_{\text{pos},\text{<outp>}}$ for the `<outp>` token. In layer 2, we broadcast $I_{\text{pos},\text{<outp>}}$ to all later tokens using an attention head with queries $(I_{\text{const}}, F_{\text{<outp>}}, F_{\text{<outp>}})$, keys $(F_{\text{<outp>}}, F_{\text{<inp>}}, F_{\text{<inp>}})$, values $I_{\text{pos},\text{<outp>}}$, writing to $I_{\text{pos},\text{<outp>}}$. At `<outp>` (i.e. at position $i_{\text{<outp>}}$) this attends to `<inp>` since $\langle q_{i_{\text{<outp>}}}, k_0 \rangle = 2$ while $\langle q_{i_{\text{<outp>}}}, k_{i_{\text{<outp>}}} \rangle = 1$, extracting 0, while later tokens $i > i_{\text{<outp>}}$ attend to `<outp>` since $\langle q_i, k_{i_{\text{<outp>}}} \rangle = 1$ but $\langle q_i, k_0 \rangle = 0$, extracting the position. The MLP in layer 2 copies $I_{\text{pos}}$ into $I^1_{\text{searchpos}}$ for output tokens (this cannot be done in layer 1, as we first need to set $F_{\text{output}}$) and the MLPs in layers 3 to $r + 2$ then subtract $I_{\text{pos},\text{<outp>}}$ from $I^1_{\text{searchpos}}$ for output tokens using Lemma C.17, obtaining

$$x_i^{(r+2)}[I^1_{\text{searchpos}}] = \text{bin}_r(i - i_{\text{<outp>}}), \quad i > i_{\text{<outp>}}$$

for output tokens.

**Head and Symbol Positions 2: `</p>`.** Next, the `</p>` token needs to acquire the head positions, which are encoded in binary format in the $r$ previous tokens. Hence it can simply use $r$ heads per tape (i.e. $Kr$ heads in total) to extract the head position bits from these $r$ tokens. These heads could in principle be split over layers in different ways, a relatively simple and efficient method uses $\frac{r}{2}$ layers and two registers for the decremented positions and works as follows.

First, add two $r$-dimensional registers $I_{\text{pos}_1}, I_{\text{pos}_2}$ and in the first layer's MLP set them to $i - 1$ and $i - 2$:

$$x_i^{(1)}[I_{\text{pos}_1}] = \text{bin}_r(i - 1) , \; x_i^{(1)}[I_{\text{pos}_2}] = \text{bin}_r(i - 2)$$

and add $2K$ 1-dimensional zero-initialized registers $I^k_{\text{bits,ex1}}, I^k_{\text{bits,ex2}}$ for $k \in \{1, \dots, K\}$

Then, for $j = 1, \dots, \frac{r}{2}$:

- For each $k \in \{1, \dots, K\}$, add an attention head in layer $j + 1$ attending with queries $I_{\text{pos}_1}$, keys $I_{\text{pos}}$, values $I^k_{\text{posbit}}$ and writing to $I^k_{\text{bits,ex1}}$.

- Do the same with $I_{\text{pos}_2}, I^k_{\text{bits,ex2}}$ instead of $I_{\text{pos}_1}, I^k_{\text{bits,ex1}}$.

- For the `</p>` token, use the subsequent MLP to copy the contents of $I^k_{\text{bits,ex1}}$ into the correct part of $I^k_{\text{searchpos}}$. In particular, in step $j$ we extracted the bits at position $r - 2j + 1$, hence copy $I^k_{\text{bits,ex1}}$ into $I^k_{\text{searchpos}}[r - 2j + 1]$. Similarly copy $I^k_{\text{bits,ex2}}$ into $I^k_{\text{searchpos}}[r - 2j]$. In the same MLP, zero out $I^k_{\text{bits,ex1}}$ and $I^k_{\text{bits,ex2}}$ for all $k$. Hence, we apply Lemma C.4 and Lemma C.3 to $2K$ dimensions in total.

- Decrement both $I_{\text{pos}_1}$ and $I_{\text{pos}_2}$ by 2 in-place using Lemma C.6 (with $k = 1$). Note that decrementing by 2 is easier than decrementing e.g. by 3, hence the choice of splitting this over $\frac{r}{2}$ layers and not fewer.

With the above operations, before the attention heads in layer $j + 1$, $I_{\text{pos}_1}$ and $I_{\text{pos}_2}$ contain the binary encodings of $i - 2j + 1$ and $i - 2j$ respectively. Hence, each token attends precisely $i - 2j + 1$ back with the first head and $i - 2j$ back with the second one, extracting the positional bits from there. In total, for even $r$, the $</\text{p}>$ token attends to the previous $r$ tokens and extracts the positional bits into the head position registers, hence correctly reconstructing the head positions after layer $\frac{r}{2} + 1 =: L_1$. Note that the layers 3 to $L_1$ used above overlap with the output-token offset subtraction in layers 3 to $r + 2$; these operations act on disjoint token types (output tokens vs. $</\text{p}>$), so their pointwise MLP computations can be merged in the shared layers using Lemma C.2 and don't interfere with each other.

**Head and Symbol Positions 3: Propagation through Run Tokens.** The next part of the construction will use $r + 1$ layers to propagate the head positions from the $</\text{p}>$ and $</\text{inp}>$ token through the up to $r$ run tokens to the $<\text{p}>$ tokens, adding the head movements encoded by each run token. Introduce a new $r$-dimensional register $I_{\text{pos}\_}$ initialized to zero and set to the decremented positional encoding $\text{bin}_r(i - 1)$ for each position $i$ e.g. in layer $\ell = 1$, using Lemma C.5 with $k = 0$

$$x_i^{(1)}[I_{\text{pos}\_}] = \text{bin}_r(i - 1) .$$

Furthermore, introduce $K$ $r$-dimensional registers $I_{\text{hpos}\_}^k$ used to extract the head positions from the previous token.

In particular, for $j \in \{1, \ldots, r + 1\}$:

- For each $k \in \{1, \ldots, K\}$ add an attention head in layer $L_1 + j$ attending with queries $I_{\text{pos}\_}$, keys $I_{\text{pos}}$, values $I_{\text{searchpos}}^k$ and writing to $I_{\text{hpos}\_}^k$.

- For run tokens $x_i = \text{run}_t = (*, *, (\Delta_t^k)_{k=1}^K)$ the extracted head positions are precisely $n_{t-1}^k$, i.e. the head positions before step $t$. The subsequent MLP is then used to add $\Delta_t^k$ (encoded in $I_{\text{move}}^k$) onto $n_{t-1}^k$ using Lemma C.16, writing the result (i.e. the head positions $n_t^k$ after step $t$) to $I_{\text{searchpos}}^k$, while at the same time zeroing the previous contents of $I_{\text{searchpos}}^k$.

- For $<\text{p}>$ tokens, $I_{\text{hpos}\_}^k$ already holds the correct head position. We copy it to $I_{\text{searchpos}}^k$ while zeroing the previous contents.

- The same MLP also zeros $I_{\text{hpos}\_}^k$ for $j \leq r$ (i.e. not in the last iteration).

After $r + 1$ such propagation steps (and hence after layer $L_1 + r + 1$), the head position has been propagated through the $r$ run tokens to the next $<\text{p}>$ token. Run tokens and $<\text{p}>$ tokens then hold the correct head positions in $I_{\text{searchpos}}^k$, while run tokens $\text{run}_t$ also hold the previous head positions $n_{t-1}^k$ in $I_{\text{hpos}\_}^k$ (as we didn't zero this register in the last iteration). Recall that for a run token $\text{run}_t = (*, (y_t^k)_{k=1}^K, *)$, the symbols $y_t^k$ are written at positions $n_{t-1}^k$ (i.e. $y_t^k = (x_t^k)_{n_{t-1}^k}$), hence $I_{\text{hpos}\_}^k$ contains the correct symbol positions which we copy into $I_{\text{spos}}^k$ in layer $L_1 + r + 2$.

**Symbol Extraction.** After layer

$$L_2 := L_1 + r + 2 ,$$

i.e. in the representations $x_i^{(L_2)}$, every token which encodes a symbol being written to position $n$ on tape $k$ holds $\text{bin}_r(n)$ in $I_{\text{spos}}^k$, and each token which needs to extract the latest symbol written on tape $k$ at position $n$ holds $\text{bin}_r(n)$ in $I_{\text{searchpos}}^k$. Now for each token $x_i$ with an $I_{\text{searchpos}}^k$ we need to extract the symbol encoded in $I_{\text{sym}}^k$ for the latest token $j$ whose symbol position is the same as the current token's position.

In particular, setting

$$J_i^k = \{j \leq i \mid x_i^{(L_2)}[I_{\text{searchpos}}^k] = x_j^{(L_2)}[I_{\text{spos}}^k] \in \{-1, 1\}^r\}$$

we want to extract $x_{\max J_i^k}^{(L_2)}[I_{\text{sym}}^k]$ into some new $d_\Gamma$-dimensional register $I_{\text{sym,ex}}^k$ for each $k$ and each $i$ where $J_i^k \neq \emptyset$, leaving $I_{\text{sym,ex}}^k$ at zero otherwise.

First, we set flags to mark tokens where $J_i^k \neq \emptyset$. In particular, for each $k$, an attention head in layer $L_2 + 1$ attends for token $x_i$ with queries

$$q_i = (x_i^{(L_2)}[I_{\text{searchpos}}^k], \underbrace{1, \ldots, 1}_{r-1}) ,$$

keys

$$k_j = x_j^{(L_2)}[I_{\text{spos}}^k, \underbrace{F_{\texttt{<inp>}}, \ldots, F_{\texttt{<inp>}}}_{r-1}] \, ,$$

values

$$v_j = F_{\text{not}\texttt{<inp>}}$$

where $F_{\text{not}\texttt{<inp>}}$ is set at embedding to 0 for $\texttt{<inp>}$ and 1 for all other tokens, and writes the output to a flag $F_{\text{exist}}^k$. If for a token $x_i$ and some $k$ the set $J_i^k$ is non-empty, then for every $j \in J_i^k$ we have $\langle q_i, k_j \rangle = r$, while for $\texttt{<inp>}$ (at position 0) we have $\langle q_i, k_0 \rangle = r - 1$. Hence these tokens attend to all $j \in J_i^k$ and extract a 1 (as $0 \notin J_i^k$). On the other hand, if the set $J_i^k$ is empty, then the dot product with $\texttt{<inp>}$ is still $r - 1$ and hence larger than the dot product with all other positions $j > 0$, where $\langle q_i, k_j \rangle \leq r - 2$ (one wrong bit reduces the dot product by 2; see Lemma B.2). Hence, these tokens extract 0. In total we get

$$x_i^{(L_2+0.5)}[F_{\text{exist}}^k] = \begin{cases} 1, & J_i^k \neq \emptyset, \\ 0, & \text{otherwise} \end{cases} .$$

The next step is to compute, for tokens were $F_{\text{exist}}^k$ is set, the absolute position of the closest matching token, i.e. $\max J_i^k$ (again for all $k$). We introduce $r$-dimensional registers $I_{\text{pos,sym}}^k$ and for tokens encoding a symbol on tape $k$ (run tokens and for $k = 1$ input tokens) copy $I_{\text{pos}}$ into $I_{\text{pos,sym}}^k$. This can be done in the second layer's MLP, hence doesn't increase depth. The goal is then to write $\text{bin}_r(\max J_i^k)$ into new registers $I_{\text{pos,max}}^k$ for each token $i$ and tape index $k$ where $J_i^k \neq \emptyset$, i.e. where $F_{\text{exist}}^k$ is 1. Then we can simply attend to this position and extract the symbol from there.

To find $\max J_i^k$, we will use a binary search over positions, in each of $r$ steps (where one step is implemented by one transformer layer) determining one more bit of

$$\text{bin}_r(\max J_i^k) = (\text{bin}_r^0(\max J_i^k), \ldots, \text{bin}_r^{r-1}(\max J_i^k)) =: (b_0, \ldots, b_{r-1})$$

starting with the most significant bit $b_{r-1}$. In the first transformer layer, we use an attention head (for each $k$) which checks if a matching token (i.e. a $j \in J_i^k$) exists at a position

$$(\ast, \ldots, \ast, 1) \, ,$$

i.e. with the most significant bit equal to 1. If so, we know that $b_{r-1} = 1$, if not but $J_i^k \neq \emptyset$ we know that $b_{r-1} = -1$. In the next iteration, we check if a matching token exists at a position

$$(\ast, \ldots, \ast, 1, b_{r-1})$$

and again set the next bit $b_{r-2}$ to 1 or $-1$ accordingly. This continues for $r$ layers, in each layer obtaining one more bit. Note that if we e.g. have $\text{bin}_r^{r-1}(i) = -1$, i.e. $i < 2^{r-1}$, the search for a bit at a position $(\ast, \ldots, \ast, 1)$ will always fail as the token at position $i$ can only attend to tokens before it, hence correctly setting $b_{r-1} = -1$.

The process is described formally as follows. First, we add an $r$-dimensional register $I_{\text{pos,max}}^k$ for each $k$.

Then, for $j = 0, 1, \ldots, r - 1$:

- Write $b := r - 1 - j$ for the bit index we set in this iteration.

- Use an attention head (for each $k$) in layer $L_2 + j + 1$ with queries and keys

$$q_i = (x_i^{(L_2+j)}[I_{\text{searchpos}}^k], \underbrace{1, \ldots, 1,}_{r+j} \qquad\qquad 1, x_i^{(L_2+j)}[I_{\text{pos,max}}^k[b+1:r]])$$

$$k_j = (x_j^{(L_2+j)}[I_{\text{spos}}^k], \qquad x_j^{(L_2+j)}\underbrace{[F_{\texttt{<inp>}}, \ldots, F_{\texttt{<inp>}}]}_{r+j}, x_j^{(L_2+j)}[I_{\text{pos,sym}}^k[b:r]])$$

and values $v_j = x_j^{(L_2+j)}[F_{\text{not}\texttt{<inp>}}]$ extracted to $F_{\exists\text{high}}^k$. Queries and keys consist of three parts, so the dot product $\langle q_i, k_j \rangle$ decomposes accordingly. The first part (head position for queries, symbol position for keys) contributes $r$ to the dot product for matching tokens and at most $r - 2$ for non-matching tokens, and 0 for attention to $\texttt{<inp>}$. The

second part contributes $r + j$ to the dot product with `<inp>`. The third part contributes $j + 1$ to the dot product for positions that match the bits determined in previous iterations and also have bit 1 at position $b$, and contributes at most $j - 1$ for other positions and 0 for `<inp>`. In total, the dot product with `<inp>` will be $r + j$, the dot product with a matching token with bits higher than $b$ as determined before and bit $b$ being 1 is $r + j + 1$, while dot products to other tokens are at most $r + j - 1$. Hence, if a token with bits higher than $b$ matching the previously determined values and bit $b$ being 1 exists, we extract a 1, otherwise a 0 (as we then attend to `<inp>`).

- Next, the MLP in the same layer $L_2 + j + 1$ is used to write a 1 to $I_{\text{pos,max}}^k[b]$ if $F_{\exists\text{high}}^k$ is 1 and a $-1$ if $F_{\exists\text{high}}^k$ is 0 but $F_{\text{exist}}^k$ is 1. Furthermore, it zeros the contents of $F_{\exists\text{high}}^k$ for the next iteration.

After layer $L_2 + r$ we have $\text{bin}_r(\max J_i^k)$ stored in $I_{\text{pos,max}}^k$ for all $k$ and $i$ where $J_i^k \neq \emptyset$. Hence, we can now simply use an attention head per $k$ in layer $L_2 + r + 1$ attending with queries $q_i = (x_i^{(L_2+r)}[I_{\text{pos,max}}^k], 1)$, keys $k_j = (x_j^{(L_2+r)}[I_{\text{pos,sym}}^k], x_j^{(L_2+r)}[F_{\texttt{<inp>}}])$ and values $x_j^{(L_2+r)}[I_{\text{sym}}^k]$. If $J_i^k$ is non-empty, then $I_{\text{pos,max}}^k$ has been set to $\text{bin}_r(\max J_i^k)$, hence the dot product to this position dominates and we extract the correct symbol. Otherwise, we attend to `<inp>` and extract nothing.

Lastly, if no token was found we need to write a blank symbol to $I_{\text{sym,ex}}^k$. In particular, we use a single MLP neuron for each $k$, e.g. in layer $L_2 + r + 1$, to write $\text{enc}_\Gamma(\sqcup)$ to $I_{\text{sym,ex}}^k$ if $F_{\text{exist}}^k, F_{\texttt{<p>}}, F_{\text{pos}}$ and $F_{\text{input}}$ are all 0, as for these token types we did not want to extract a symbol anyway.

This process is finished after layer

$$L_3 := L_2 + r + 1 .$$

**Positional Blocks.** In parallel to the symbol extraction described previously, the positional bits and `<p>` token need to be handled. After layer $L_2$, the `<p>` tokens have the correct head positions stored in $I_{\text{searchpos}}^k$ and they need to be written out in the next $r$ tokens. First, we copy the contents of $I_{\text{searchpos}}^k$ into a new register $I_{\text{hpos},\texttt{<p>}}^k$ for each $k$. Next, we decrement $I_{\text{pos}}$ by 1, writing the result into a new register $I_{\text{pos}'}$. We introduce $K$ more 1-dimensional registers $I_{\text{nextbit}}^k$. Then, for $p = 1, \ldots, r - 1$ we

- Use an attention head in layer $L_2 + p$ to attend with queries $I_{\text{pos}'}$, keys $I_{\text{pos}}$ and extract the $p$-th bit of $I_{\text{hpos},\texttt{<p>}}^k$, i.e. $I_{\text{hpos},\texttt{<p>}}^k[p]$, into $I_{\text{nextbit}}^k$ for all $k$.

- The subsequent MLP in layer $L_2 + p$ is used to decrement $I_{\text{pos}'}$ by 1 in-place using Lemma C.6 (with $k = 0$).

Note that the attention head will only write something to the token exactly $p$ positions after `<p>` token (which needs to output the bits at position $p$, where the least significant bit corresponds to position 0), hence copying the bits from $I_{\text{hpos},\texttt{<p>}}^k$ into $I_{\text{nextbit}}^k$ for the correct token.

Next, we attend $r$ positions back (using queries $I_{\text{pos}'}$ and keys $I_{\text{pos}}$), extract $F_{\texttt{<p>}}$ and write it to a new flag $F_{\to\texttt{</p>}}$. This marks the token precisely $r$ positions after a `<p>` token, which needs to output `</p>` instead of another position token

Next, for the `<p>` token we copy the contents of $I_{\text{searchpos}}^k[0]$ into $I_{\text{nextbit}}^k$ for each $k$ using Lemma C.4.

Lastly, we decrement $I_{\text{pos}'}$ one last time, but this time by 2 in-place using Lemma C.6 (with $k = 1$), so it now points $r + 2$ tokens back. We attend $r + 2$ tokens back and extract the state from there to a new register $I_{\text{state,ex}}$. For `</p>` tokens this is precisely the state of the last run token, which we hence copy (only for `</p>` tokens) into $I_{\text{state}}$.

**Transition Function.** Tokens which need to output a run token have the symbols at the current head position in $I_{\text{sym,ex}}^k$ and the state in $I_{\text{state}}$, as the state was set at initialization for run tokens and `</inp>` and was set in the previous paragraph for `</p>` tokens. We can now compute the new state, new symbols and new head movements by using an MLP to compute the transition function $\delta$ of the turing machine. In particular, for each state $q$ and symbols $(y_k)_{k=1}^K$, we add a single neuron to the MLP in layer $L_3 + 1$ which fires if $I_{\text{state}}$ is set to $\text{enc}_Q(q)$ and $I_{\text{sym,ex}}^k$ is set to $\text{enc}_\Gamma(y_k)$ for each $k$, and for

$$\delta(q, (y_k)_{k=1}^K) =: (q', (y_k')_{k=1}^K, (\Delta_k')_{k=1}^K)$$

writes $\text{enc}_Q(q')$ into a new register $I_{\text{state,new}}$, $\text{enc}_\Gamma(y_k')$ into a new register $I_{\text{sym,new}}^k$ for each $k$ and $\text{enc}_\Delta(\Delta_k')$ into a new register $I_{\text{move,new}}^k$ for each $k$.

**Output.**   Finally we need to decide for each token what to output.

First, we set a flag in $F_{\text{halt}}$ at embedding for run tokens $(q_{\text{halt}}, *, *)$. Furthermore, we attend $r - 1$ positions back in layer $L_2 + r - 1$ when $\text{bin}_r(i - (r - 1))$ is stored in $I_{\text{pos}'}$ anyway and extract $F_{\text{run}}$ into some new flag $F_{\text{last run}}$. For run tokens, this flag marks the $r$'th run token, which needs to output `<p>` instead of another run token. We then set another flag $F_{\to \texttt{<p>}}$ to 1 precisely if $F_{\text{last run}}$ and $F_{\text{run}}$ are 1 and $F_{\text{halt}}$ is 0 (for a run token with a halting state we output the `<outp>` token instead of another positional block). If either $F_{\to \texttt{<p>}}$ or $F_{\text{halt}}$ are set to 1, we zero out the contents of $I_{\text{state,new}}$, $I_{\text{sym,new}}^k$ and $I_{\text{move,new}}^k$ for all $k$. We set the unembeddings as

$$\text{unemb}(v)[F_{\text{halt}}] = \begin{cases} 1, & v = \texttt{<outp>}, \\ 0, & \text{otherwise} \end{cases},$$

$$\text{unemb}(v)[F_{\to \texttt{<p>}}] = \begin{cases} 1, & v = \texttt{<p>} \\ 0, & \text{otherwise} \end{cases},$$

$$\text{unemb}(v)[I_{\text{state,new}}] = \begin{cases} \text{enc}_Q(q), & v = (q, *, *) \in \mathcal{V}_{\text{run}}, \\ 0, & \text{otherwise} \end{cases}$$

and analogously for $I_{\text{sym,new}}^k$ and $I_{\text{move,new}}^k$ for all $k$. Hence, for run tokens with a halting state we output `<outp>`, for a run token which is the $r$'th run token in a sequence but does not have a halting state we output `<p>`, and for other run tokens and `</inp>` and `</p>` we output the correct next run token corresponding to the contents of $I_{\text{state,new}}, I_{\text{sym,new}}^k, I_{\text{move,new}}^k$.

Next, for positional tokens we output `</p>` if the flag $F_{\to \texttt{</p>}}$ is set:

$$(\text{unemb}(v))[F_{\to \texttt{</p>}}] = \begin{cases} 1, & v = \texttt{</p>}, \\ 0, & \text{otherwise} \end{cases}.$$

For positional tokens without this flag set and for `<p>` tokens we output the next position token by setting

$$(\text{unemb}(v))[I_{\text{nextbit}}^k] = \begin{cases} v_k, & v = (v_j)_{j=1}^K \in \mathcal{V}_{\text{pos}} \\ 0, & \text{otherwise} \end{cases}.$$

Lastly, we handle the generation of the output. We need to output `</outp>` precisely if we are at `<outp>` or an output token and the next symbol on tape 1 is a blank symbol (as we do not include blank symbols in the output). Hence, we set a flag $F_{\text{blank}}$ to 1 if the contents of $I_{\text{sym,ex}}^1$ is $\text{enc}_\Gamma(\sqcup)$ in layer $L_3 + 1$. Then, in layer $L_3 + 2$ we set a flag $F_{\to \texttt{</outp>}}$ to 1 precisely if $F_{\texttt{<outp>}}$ or $F_{\text{output}}$ is set and $F_{\text{blank}}$ is set to 1. We then set

$$\text{unemb}(v)[F_{\to \texttt{</outp>}}] = \begin{cases} 1, & v = \texttt{</outp>} \\ 0, & \text{otherwise} \end{cases}.$$

Next, we set a flag $F_{\to \Sigma}$ in layer $L_3 + 3$ to 1 if $F_{\exists \texttt{<outp>}}$ is set (which is true for `<outp>` and output tokens) and $F_{\to \texttt{</outp>}}$ is set to 0. Then, in layer $L_3 + 4$ we copy $I_{\text{sym,ex}}^1$ to a new register $I_{\text{newsym}\Sigma}$ if $F_{\to \Sigma}$ is set to 1 and set the unembeddings to

$$\text{unemb}(v)[I_{\text{newsym}\Sigma}] = \begin{cases} \text{enc}_\Gamma(v), & v \in \Sigma \\ 0, & \text{otherwise} \end{cases}$$

hence outputting the correct output symbol for `<outp>` and for output tokens that don't need to output `</outp>`.

**Dimensions.**   The residual dimension $d$ is the sum of the sizes of all registers and flags introduced in the construction, hence $d = 6Kr + 6r + 3d_Q + (3K + 1)d_\Gamma + 10K + 21$. The largest key-query dimension occurs in the binary search for $\max J_i^k$ and is $d_k = 4r - 1$, while values are at most $r$-bit positions or $d_Q/d_\Gamma$-dimensional encodings, hence $d_v = \max\{r, d_Q, d_\Gamma\}$. The maximum number of heads used in a single layer is $3K$ (in layer $L_2 + 1$, which has $K$ heads for $F_{\text{exist}}^k$, $K$ for binary search and $K$ for positional bit extraction), and the depth is $L = L_3 + 4 = \frac{5r}{2} + 8$. Finally, the required MLP width is $d_{\text{ff}} = \max\{18r + 2, 14Kr + 2r, |Q||\Gamma|^K + 1\}$, as the widest MLP occurs either during the head position propagation (together with the output subtraction), in the initial layer, or when computing the transition function. $\qquad \square$

## C.4. Proof of Theorem 3.3

To prove Theorem 3.3, we first define the vocabulary used by the constructed transformer and define the segments of the token sequence that the transformer will produce when running with SCoT. We bound the length of each segment and the total length of all segments. Then we restate the theorem in more formal terms, see how Theorem 3.3 follows from that restatement and finally prove the restated theorem by constructing a transformer generating the stated segments when being run with SCoT.

### C.4.1. VOCABULARY

The vocabulary for SCoT simply extends that for CoT by the summary delimiters and tokens used in the summaries to encode tape contents with head positions and the state:

**Definition C.18.** Let $M = (K, Q, \Sigma, \Gamma, \delta, \sqcup, q_{\text{init}}, q_{\text{halt}})$ be a Turing machine. We define the vocabulary used by our SCoT construction as

$$\mathcal{V} = \mathcal{V}_M^{\text{SCoT}} = \mathcal{V}_{\text{delim}} \cup \mathcal{V}_{\text{run}} \cup \mathcal{V}_{\text{pos}} \cup \Sigma \cup \mathcal{V}_{\text{tape}} \cup Q$$

with $\mathcal{V}_{\text{run}}, \mathcal{V}_{\text{pos}}$ defined as in Definition C.11 and

$$\mathcal{V}_{\text{delim}} = \{\texttt{<inp>}, \texttt{</inp>}, \texttt{<outp>}, \texttt{</outp>}, \texttt{<p>}, \texttt{</p>}, \texttt{<summ>}, \texttt{</summ>}\} \,,$$

and

$$\mathcal{V}_{\text{tape}} = (\Gamma \cup \{\hat{\gamma} \mid \gamma \in \Gamma\})^K \,.$$

### C.4.2. TOKEN SEQUENCE

In order to define the segments of the token sequence produced by the SCoT transformer construction for a Turing machine $M$ and input $w$, we first define how to encode a configuration $C_t$ of a Turing machine $M$ occurring at a time $t$ when running on input $w$ into a summary. The summary consists of the tape contents at time $t$ written out using the tokens $\mathcal{V}_{\text{tape}}$ and the state at time $t$. The number of tokens used to write the tape contents is precisely the number of cells accessed on any tape up to time $t$.

**Definition C.19.** Let $M$ be a $K$-tape Turing machine and let $w \in \Sigma^*$ be an input. Consider the sequence of configurations

$$C_t = (q_t, (x_t^k)_{k=1}^K, (n_t^k)_{k=1}^K), \qquad 0 \leq t \leq t_M(w)$$

of $M$ on input $w$ defined in Definition B.6 with state $q_t$, tape contents $x_t^k$ and head positions $n_t^k$. Define the space usage at time $t$ inductively as

$$s_0 := |w|, \qquad s_t = \max(s_{t-1}, 1 + n_t^1, \ldots, 1 + n_t^K) \text{ for } t \in \{1, \ldots, t_M(w)\} \,.$$

Note that $s_{t_M(w)} = s_M(w)$ with the space usage $s_M(w)$ defined in Definition B.6. We then define

$$\text{summary}(M, w, t) := (\texttt{<summ>}, ((\tilde{x}_t^k)_0)_{k=1}^K, ((\tilde{x}_t^k)_1)_{k=1}^K, \ldots, ((\tilde{x}_t^k)_{s_t-1})_{k=1}^K, q_t, \texttt{</summ>})$$

where

$$(\tilde{x}_t^k)_i = \begin{cases} (\hat{x}_t^k)_i, & \text{if } n_t^k = i \\ (x_t^k)_i, & \text{if } n_t^k \neq i \end{cases},$$

i.e. the head positions are indicated using the $\hat{\gamma}$ symbols from $\mathcal{V}_{\text{tape}}$. Note that the length of each such summary is bounded by

$$|\text{summary}(M, w, t)| = 3 + s_t \leq 3 + s_M(w) \,.$$

**Definition C.20.** Consider a Turing machine $M$, a constant $r \in \mathbb{N}$ and an input $w = (w_0, \ldots, w_{n-1}) \in \Sigma^*$ such that $f_M(w) = u = (u_0, \ldots, u_{m-1}) \in \Sigma^*$. Let $(C_t)_{t=0}^{t_M(w)}$ be the configuration sequence of $M$ on $w$; see Definition B.6, and let $\text{run}_t$ ($t \in \{1, \ldots, t_M(w)\}$) be the run token encoding step $t$ as in Definition C.12. We define the SCoT token segments $\text{seg}_0, \text{seg}_1, \cdots \in \mathcal{V}_M^{\text{SCoT}}$ as follows, again dropping the dependence on $M$, $w$ and $r$.

We then inductively define the times $t_0, t_1, \cdots \in \{0, \ldots, t_M(w)\}$, summaries $\text{summary}_0, \text{summary}_1 \in \mathcal{V}^+$ (using the vocabulary defined in Definition C.18), traces $\text{trace}_1, \text{trace}_2, \ldots$ and segments $\text{seg}_1, \text{seg}_2, \cdots \in \mathcal{V}^+$ as follows. For $i = 0$, set $t_0 = 0$ and $\text{summary}_0 = (\texttt{<inp>}, w_0, \ldots, w_{n-1}, \texttt{</inp>})$. Then for $i = 1, 2, \ldots$ we set:

- $\text{trace}_i$ alternates between the next $r$ run tokens, starting at $t_{i-1} + 1$, and the head positions after the last of this run token. This continues until either a run token has a halting state or the length of this sequence is at least $3(|\text{summary}_{i-1}| - 1)^5$ and the last token is a run token. In particular, if this length cap is hit while writing the head positions, the first run token after the head positions finishes $\text{trace}_i$.

- In both cases in the previous item, we set $t_i$ as the index of the run token finishing $\text{trace}_i$, i.e. the last token in $\text{trace}_i$ is $\text{run}_{t_i}$. If the first case occurs, $\text{summary}_i$ is the output $(\texttt{<outp>}, u_0, \ldots, u_{m-1}, \texttt{</outp>})$ and $i_{\max} = i$, i.e. this is the last segment. If the latter case occurs, $\text{summary}_i$ is $\text{summary}(M, w, t_i)$ from Definition C.19

For $i \geq 1$ we define $\text{seg}_i$ as the concatenation $\text{summary}_{i-1}, \text{trace}_i, \text{summary}_i$.

**Lemma C.21.** *Consider the setting from Definition C.20. Assume $r \geq 2$. Then each segment has length bounded as*

$$|\text{seg}_i| \leq 8(s_M(w) + 3)$$

*and the total length of all segments is bounded as*

$$\sum_{i=1}^{i_{\max}} |\text{seg}_i| \leq 8t_M(w) + 2|w| + 4$$

*Proof.* Each summary (including the encoded input and output) uses at most $3 + s_M(w)$ tokens. Hence,

$$|\text{seg}_i| \leq 2(s_M(w) + 3) + |\text{trace}_i| .$$

Now consider two cases. The first case is where the initial summary $\text{summary}_{i-1}$ has length at most $\frac{r}{3} - 1$. Then the length cap is hit in $\text{trace}_i$ before the first head position encoding, i.e. at a run token, hence $|\text{trace}_i| = 3(|\text{summary}_{i-1}| - 1)$. The second case is where $|\text{summary}_{i-1}| \geq \frac{r}{3}$. Then the trace has maximum length if the length cap is hit at a $\texttt{<p>}$ token, in this case $r + 2$ tokens (the full head position encoding) are added to $\text{trace}_i$ in addition to the length cap. Hence $|\text{trace}_i| \leq 3(|\text{summary}_{i-1}| - 1) + r + 2 \leq 6|\text{summary}_{i-1}|$. In both cases it holds that $|\text{trace}_i| \leq 6|\text{summary}_{i-1}| \leq 6(s_M(w) + 3)$ and thus

$$|\text{seg}_i| \leq 8(s_M(w) + 3) .$$

For the second statement, note that each Turing machine step increases the number of used cells ($s_t$ in Definition C.19) by at most 1. Hence, it holds that

$$|\text{summary}_i| \leq |\text{summary}_{i-1}| + t_i - t_{i-1}$$

as $t_i - t_{i-1}$ is the number of run tokens and thus Turing machine steps between the two configurations. Furthermore, as each trace alternates between $r$ run tokens and $r + 2$ tokens encoding the head positions, it holds that

$$|\text{trace}_i| \leq \frac{2r + 2}{r}(t_i - t_{i-1}) \leq 3(t_i - t_{i-1})$$

using that $r \geq 2$. Hence we can bound the length of each segment $i < i_{\max}$ as

$$
\begin{aligned}
|\text{seg}_i| &= |\text{summary}_{i-1}| + |\text{trace}_i| + |\text{summary}_i| \\
&\leq 2|\text{summary}_{i-1}| + |\text{trace}_i| + t_i - t_{i-1} \\
&\leq 2(\frac{1}{3}|\text{trace}_i| + 1) + |\text{trace}_i| + t_i - t_{i-1} \\
&\leq \frac{5}{3}3(t_i - t_{i-1}) + 2 + t_i - t_{i-1} \\
&= 6(t_i - t_{i-1}) + 2
\end{aligned}
$$

where we used $|\text{trace}_i| \geq 3(|\text{summary}_{i-1}| - 1)$ in the third line. The last segment satisfies

$$
\begin{aligned}
|\text{seg}_{i_{\max}}| &\leq |\text{summary}_{i_{\max}-1}| + |\text{trace}_{i_{\max}}| + |\text{summary}_{i_{\max}}| \\
&\leq 2|\text{summary}_{i_{\max}-1}| + 3(t_{i_{\max}} - t_{i_{\max}-1})
\end{aligned}
$$

---

[5] The reason for this choice will become clear in the construction, although one could replace the 3 with any number $2^m - 1$ with minimal changes to the construction.

Hence

$$\sum_{i=1}^{i_{\max}} |\text{seg}_i| \le 6 \sum_{i=1}^{i_{\max}} (t_i - t_{i-1}) + 2|\text{summary}_{i_{\max}-1}|$$
$$\le 6t_M(w) + 2t_M(w) + 2|\text{summary}_0|$$
$$= 8t_M(w) + 4 + 2|w|$$

where in the second line we used

$$|\text{summary}_i| \le |\text{summary}_{i-1}| + (t_i - t_{i-1})$$
$$\le |\text{summary}_{i-2}| + (t_{i-1} - t_{i-2}) + (t_i - t_{i-1})$$
$$= |\text{summary}_{i-2}| + t_i - t_{i-2}$$
$$\le \dots$$
$$\le |\text{summary}_0| + t_i - t_0$$
$$\le |\text{summary}_0| + t_M(w)$$

$\square$

### C.4.3. MAIN CONSTRUCTION

Now we restate Theorem 3.3 in a more explicit way:

**Theorem C.22.** *Let $M = (K, Q, \Sigma, \Gamma, \delta, \sqcup, q_{init}, q_{halt})$ be a Turing machine and let $r \in \mathbb{N}$ be even with $r \ge 4$. Set $d_Q = \lceil \log_2 |Q| \rceil$ and $d_\Gamma = \lceil \log_2 |\Gamma| \rceil$. Then there exists a transformer $T = T_M^r \in \mathcal{T}^{\text{hard}}(d, d_{ff}, d_k, d_v, H, L, \mathcal{V})$ with depth $L = \frac{5r}{2} + 8$, $H = 3K + 2$ heads per layer using dimensions $d_k = 4r - 1$, $d_v = \max\{r, d_Q, d_\Gamma\}$,*

$$d_{ff} = \max\{22r + 11, 18Kr + 2r + 1, |Q||\Gamma|^K + 4Kd_\Gamma + 4K + 1\}$$

*and*

$$d = 7Kr + 9r + 5d_Q + (4K + 1)d_\Gamma + 13K + 31$$

*and the vocabulary defined in Definition C.18 such that for all $w \in \Sigma^*$ the following holds. If $M$ halts on $w$ after $t_M(w)$ steps with output $f_M(w) \in \Sigma^*$ using $s_M(w)$ space, then running $T$ on input $w$ using SCoT (see Definition 2.2) generates the segments defined in Definition C.20 if each of these segments has length bounded by $2^r$.*

*Remark* C.23. To see how Theorem 3.3 follows from Theorem C.22, consider a space bound $\hat{s} \in \mathbb{N}$. Set

$$r = 2\lceil \frac{1}{2} \log_2(8(\hat{s} + 3)) \rceil .$$

Then $r$ is even and for each input $w$ with $s_M(w) \le \hat{s}$, Lemma C.21 ensures that each segment has length bounded by $2^r$ and hence $T$ produces these segments when running with SCoT on $w$. Again using Lemma C.21, each segment's length (and hence the context size) is bounded by $8(s_M(w) + 3) = \mathcal{O}(s_M(w))$ and the total length of all segments is bounded by $8t_M(w) + 4 + 2|w| = \mathcal{O}(t_M(w) + |w|)$. Moreover, the transformer $T$ has ternary weights and ternary activations (in the sense of Definition B.20) on the full set of sequences arising during SCoT generation; see Lemma D.15 (Items 4 and 6).

*Proof of Theorem C.22.* The construction is obtained by modifying the transformer from Theorem C.14. We keep the entire CoT construction (in particular head position propagation, symbol extraction and transition computation) and only add the additional registers/flags needed to (i) continue from a summary, (ii) decide when to write a summary, and (iii) write the summary itself. Write $L_1, L_2, L_3$ for the layer indices from the proof of Theorem C.14. All additional operations below can be scheduled in parallel in the stated layers, hence do not increase the depth beyond the CoT construction.

**Embeddings.** We extend the embeddings from Theorem C.14 as follows. We add flags $F_{\texttt{<summ>}}, F_{\texttt{</summ>}}$ for $\texttt{<summ>}, \texttt{</summ>}$, and extend $F_Q$ and $F_{\text{tape}}$ as type flags for $Q$-tokens and tape tokens. For state tokens $q \in Q$ we embed $\text{enc}_Q(q)$ into $I_{\text{state}}$ (analogous to run tokens), and for tape tokens $(\tilde{\gamma}^k)_{k=1}^K \in \mathcal{V}_{\text{tape}}$ we embed $\text{enc}_\Gamma(\gamma^k)$ into $I_{\text{sym}}^k$, where $\tilde{\gamma}^k \in \{\gamma^k, \hat{\gamma}^k\}$ and we set flags $F_{\text{head}}^k$ indicating $\tilde{\gamma}^k = \hat{\gamma}^k$.

**Prompt Structure Flags (Layer 1).**    We need to distinguish whether the current segment starts from the input (ending in `</inp>`) or from a summary (ending in `</summ>`), and we need to know whether we are currently in the prompt summary or in the final generated summary. This is done analogously to $F_{\exists<\text{outp}>}$ in Theorem C.14:

- We add two heads in layer 1 with queries $I_{\text{const}}$ and keys/values $F_{</\text{inp}>}$ and $F_{</\text{summ}>}$, writing to flags $F_{\exists</\text{inp}>}$ and $F_{\exists</\text{summ}>}$. Hence these flags indicate whether `</inp>` or `</summ>` occurred in context.

- The MLP in layer 1 sets $F_{\text{final}<\text{summ}>}$ if $F_{<\text{summ}>} = 1$ and either $F_{\exists</\text{inp}>} = 1$ or $F_{\exists</\text{summ}>} = 1$. This marks precisely the `<summ>` token initiating the final summary of the segment.

- For segments starting with a summary there is no `<inp>` token, but the CoT construction uses `<inp>` as a base token in several places (e.g. in the definition of $F_{\text{exist}}^k$). Hence, the MLP in layer 1 sets $F_{<\text{inp}>}$ for the *first* `<summ>` token of the segment, i.e. when $F_{<\text{summ}>} = 1$ and $F_{\exists</\text{inp}>} = F_{\exists</\text{summ}>} = 0$. In the same case it writes $-1$ to $F_{\text{not}<\text{inp}>}$ so that $F_{\text{not}<\text{inp}>}$ becomes 0 on this token (at embedding it was 1).

- The MLP in layer 1 sets $F_{\text{tape,init}}$ for tape tokens with $F_{\text{tape}} = 1$ and $F_{\exists</\text{inp}>} = F_{\exists</\text{summ}>} = 0$ (tape tokens in the prompt summary), and $F_{\text{tape,fin}}$ for tape tokens with $F_{\text{tape}} = 1$ and ($F_{\exists</\text{inp}>} = 1$ or $F_{\exists</\text{summ}>} = 1$) (tape tokens in the final summary).

- Lastly, we store the absolute position of the prompt-ending token (either `</inp>` or `</summ>`) in a new register $I_{\text{pos,promptend}}$. Concretely, in layer 1 the MLP copies $I_{\text{pos}}$ to $I_{\text{pos,promptend}}$ once when $F_{</\text{inp}>} = 1$ and once when $F_{</\text{summ}>} = 1$. This is needed for the length cap condition.

**Continuing from a Summary.**    Tape tokens in the prompt summary already carry their symbol encodings $I_{\text{sym}}^k$, but to make them accessible for later symbol extraction we need their symbol-position registers. In layer 2 we therefore:

- Decrement $I_{\text{pos}}$ by 1 and write the result to $I_{\text{spos}}^k$ whenever $F_{\text{tape,init}}$ is set (for all $k$), i.e. the tape token at offset $p + 1$ after `<summ>` gets $\text{bin}_r(p)$. This is analogous to what is already being done for input tokens, but for all $k$ instead of just $k = 1$.

- Copy $I_{\text{pos}}$ to $I_{\text{pos,sym}}^k$ whenever $F_{\text{tape,init}}$ is set (for all $k$), so that the prompt tape tokens can serve as symbol sources in the binary search of Theorem C.14.

- Broadcast $I_{\text{pos,promptend}}$ to all later tokens analogously to $I_{\text{pos},<\text{outp}>}$ in Theorem C.14.

In layer 3, the unique `</summ>` token initializes the simulation state by extracting the head positions and state from the prompt summary: for each tape $k$, an attention head at layer 3 attends (only when $F_{</\text{summ}>} = 1$) to the unique tape token with $F_{\text{head}}^k = 1$ and extracts its $I_{\text{spos}}^k$ into $I_{\text{searchpos}}^k$, and a second head attends to the unique state token (with $F_Q = 1$) and extracts its $I_{\text{state}}$. From this point on, `</summ>` is treated exactly like `</inp>` in the CoT construction: head positions are propagated through run tokens using $I_{\text{move}}^k$, symbols are extracted as before, and run tokens are generated by the same transition MLP.

**Initiating the Final Summary.**    In addition to outputting `<outp>` when $F_{\text{halt}} = 1$, a run token should output `<summ>` once the length cap from Definition C.20 is reached. Let $j$ be the position of the prompt-ending token (either `</inp>` or `</summ>`). Then $|\text{trace}_i| \geq 3(|\text{summary}_{i-1}| - 1)$ is equivalent to being at an absolute position $i \geq 4j$. In our $r$-bit positional encoding this is detected via the identity $i = 4j \iff \text{bin}_r^s(i) = \text{bin}_r^{s-2}(j)$ for $s \geq 2$ and $\text{bin}_r^0(i) = \text{bin}_r^1(i) = -1$, together with the additional condition $\text{bin}_r^{r-2}(j) = \text{bin}_r^{r-1}(j) = -1$ (equivalently $j < 2^{r-2}$), ruling out wrap-around. Concretely:

- In layer 2, for each $s \in \{0, \ldots, r-3\}$ we set a flag $F_{\text{bit equal}}^s$ to 1 iff $I_{\text{pos}}[s+2]$ equals $I_{\text{pos,promptend}}[s]$.

- In layer 3 we set $F_{\text{length cap}}$ to 1 iff all $F_{\text{bit equal}}^s$ are 1, $I_{\text{pos}}[0] = I_{\text{pos}}[1] = -1$ and $I_{\text{pos,promptend}}[r-2] = I_{\text{pos,promptend}}[r-1] = -1$.

- In layer 4 an attention head propagates $F_{\text{length cap}}$ to all later tokens analogously to $I_{\text{pos},<\text{outp}>}$ in Theorem C.14.

- The MLP in layer 4 sets $F_{\to<\text{summ}>}$ precisely if $F_{\text{run}} = 1$, $F_{\text{length cap}} = 1$ and $F_{\text{halt}} = 0$.

- In the output logic of Theorem C.14 we add the case that $F_{\to\texttt{<summ>}}$ triggers output $\texttt{<summ>}$, and we suppress the other possible emissions: in the next layer we zero $F_{\to\texttt{<p>}}$ when $F_{\to\texttt{<summ>}} = 1$, and in the layer where $I_{\text{state,new}}, I^k_{\text{sym,new}}, I^k_{\text{move,new}}$ are already zeroed for $F_{\text{halt}}$ and $F_{\to\texttt{<p>}}$ we additionally zero them for $F_{\to\texttt{<summ>}}$ and for $F_Q$.

**Writing the Final Summary.**  Once the $\texttt{<summ>}$ token is emitted (marked by $F_{\text{final}\texttt{<summ>}}$), we generate the tape tokens and state token of $\text{summary}(M, w, t_i)$. The main idea is to treat the final summary like the output generation in Theorem C.14, except that we write tape cells instead of output symbols and additionally mark head positions.

- (Offset to the final-summary $\texttt{<summ>}$ token.) In layer 2 we copy $I_{\text{pos}}$ at the $F_{\text{final}\texttt{<summ>}}$ token to a register $I_{\text{pos},\texttt{<summ>}}$ and broadcast it in layer 3 analogously to $I_{\text{pos},\texttt{<outp>}}$ in Theorem C.14. Then (for the $F_{\text{final}\texttt{<summ>}}$ token and all tape tokens with $F_{\text{tape,fin}} = 1$) we compute $\text{bin}_r(i - i_{\texttt{<summ>}})$ by copying $I_{\text{pos}}$ into $I^k_{\text{searchpos}}$ and subtracting $I_{\text{pos},\texttt{<summ>}}$ in-place using Lemma C.17 starting at layer 4. Thus these tokens store the tape offset $p = i - i_{\texttt{<summ>}}$ in $I^k_{\text{searchpos}}$ for all $k$.

- (Extract final state and head positions.) In layer $L_2$ the $F_{\text{final}\texttt{<summ>}}$ token attends one position back (using $I_{\text{pos}\_}$) to extract the state $I_{\text{state}}$ and head positions $I^k_{\text{searchpos}}$ of the preceding run token into registers $I_{\text{state,fin}}$ and $I^k_{\text{hpos,fin}}$. In the next layer we zero these registers for all tokens except the $F_{\text{final}\texttt{<summ>}}$ token. Lastly, we broadcast $I_{\text{state,fin}}$ to all subsequent tokens in layer $L_2 + 2$ analogously to $I_{\text{pos},\texttt{<outp>}}$.

- (Marking heads.) In layer $L_2 + 2$ we set $F^k_{\text{head,next}}$ for each $k$ via an attention head analogous to $F^k_{\text{exist}}$: $I^k_{\text{hpos,fin}}$ remains non-zero only at the $F_{\text{final}\texttt{<summ>}}$ token and serves as the key source, so a tape token attends to the $F_{\text{final}\texttt{<summ>}}$ token iff its current offset $I^k_{\text{searchpos}}$ equals $I^k_{\text{hpos,fin}}$, and otherwise attends to the base token and outputs $0$.

- (Whether to output another tape token.) In layer $L_2 + 3$ we set a flag $F_{\text{next tape}}$ (for the $F_{\text{final}\texttt{<summ>}}$ token and all tape tokens with $F_{\text{tape,fin}} = 1$) indicating whether we should output another tape token, namely the OR over all $F^k_{\text{exist}}$ and $F^k_{\text{head,next}}$ flags.

- (Emitting tape tokens vs. the state token.) If $F_{\text{next tape}} = 1$, we copy the extracted tape symbols $I^k_{\text{sym,ex}}$ to registers $I^k_{\text{sym,next}}$ and map $F^k_{\text{head,next}}$ to $I^k_{\text{head,next}} \in \{-1, 1\}$ in layer $L_3 + 1$ so that the unembedding outputs the next tape token from $\mathcal{V}_{\text{tape}}$. If $F_{\text{next tape}} = 0$, we copy $I_{\text{state,fin}}$ to $I_{\text{state,fin,out}}$ in layer $L_2 + 4$ so that the unembedding outputs the corresponding state token. Finally, a state token always outputs $\texttt{</summ>}$.

**Dimensions.**  The depth $L = \frac{5r}{2} + 8$ and key/value dimensions $d_k = 4r - 1$, $d_v = \max\{r, d_Q, d_\Gamma\}$ are unchanged from the CoT construction. The number of heads per layer increases from $H = 3K$ to $H = 3K + 2$ because layer 3 already uses $2K$ heads for $\texttt{</p>}$ head-position extraction (from Theorem C.14) and we add $K$ heads to extract head positions and $1$ head to extract the state from the prompt summary, plus $1$ head to broadcast $I_{\text{pos},\texttt{<summ>}}$.

For $d_{\text{ff}}$, the first term increases from $18r + 2$ to $22r + 11$ due to the additional MLP operations in layers 1–4 for segment structure detection and length cap computation (in particular, writing $I_{\text{pos,promptend}}$ uses two $r$-bit conditional copies, one at $\texttt{</inp>}$ and one at $\texttt{</summ>}$). The second term increases from $14Kr + 2r$ to $18Kr + 2r + 1$ because the full subtraction operations for computing summary offsets (at layers 4 to $4 + r - 1$) overlap with the head propagation layers (at layers $L_1 + 1$ to $L_2 - 1$), adding $4Kr$ neurons to the head propagation layers. The third term gains $4Kd_\Gamma + 4K$ neurons for the tape token symbol extraction in the final summary.

The embedding dimension $d$ increases due to additional registers: $I_{\text{pos,promptend}}$ and $I_{\text{pos},\texttt{<summ>}}$ ($r$ bits each), $I_{\text{state,fin}}$ and $I_{\text{state,fin,out}}$ ($d_Q$ bits each), $I^k_{\text{hpos,fin}}$ ($Kr$ bits), $I^k_{\text{sym,next}}$ ($Kd_\Gamma$ bits), and $I^k_{\text{head,next}}$ ($K$ bits). Additional flags are $F_{\texttt{<summ>}}$, $F_{\texttt{</summ>}}$, $F_{\exists\texttt{</inp>}}$, $F_{\exists\texttt{</summ>}}$, $F_{\text{tape}}$, $F_Q$, $F_{\text{final}\texttt{<summ>}}$, $F_{\text{tape,init}}$, $F_{\text{tape,fin}}$, $F_{\text{length cap}}$, $F_{\to\texttt{<summ>}}$, $F_{\text{next tape}}$, $F^k_{\text{head}}$ ($K$ flags), $F^k_{\text{head,next}}$ ($K$ flags), and $F^s_{\text{bit equal}}$ for $s \in \{0, \ldots, r - 3\}$. $\qquad\square$

## C.5. Compiling Turing Machines into Transformers

**Compilation.**  To validate the correctness of the Turing machine constructions, we implement them in python. In particular, our code release implements a simple framework for constructing transformers by specifying registers and flags and applying basic reusable operations to manipulate them, fully analogous to how the constructions above work, without the need to manually specify parameters once the basic operations are implemented. This is used to implement the CoT and SCoT construction, i.e. write functions that compile a turing machine (and a length bound) into a PyTorch transformer simulating it with either CoT or SCoT.

**Validation.** For testing, we sample random Turing machines with a small number of states and tape symbols with all transitions chosen randomly, and corresponding short inputs. Each Turing machine is run on its input and, if it halts after a set maximum number of steps with an output in $\Sigma^*$, the corresponding transformer from Theorems C.14 and C.22 is constructed with $r$ chosen randomly from $\{r_{\min}, r_{\min} + 2, \ldots, r_{\min} + 8\}$ where $r_{\min}$ is the minimum even $r$ provided by the theorems. Finally, it is checked whether the transformer correctly generates the token sequences from Definitions C.12 and C.20. We ran this program for 10000 random Turing machine input pairs, where around $70 - 80\%$ are skipped due to not halting or halting with an output containing blank symbols. Details can be found in the provided code.

## D. Additional Material for Section 4

The goal of this section is to analyze the conversion of hardmax transformers into $c$-scaled softmax transformers, where query and key projections have been scaled by a constant $c > 0$ bringing the softmax attention weights closer to the hardmax ones. Additionally, the softmax transformer employs different rounding scenarios, whose error introduction and propagation will be analyzed. The goal in the end is to choose $c$ and the rounding precisions sufficiently high so that the softmax transformer generates the same tokens as the hardmax version. It should be noted that hardmax to softmax attention conversion via such scaling has been analyzed before, e.g. in (Yang et al., 2026) and (Liu et al., 2023), but without the rounding analysis and the adaptation to the exact properties of our hardmax constructions.

### D.1. Tools

**Definition D.1** (Separation). Let $s_0, \ldots, s_{n-1} \in \mathbb{R}$ and let $J \subset \{0, \ldots, n-1\}$ be non-empty. Then we define the *separation with respect to $J$* of the $s_j$ as

$$\text{sep}_J(s_0, \ldots, s_{n-1}) = \min_{j \in J} s_j - \max_{j \notin J} s_j$$

where $\max \emptyset = -\infty$, hence if $J = \{0, \ldots, n-1\}$ we have $\text{sep}_J(s_0, \ldots, s_{n-1}) = \infty$. Furthermore, let $s^* = \max s_j$ and $J^* = \{j \mid s_j = \max s_j\}$. Then we define the *separation* of the $s_j$ as

$$\text{sep}(s_0, \ldots, s_{n-1}) = \text{sep}_{J^*}(s_0, \ldots, s_{n-1}) .$$

**Lemma D.2.** *Let $s_0, \ldots, s_{n-1} \in \mathbb{R}$, $(\alpha_j)_{j=0}^{n-1} = \text{softmax}((s_j)_{j=0}^{n-1})$ and $J \subset \{0, \ldots, n-1\}$. Let $\beta = \text{sep}_J(s_0, \ldots, s_{n-1})$. Then*

$$\sum_{j \notin J} \alpha_j \leq ne^{-\beta} .$$

*Proof.* Let $j^* \in J$ arbitrary and $j \notin J$. Then

$$\alpha_j = \frac{e^{s_j}}{\sum_l e^{s_l}} \leq \frac{e^{s_j}}{e^{s_{j^*}}} = e^{s_j - s_{j^*}} \leq e^{-\beta}$$

and hence

$$\sum_{j \notin J} \alpha_j \leq (n - |J|)e^{-\beta} \leq ne^{-\beta}$$

$\square$

The following lemma gives a bound on the difference between hardmax and softmax, which could be applied in a very general setting to bound the difference between hardmax and softmax attention and derive a theorem like Theorem D.17. Note however that in this section we only use the more specialized statement above to get tighter bounds.

**Lemma D.3.** *Let $s_0, \ldots, s_{n-1} \in \mathbb{R}$ and let $\beta = \text{sep}(s_0, \ldots, s_{n-1})$. Then*

$$\| \text{hardmax}((s_j)_{j=0}^{n-1}) - \text{softmax}((s_j)_{j=0}^{n-1}) \|_1 \leq 2ne^{-\beta}$$

*where $\|x\|_1 = \sum_j |x_j|$ is the usual 1-norm.*

*Proof.* Let $J = \{j \mid s_j = \max_l s_l\}$, i.e. $\beta = \operatorname{sep}_J(s_0, \ldots, s_{n-1})$. Let $(\alpha_j)_{j=0}^{n-1} = \operatorname{softmax}((s_j)_{j=0}^{n-1})$ and $(\gamma_j)_{j=0}^{n-1} = \operatorname{hardmax}((s_j)_{j=0}^{n-1})$. Then

$$
\begin{aligned}
\sum_j |\alpha_j - \gamma_j| &= \sum_{j \in J} |\alpha_j - \gamma_j| + \sum_{j \notin J} |\alpha_j - \gamma_j| \\
&= \sum_{j \in J} |\alpha_j - \frac{1}{|J|}| + \sum_{j \notin J} \alpha_j \\
&= \sum_{j \in J} (\frac{1}{|J|} - \alpha_j) + \sum_{j \notin J} \alpha_j \\
&= 1 - (1 - \sum_{j \notin J} \alpha_j) + \sum_{j \notin J} \alpha_j \\
&\leq 2n e^{-\beta}
\end{aligned}
$$

where in the third step we used that for $j \in J$

$$
\alpha_j \leq \frac{e^{s_j}}{\sum_{j' \in J} e^{s_{j'}}} = \frac{1}{|J|} \, ,
$$

in the fourth step we used that the $\alpha_j$ sum to 1 and in the last step applied Lemma D.2. $\qquad \square$

### D.2. Error Propagation in a Single Transformer Layer

We will first analyze the conversion of a single hardmax transformer layer into a softmax variant, where rounding potentially introduces multiple errors at multiple points and a propagated input error might be present.

Here we set some notation for a single hardmax transformer layer acting on a single input, and its softmax variant with several (rounding) errors acting on a perturbed input. Note that we will later view the hardmax transformers as already incorporating the $c$-scaling, which does not alter their behavior, hence there is no reference to $c$ for now.

**Definition D.4** (Hardmax to Softmax Layer Setting). Let $d, d_{\mathrm{ff}}, d_k, d_v, H \in \mathbb{N}$ and consider parameters of a single transformer layer:
$$
\theta = ((W_Q^h, W_K^h, W_V^h, W_O^h)_{h=1}^H, (W_1, b, W_2)) \in \Theta^{\mathrm{TL}}(d, d_{\mathrm{ff}}, d_k, d_v, H)
$$
using the notation from Definition B.11. Let $x^{(\ell-1)} \in (\mathbb{R}^d)^n$ be an input[6] and $\tilde{x}^{(\ell-1)} \in (\mathbb{R}^d)^n$ a perturbed input.

We define the activations of the hardmax transformer layer with parameters $\theta$ as usual :

$$
q_i^h = W_Q^h x_i^{(\ell-1)}, \qquad k_j^h = W_K^h x_j^{(\ell-1)}, \qquad v_j^h = W_V^h x_j^{(\ell-1)},
$$

$$
(s_{ij}^h)_{j=0}^i = (\frac{\langle q_i^h, k_j^h \rangle}{\sqrt{d_k}})_{j=0}^i, \qquad (\alpha_{ij}^h)_{j \leq i} = \operatorname{hardmax}((s_{ij}^h)_{j \leq i}),
$$

$$
o_i^h = \sum_{j \leq i} \alpha_{ij}^h v_j^h, \qquad y_i = \sum_{h=1}^H W_O^h o_i^h, \qquad x_i^{(\ell-0.5)} = x_i^{(\ell-1)} + y_i,
$$

$$
a_i = \operatorname{ReLU}(W_1 x_i^{(\ell-0.5)} + b), \qquad z_i = W_2 a_i, \qquad x_i^{(\ell)} = x_i^{(\ell-0.5)} + z_i.
$$

Similarly we define the activations of the rounded softmax variant on the perturbed input. Let $\mathbb{F}_{\mathrm{act}}, \mathbb{F}_{\mathrm{att}}$ be finite sets (e.g. floating point sets from Definition B.18) or $\mathbb{R}$, where $[\cdot]_{\mathbb{F}}$ denotes (coordinate-wise) rounding to a set $\mathbb{F}$ (see Definition B.18) and rounding to $\mathbb{R}$ does nothing. Exact arithmetic yields activations like e.g. $\bar{q}_i$, rounding then obtains $\tilde{q}_i$.

$$
\bar{q}_i^h = W_Q^h \tilde{x}_i^{(\ell-1)}, \tilde{q}_i^h = [\bar{q}_i^h]_{\mathbb{F}_{\mathrm{act}}}, \qquad \bar{k}_j^h = W_K^h \tilde{x}_j^{(\ell-1)}, \tilde{k}_j^h = [\bar{k}_j^h]_{\mathbb{F}_{\mathrm{act}}}, \qquad \bar{v}_j^h = W_V^h \tilde{x}_j^{(\ell-1)}, \tilde{v}_j^h = [\bar{v}_j^h]_{\mathbb{F}_{\mathrm{act}}},
$$

---

[6]We only consider a single layer, hence $\ell$ has no value and is only used for notational clarity to distinguish activations before attention, after attention and after the MLP. The layer index is omitted in intermediate activations.

$$(\tilde{s}_{ij}^h)_{j=0}^i = (\frac{\langle \tilde{q}_i^h, \tilde{k}_j^h \rangle}{\sqrt{d_k}})_{j=0}^i, \qquad (\bar{\alpha}_{ij}^h)_{j \leq i} = \mathrm{softmax}\big((\tilde{s}_{ij}^h)_{j \leq i}\big), \tilde{\alpha}_{ij}^h = [\bar{\alpha}_{ij}^h]_{\mathbb{F}_{\mathrm{att}}},$$

$$\bar{o}_i^h = \sum_{j \leq i} \tilde{\alpha}_{ij}^h \tilde{v}_j^h, \tilde{o}_i^h = [\bar{o}_i^h]_{\mathbb{F}_{\mathrm{act}}}, \qquad \bar{y}_i = \sum_{h=1}^H W_O^h \tilde{o}_i^h, \tilde{y}_i = [\bar{y}_i]_{\mathbb{F}_{\mathrm{act}}},$$

$$\bar{x}_i^{(\ell-0.5)} = \tilde{x}_i^{(\ell-1)} + \tilde{y}_i, \qquad \tilde{x}_i^{(\ell-0.5)} = [\bar{x}_i^{(\ell-0.5)}]_{\mathbb{F}_{\mathrm{act}}},$$

$$\bar{a}_i = \mathrm{ReLU}(W_1 \tilde{x}_i^{(\ell-0.5)} + b), \tilde{a}_i = [\bar{a}_i]_{\mathbb{F}_{\mathrm{act}}}, \qquad \bar{z}_i = W_2 \tilde{a}_i, \tilde{z}_i = [\bar{z}_i]_{\mathbb{F}_{\mathrm{act}}}, \qquad \bar{x}_i^{(\ell)} = \tilde{x}_i^{(\ell-0.5)} + \tilde{z}_i, \qquad \tilde{x}_i^{(\ell)} = [\bar{x}_i^{(\ell)}]_{\mathbb{F}_{\mathrm{act}}}.$$

Next, we define the maximum errors between $\bar{\cdot}, \tilde{\cdot}$ and their unperturbed counterparts. For the input we set

$$\Delta_{x^{(\ell-1)}} = \max_i \|\tilde{x}_i^{(\ell-1)} - x_i^{(\ell-1)}\|_\infty .$$

For any vector-valued activation $u \in \{q, k, v, o, y, x^{(\ell-0.5)}, a, z, x^{(\ell)}\}$ we define

$$\bar{\Delta}_u = \max_\alpha \|\bar{u}_\alpha - u_\alpha\|_\infty, \qquad \Delta_u = \max_\alpha \|\tilde{u}_\alpha - u_\alpha\|_\infty,$$

where the maximum ranges over all indices of $u$ (heads and positions when present), e.g.

$$\bar{\Delta}_q = \max_{h,i} \|\bar{q}_i^h - q_i^h\|_\infty .$$

Moreover, we set

$$\Delta_s = \max_{h,i,j \leq i} |\tilde{s}_{ij}^h - s_{ij}^h|, \qquad \bar{\Delta}_\alpha = \max_{h,i} \sum_{j \leq i} |\bar{\alpha}_{ij}^h - \alpha_{ij}^h|, \qquad \Delta_\alpha = \max_{h,i} \sum_{j \leq i} |\tilde{\alpha}_{ij}^h - \alpha_{ij}^h|$$

and define the total attention-weight rounding error as

$$\delta_\alpha^{\mathrm{round}} = \max_{h,i} \sum_{j \leq i} \big| [\bar{\alpha}_{ij}^h]_{\mathbb{F}_{\mathrm{att}}} - \bar{\alpha}_{ij}^h \big| .$$

Moreover we define

$$M = \max_i \|x_i^{(\ell-1)}\|_\infty$$

and

$$L_Q := \max_{h \in \{1,...,H\}} \|W_Q^h\|_\infty, \qquad L_K := \max_{h \in \{1,...,H\}} \|W_K^h\|_\infty, \qquad L_V := \max_{h \in \{1,...,H\}} \|W_V^h\|_\infty, \qquad L_O := \max_{h \in \{1,...,H\}} \|W_O^h\|_\infty,$$

*Remark* D.5. The setting above defines the points where rounding is applied in our analyses. In particular, queries, keys, values, hidden representations ($x^{(0)}, x^{(0.5)}, x^{(1)}, \dots$), the outputs of each attention head before projecting with $W_O^h$, the output of each multihead attention, the hidden activations inside each MLP and the output of each MLP are rounded to $\mathbb{F}_{\mathrm{act}}$ while the attention weights are rounded to $\mathbb{F}_{\mathrm{att}}$. While there does not appear to be a single standard rounding scheme, running a transformer or other neural network at some low precision (say 16 bit) usually means that large matrix multiplications inside attention and the MLP have 16 bit inputs and 32 bit accumulation and outputs (Micikevicius et al., 2018). Applying rounding to all intermediate activations before being used in large matrix multiplications precisely yields the rounding points specified in Definition D.4, which is the motivation for this choice. In this regime, the attention scores $s_{ij}$ are not rounded as they are only fed into the softmax, and this seems to be the case as well in fused attention implementations (Dao et al., 2022; Dao, 2024). Furthermore, we do not include errors from rounding to higher precision (e.g. to 32-bit during accumulation) and errors made when applying softmax, as these are typically small compared to errors from rounding to 16-bit precision or lower. However, we note that such errors and in fact any general perturbations to intermediate activations could be analyzed and bounded with the same techniques used in this section leading to slightly different statements.

Next, we show upper bounds for the errors in Definition D.4. First, we bound all the pre-rounding errors $\bar{\Delta}_{...}$ from the post-rounding errors $\tilde{\Delta}_{...}$ of their inputs, then consider different rounding scenarios to bound the rounding errors.

**Pre-rounding Errors**

**Lemma D.6.** *Consider the setting of Definition D.4. Then*

$$\bar{\Delta}_q \leq L_Q \Delta_{x^{(\ell-1)}}, \qquad \bar{\Delta}_k \leq L_K \Delta_{x^{(\ell-1)}}, \qquad \bar{\Delta}_v \leq L_V \Delta_{x^{(\ell-1)}} .$$

*Proof.* It holds that

$$\|\bar{q}_i^h - q_i^h\|_\infty = \|W_Q^h(\tilde{x}_i^{(\ell-1)} - x_i^{(\ell-1)})\|_\infty \leq \|W_Q^h\|_\infty \|\tilde{x}_i^{(\ell-1)} - x_i^{(\ell-1)}\|_\infty \leq L_Q \Delta_{x^{(\ell-1)}} .$$

Taking the maximum over $h, i$ yields the statement for $q$. The bounds for $k$ and $v$ are analogous. □

**Lemma D.7.** *Consider the setting of Definition D.4. Then*

$$\Delta_s \leq \sqrt{d_k}\Big(\Delta_q(L_K M + \Delta_k) + \Delta_k L_Q M\Big) .$$

*Proof.* It holds that

$$|\tilde{s}_{ij}^h - s_{ij}^h| = \frac{1}{\sqrt{d_k}}|\langle \tilde{q}_i^h, \tilde{k}_j^h\rangle - \langle q_i^h, k_j^h\rangle| \leq \frac{1}{\sqrt{d_k}}\Big(|\langle \tilde{q}_i^h - q_i^h, \tilde{k}_j^h\rangle| + |\langle q_i^h, \tilde{k}_j^h - k_j^h\rangle|\Big) \leq \sqrt{d_k}\Delta_q\|\tilde{k}_j^h\|_\infty + \sqrt{d_k}\|q_i^h\|_\infty\Delta_k .$$

where we used $|\langle a, b\rangle| \leq d_k\|a\|_\infty\|b\|_\infty$ for $a, b \in \mathbb{R}^{d_k}$. Furthermore,

$$\|q_i^h\|_\infty \leq L_Q\|x_i^{(\ell-1)}\|_\infty \leq L_Q M$$

and

$$\|\tilde{k}_j^h\|_\infty \leq \|k_j^h\|_\infty + \|\tilde{k}_j^h - k_j^h\|_\infty \leq L_K M + \Delta_k ,$$

taking the maximum over $h, i, j \leq i$ yields the assertion. □

**Lemma D.8.** *Consider the setting from Definition D.4 and let*

$$\beta := \min_{h,i} \text{sep}((s_{ij}^h)_{j=0}^i)$$

*be the minimal separation of the unperturbed scores (see Definition D.1). Assume that values are* tie-invariant*: for each $h, i$ let*

$$J_i^h := \{j \leq i \mid s_{ij}^h = \max_{l \leq i} s_{il}^h\}$$

*be the set of indices that position $i$ attends to in the hardmax attention. Then we assume that $v_j^h = v_{j'}^h$ for all $j, j' \in J_i^h$, i.e. the values are the same for all positions being attended to. This is a slight generalization of the assumption of maxima being unique in the hardmax attention. Then*

$$\bar{\Delta}_o \leq (1 + \delta_\alpha^{\text{round}})\Delta_v + \big(2ne^{-(\beta-2\Delta_s)} + \delta_\alpha^{\text{round}}\big)L_V M$$

*Proof.* It holds that

$$\|\bar{o}_i^h - o_i^h\|_\infty = \|\sum_{j \leq i}\tilde{\alpha}_{ij}^h\tilde{v}_j^h - \sum_{j \leq i}\alpha_{ij}^h v_j^h\|_\infty \leq \underbrace{\|\sum_{j \leq i}\tilde{\alpha}_{ij}^h(\tilde{v}_j^h - v_j^h)\|_\infty}_{I_1} + \underbrace{\|\sum_{j \leq i}(\tilde{\alpha}_{ij}^h - \alpha_{ij}^h)v_j^h\|_\infty}_{I_2} .$$

The first term can be bounded using the triangle inequality:

$$I_1 \leq \sum_{j \leq i}|\tilde{\alpha}_{ij}^h|\|\tilde{v}_j^h - v_j^h\|_\infty \leq \sum_{j \leq i}(\bar{\alpha}_{ij}^h + |\tilde{\alpha}_{ij}^h - \bar{\alpha}_{ij}^h|)\Delta_v \leq \Delta_v(1 + \delta_\alpha^{\text{round}})$$

using that the $\bar{\alpha}_{ij}^h$ sum to 1. To bound the second term we split it into the sum over $J_i^h$ and the rest, use that the $\alpha_{ij}^h$ are only non-zero on $J_i^h$ and the tie-invariance assumption. In particular, let $j^* \in J_i^h$ be arbitrary. Then

$$
\begin{aligned}
I_2 &= \|\sum_{j \le i} \alpha_{ij}^h v_j^h - \sum_{j \le i} \tilde{\alpha}_{ij}^h v_j^h\|_\infty \\
&= \|v_{j^*}^h - \sum_{j \le i} \tilde{\alpha}_{ij}^h v_j^h\|_\infty \\
&= \|v_{j^*}^h - \sum_{j \in J_i^h} \bar{\alpha}_{ij}^h v_j^h - \sum_{j \notin J_i^h} \bar{\alpha}_{ij}^h v_j^h - \sum_{j \le i} ([\bar{\alpha}_{ij}^h]_{\mathbb{F}_{att}} - \bar{\alpha}_{ij}^h) v_j^h\|_\infty \\
&\le |1 - \sum_{j \in J_i^h} \bar{\alpha}_{ij}^h| \|v_{j^*}^h\|_\infty + \sum_{j \notin J_i^h} \bar{\alpha}_{ij}^h \|v_j^h\|_\infty + \sum_{j \le i} |[\bar{\alpha}_{ij}^h]_{\mathbb{F}_{att}} - \bar{\alpha}_{ij}^h| \|v_j^h\|_\infty \\
&\le (2 \sum_{j \notin J_i^h} \bar{\alpha}_{ij}^h + \delta_\alpha^{round}) L_V M
\end{aligned}
$$

Next, we use Lemma D.2 to bound the first sum in the last term. For $j \in J_i^h$ and $l \notin J_i^h$ it holds that

$$
\tilde{s}_{ij}^h - \tilde{s}_{il}^h \ge (s_{ij}^h - \Delta_s) - (s_{il}^h + \Delta_s) \ge \beta - 2\Delta_s,
$$

hence we can apply Lemma D.2 with $J = J_i^h$ and separation at least $\beta - 2\Delta_s$ to obtain

$$
I_2 \le \left(2n e^{-(\beta - 2\Delta_s)} + \delta_\alpha^{round}\right) L_V M .
$$

Combining the bounds yields the statement. □

Next, we bound $\Delta_y$ from $\Delta_o$ under the assumption that all heads write their output to disjoint subspaces, again an assumption which holds in our hardmax constructions.

**Lemma D.9.** *Consider the setting of Definition D.4. Assume that for each coordinate $l \in \{1, \ldots, d\}$ there exists at most one head $h \in \{1, \ldots, H\}$ such that row $l$ of $W_O^h$ is non-zero. Then*

$$
\bar{\Delta}_y \le L_O \Delta_o
$$

*Proof.* We have

$$
\begin{aligned}
\|\bar{y}_i - y_i\|_\infty &= \|\sum_{h \in \{1, \ldots, H\}} W_O^h (\tilde{o}_i^h - o_i^h)\|_\infty \\
&= \max_{l \in \{1, \ldots, d\}} |\sum_{h \in \{1, \ldots, H\}} \langle (W_O^h)_{l,:}, \tilde{o}_i^h - o_i^h \rangle| \\
&\le \max_{l \in \{1, \ldots, d\}} \max_{h \in \{1, \ldots, H\}} |\langle (W_O^h)_{l,:}, \tilde{o}_i^h - o_i^h \rangle| \\
&= \max_{h \in \{1, \ldots, H\}} \|W_O^h (\tilde{o}_i^h - o_i^h)\|_\infty \\
&\le L_O \Delta_o
\end{aligned}
$$

where $A_{l,:}$ denotes the $l$'th row of a matrix $A$ and we used the assumption in the third line. □

**Lemma D.10.** *Consider the setting of Definition D.4. Then*

$$
\bar{\Delta}_{x^{(\ell - 0.5)}} \le \Delta_y + \Delta_{x^{(\ell-1)}}
$$

*Proof.* We have

$$
\|\bar{x}_i^{(\ell-0.5)} - x_i^{(\ell-0.5)}\|_\infty = \|\tilde{x}_i^{(\ell-1)} + \tilde{y}_i - (x_i^{(\ell-1)} + y_i)\|_\infty \le \|\tilde{x}_i^{(\ell-1)} - x_i^{(\ell-1)}\|_\infty + \|\tilde{y}_i - y_i\|_\infty \le \Delta_{x^{(\ell-1)}} + \Delta_y
$$

□

**Lemma D.11.** *Consider the setting from Definition D.4. Then*

$$\bar{\Delta}_a \leq \|W_1\|_\infty \Delta_{x^{(\ell-0.5)}}, \quad \bar{\Delta}_z \leq \|W_2\|_\infty \Delta_a, \quad \bar{\Delta}_{x^{(\ell)}} \leq \Delta_z + \Delta_{x^{(\ell-0.5)}}$$

*Proof.* The proof is analogous to Lemmas D.6 and D.10 using that ReLU is 1-Lipschitz and hence omitted. □

**Rounding Errors.** Next we consider the errors introduced during rounding. Clearly, the case where $\mathbb{F}_{act} = \mathbb{R}$ trivially leads to $\Delta_q = \bar{\Delta}_q$ etc, similarly for $\mathbb{F}_{att} = \mathbb{R}$.

Another easy to handle case is when the unperturbed activations are already in $\mathbb{F}_{act}$, in which case rounding can at most double the perturbation by the following lemma:

**Lemma D.12.** *Let $\mathbb{F} \subset \mathbb{R}$ finite, $x \in \mathbb{F}$ and $y \in \mathbb{R}$. Then*

$$|[x+y]_\mathbb{F} - x| \leq 2|y|$$

*Proof.* This follows directly from the definition of rounding as projecting to the closest number in $\mathbb{F}$: as $x \in \mathbb{F}$, it must hold that

$$|(x+y) - [x+y]_\mathbb{F}| \leq |(x+y) - x| = |y|$$

and hence

$$|[x+y]_\mathbb{F} - x| \leq |[x+y]_\mathbb{F} - (x+y)| + |(x+y) - x| \leq |y| + |y| = 2|y| \;.$$

□

This is applied several times to activations, but never to weights as weights in the hardmax transformer are of the form $\alpha_{ij}^h = \frac{1}{k}$, which are generally (e.g. for $n = 3$) not representable in any floating point format. The following corollary formalizes the application to activations in $\mathbb{F}_{act}$ and follows trivially from the previous lemma.

**Corollary D.13.** *Consider the setting from Definition D.4. Assume that activations $x^{(\ell-0.5)}, x^{(\ell)}, q_i^h, k_j^h, v_j^h, o_i^h, y_i, a_i, z_i$ have entries in $\mathbb{F}_{act}$. Then rounding at most doubles their perturbation, i.e. $\Delta_q \leq 2\bar{\Delta}_q, \Delta_k \leq 2\bar{\Delta}_k$ etc.*

Properly bounding the summed rounding errors for attention weights occurring in Lemma D.8 requires viewing weights $\alpha_{ij}^h$ for $j$ in $J_i^h$ and $j \notin J_i^h$ separately. We will only consider rounding of attention weights in Theorem D.17, where we will have $\tilde{s}_{ij}^h = s_{ij}^h$ (i.e. $\Delta_s = 0$) and hence only treat that case.

**Lemma D.14.** *Consider the setting of Definition D.4 and let*

$$\beta := \min_{h,i} \text{sep}\left((s_{ij}^h)_{j=0}^i\right)$$

*be the minimal separation of the unperturbed scores (see Definition D.1). Assume $\mathbb{F}_{att} = \mathbb{F}_{b_m,b_e}$ for some $b_m \in \mathbb{N}, b_e \in \mathbb{N}_{\geq 2}$ where $\frac{1}{n} \geq 2^{E_{\min}}$ and assume $\Delta_s = 0$. Then for each $i$ and $h$ it holds that*

$$\sum_{j \leq i} |\bar{\alpha}_{ij}^h - [\bar{\alpha}_{ij}^h]_{\mathbb{F}_{att}}| \leq 2^{-b_m-1} + ne^{-\beta}$$

*Proof.* Let $J_i^h = \{j \leq i \mid s_{ij}^h = \max_{l \leq i} s_{il}^h\}$. Then

$$\sum_{j \leq i} |\bar{\alpha}_{ij}^h - [\bar{\alpha}_{ij}^h]_{\mathbb{F}_{att}}| \leq \sum_{j \in J_i^h} |\bar{\alpha}_{ij}^h - [\bar{\alpha}_{ij}^h]_{\mathbb{F}_{att}}| + \sum_{j \notin J_i^h} |\bar{\alpha}_{ij}^h - [\bar{\alpha}_{ij}^h]_{\mathbb{F}_{att}}|$$

For the first term, note that for $j \in J_i^h$ we have

$$\bar{\alpha}_{ij}^h = \frac{e^{s_{ij}^h}}{\sum_{l \leq i} e^{s_{il}^h}} \geq \frac{e^{s_{ij}^h}}{ne^{s_{ij}^h}} = \frac{1}{n}$$

and hence the attention weight $\bar{\alpha}_{ij}^h$ is in the normal range of $\mathbb{F}_{\text{att}}$. Thus we can apply Lemma B.19 to obtain

$$\sum_{j \in J_i^h} |\bar{\alpha}_{ij}^h - [\bar{\alpha}_{ij}^h]_{\mathbb{F}_{\text{att}}}| \leq 2^{-b_m-1} \sum_{j \in J_i^h} \bar{\alpha}_{ij}^h \leq 2^{-b_m-1} .$$

For the second term, we note that $0 \in \mathbb{F}_{\text{att}}$, and hence $|\bar{\alpha}_{ij}^h - [\bar{\alpha}_{ij}^h]_{\mathbb{F}_{\text{att}}}| \leq |\bar{\alpha}_{ij}^h - 0|$ due to the definition of rounding (Definition B.18). Thus

$$\sum_{j \notin J_i^h} |\bar{\alpha}_{ij}^h - [\bar{\alpha}_{ij}^h]_{\mathbb{F}_{\text{att}}}| \leq \sum_{j \notin J_i^h} \bar{\alpha}_{ij}^h \leq ne^{-\beta}$$

where in the second step we applied Lemma D.2, noting that the separation between scores $s_{ij}^h$ is at least $\beta$. Together these yield the statement. $\qquad\square$

### D.3. Properties of Hardmax Transformer Constructions

Next, we list some statements about the hardmax transformers from our constructions.

**Lemma D.15.** *Let $T$ be one of the hardmax transformers from Theorems C.8, C.14 and C.22 with vocabulary $\mathcal{V}$ and let $\mathcal{X} \subset \mathcal{V}^+$ be the set of inputs it can receive, for the CoT/SCoT transformers (Theorems C.14 and C.22) that includes any intermediate token sequence during generation. Then*

1. *In each layer, the attention matrices $W_Q^{\ell,h}, W_K^{\ell,h}, W_V^{\ell,h}, W_O^{\ell,h}$ have at most one non-zero entry per row which is $1$.*

2. *In each layer, the output projections $(W_O^{\ell,h})_{h=1}^H$ write to disjoint coordinates, i.e. they satisfy the precondition from Lemma D.9.*

3. *In each layer, the MLP matrices $W_1, W_2$ satisfy*

$$\|W_1\|_\infty \leq d , \ \|W_2\|_\infty \leq d_{ff} .$$

4. *The embeddings and unembeddings have entries in $\{-1, 0, 1\}$.*

5. *For each $x \in \mathcal{X}$, all heads and layers have tie-invariant values, i.e. the condition of Lemma D.8 is satisfied in each layer.*

6. *For each $x \in \mathcal{X}$, all activations have entries in $\{-1, 0, 1\}$.*

7. *The output scores*

$$\langle unemb_v, x_{n-1}^{(L)} \rangle$$

*have a unique maximum which is larger than other scores by at least $1$.*

*Proof.* **Item 1** holds because all queries, keys and values are concatenations of registers/flags and write their output to some register/flag (cf. Appendix C.1). **Item 2** holds because no two heads in the same layer write to the same output register/flag. **Item 3** follows from all entries of $W_1$ and $W_2$ being ternary, hence bounded by 1, and from

$$\|A\|_\infty = \max_i \sum_j |A_{ij}| .$$

**Item 4** simply holds by definition of the embeddings and unembeddings for each of the constructions. **Item 5** requires to distinguish several types of attention heads in the hardmax transformer constructions. Some attention heads always lead to unique maximizers by design, which automatically leads to tie-invariant values. This holds for heads that attend to a specific position (e.g. attending $s$ positions back) or a specific token that occurs only once in context (e.g. an attention head attending to `</inp>` if it exists and otherwise falls back to `<inp>`). This holds for all attention heads in Theorem C.8, but not in Theorems C.14 and C.22. Other heads broadcast register to later positions, e.g. the position of `<outp>` in the construction for Theorem C.14. For these heads, positions before the respective token has occured attend uniformly to previous tokens which all have value 0, i.e. again tie-invariant. Next, the attention heads in Theorems C.14 and C.22 which check if a token at the current head position exists at all (writing to $F_{\text{exist}}^k$) or exists in the narrowed-down range in the binary search for the

position of this token (writing to $F^k_{\exists\text{high}}$) attend to multiple positions uniformly if a token is found and otherwise to the first token. The values however are the same for all tokens except the first one, hence again leading to tie-invariant values. Lastly, some layers don't use all $H$ heads, as explained in Appendix C.1, for these heads we simply set all attention matrices to 0 which trivially leads to tie-invariant values (all zero). **Item 6** holds at initialization because of Item 4, holds for queries, keys and values because of Item 1, and holds for attention outputs because of Item 1 together with Item 5 (in the tie case, hardmax averages over identical values). It holds for MLP hidden activations and outputs by construction of the MLP gadgets, and for the hidden representations because MLPs and attention heads only write non-zero output to a register/flag if the register/flag was zero before at that position. **Item 7** holds because output unembeddings are constructed to make the maximum score unique, and due to Item 6 and Item 4 scores are in $\mathbb{Z}$ and hence the gap is at least 1. $\qquad\square$

### D.4. Softmax Conversion with Exact Attention

**Theorem D.16.** *Let $T \in \mathcal{T}^{\text{hard}}(d, d_{ff}, d_k, d_v, H, L, \mathcal{V})$ be a hardmax transformer and let $\mathcal{X} \subseteq \mathcal{V}^+$ be a set of inputs of length at most $N$ such that the statements in Lemma D.15 are satisfied for $T$ and $\mathcal{X}$. Let $\mathbb{F}_{act}$ be a floating point format (see Definition B.18). Let $c \in \mathbb{F}_{act}$ such that*

$$c \geq \left(2\sqrt{d_k}\log\left(\frac{16}{6}\max\{4d, 20d_k\}N\left(48dd_{ff} + 12\right)^L\right)\right)^{\frac{1}{2}} . \tag{3}$$

*and consider the softmax transformer $\tilde{T}$ with the same parameters as $T$ except that query- and key-projections are scaled by $c$ and activations are rounded to $\mathbb{F}_{act}$ but attention weights are not rounded (i.e. $\mathbb{F}_{att} = \mathbb{R}$ in the notation of Definition D.4). Then*

$$\tilde{T}|_{\mathcal{X}} = T|_{\mathcal{X}} ,$$

*i.e. $\tilde{T}$ outputs the same tokens as $T$ for inputs from $\mathcal{X}$.*

*Proof.* Let $T_c$ be the hardmax transformer obtained from $T$ by scaling query- and key projections by $c$, i.e. $T_c$ and $\tilde{T}$ have the same parameters. This does not influence the output, i.e. $T|_{\mathcal{X}} = T_c|_{\mathcal{X}}$. Furthermore, as $T$ has ternary activations (Item 6) on $\mathcal{X}$, the inner products $\langle q_i, k_j \rangle$ are in $\mathbb{Z}$ and after scaling by $\frac{1}{\sqrt{d_k}}$ the scores $s_{ij}$ have separation at least $\frac{1}{\sqrt{d_k}}$. Hence, $T_c$ has separation at least

$$\beta := \frac{c^2}{\sqrt{d_k}} .$$

Now let $x \in \mathcal{X}$ be arbitrary. Denoting by $x^{(\ell)}, \tilde{x}^{(\ell)}$ the hidden activations of $T_c$ and $\tilde{T}$ on $x$. Define

$$\varepsilon_\ell := \max_i \|\tilde{x}_i^{(\ell)} - x_i^{(\ell)}\|_\infty .$$

We will first show a recurrence $\varepsilon_\ell \leq A\varepsilon_{\ell-1} + B$ and then use that to bound the error in $\varepsilon_L$ to ensure that the outputs are the same.

**Layerwise error propagation.** Let $\ell \in \{1, \ldots, L\}$ be arbitrary. We now use the notation from Definition D.4, dropping layer indices for simplicity. We write $\varepsilon := \varepsilon_{\ell-1}$ which in the notation of Definition D.4 is precisely $\Delta_{x^{(\ell-1)}}$. For this layerwise error bound we assume

$$\varepsilon_{\ell-1} = \varepsilon \leq \min(\frac{1}{4d}, \frac{1}{20d_k}) . \tag{4}$$

Note that all activations of $T_c$ are in $\{0, \pm1, \pm c\} \subset \mathbb{F}_{act}$, hence we can always bound the rounding error using Lemma D.12. Hence by Lemma D.6 we obtain

$$\Delta_q \leq 2\bar{\Delta}_q \leq 2L_Q\varepsilon \leq 2c\varepsilon$$

using that $L_Q \leq c$ due to Item 1, and analogously

$$\Delta_k \leq 2c\varepsilon .$$

Due to Item 1, entries of the values $\bar{v}_j$ are just entries from $\tilde{x}^{(\ell-1)}$, which are already rounded to $\mathbb{F}_{act}$, hence rounding has no effect and we obtain

$$\Delta_v = \bar{\Delta}_v \leq \varepsilon .$$

Next we apply Lemma D.7 to bound the deviation in the scores as

$$
\begin{aligned}
\Delta_s &\leq \sqrt{d_k}\Big(\Delta_q(L_K M + \Delta_k) + \Delta_k L_Q M\Big) \\
&\leq \sqrt{d_k}(2c\varepsilon(c + 2c\varepsilon) + 2c\varepsilon c) \\
&= \sqrt{d_k}c^2\varepsilon(4 + 4\varepsilon) \\
&\leq 5\sqrt{d_k}c^2\varepsilon
\end{aligned}
$$

where in the last step we used $\varepsilon \leq \frac{1}{4d}$.

Now apply Lemma D.8, where the weight-rounding terms are zero as weights are not rounded, to obtain

$$
\begin{aligned}
\Delta_o &\leq 2\bar{\Delta}_o \\
&\leq 2(\Delta_v + 2ne^{-(\beta - 2\Delta_s)}L_V M) \\
&\leq 2(\varepsilon + 2ne^{-(\beta - 2\Delta_s)})
\end{aligned}
$$

As $\beta \geq \frac{c^2}{\sqrt{d_k}}$ we have

$$
\beta - 2\Delta_s \geq \frac{c^2}{\sqrt{d_k}}(1 - 2(5d_k\varepsilon)) \geq \frac{1}{2}\frac{c^2}{\sqrt{d_k}}
$$

using $\varepsilon \leq \frac{1}{20d_k}$ in the second step. Hence we obtain

$$
\Delta_o \leq 2\varepsilon + 4ne^{-\frac{1}{2}\frac{c^2}{\sqrt{d_k}}} .
$$

Due to Items 1 and 2, applying the output projections $W_O^h$ and summing introduces no new rounding error as each entry of $\bar{y}_i$ is an entry of some $\tilde{o}_i^h$, hence already in $\mathbb{F}_{\text{act}}$. Thus

$$
\Delta_y = \bar{\Delta}_y \leq L_O\Delta_o \leq \Delta_o .
$$

where we used Lemma D.9 and $L_O \leq 1$.

Next, we can bound the error in $x^{(\ell-0.5)}$ using Lemma D.10 as

$$
\Delta_{x^{(\ell-0.5)}} \leq 2\bar{\Delta}_{x^{(\ell-0.5)}} \leq 2(\Delta_{x^{(\ell-1)}} + \Delta_y) \leq 2(\varepsilon + 2\varepsilon + 4ne^{-\frac{1}{2}\frac{c^2}{\sqrt{d_k}}}) = 6\varepsilon + 8ne^{-\frac{1}{2}\frac{c^2}{\sqrt{d_k}}}
$$

Lastly we apply Lemma D.11 and Lemma D.12 to obtain

$$
\begin{aligned}
\Delta_{x^{(\ell)}} &\leq 2\bar{\Delta}_{x^{(\ell)}} \\
&\leq 2(\Delta_z + \Delta_{x^{(\ell-0.5)}}) \\
&\leq 2\big(2\|W_2\|_\infty(2\|W_1\|_\infty\Delta_{x^{(\ell-0.5)}}) + \Delta_{x^{(\ell-0.5)}}\big) \\
&= (8\|W_1\|_\infty\|W_2\|_\infty + 2)\Delta_{x^{(\ell-0.5)}} \\
&\leq \underbrace{(8dd_{\text{ff}} + 2)}_{=:L_{\text{ff}}}(6\varepsilon + 8ne^{-\frac{1}{2}\frac{c^2}{\sqrt{d_k}}})
\end{aligned}
$$

where in the last step we applied Item 3.

**End-to-end propagation.** We have shown that for each $\ell \in \{1, \ldots, L\}$ where $\varepsilon_{\ell-1} \leq \min(\frac{1}{4d}, \frac{1}{20d_k})$ it holds that

$$
\varepsilon_\ell \leq A\varepsilon_{\ell-1} + B
$$

with

$$
A = 6L_{\text{ff}}, B = 8L_{\text{ff}}ne^{-\frac{1}{2}\frac{c^2}{\sqrt{d_k}}} .
$$

We show inductively that the condition holds for all $\ell$ and at the same time bound $\varepsilon_L$. Since $T_c$ and $\tilde{T}$ share the same embeddings, which are ternary by Item 4 and hence exactly representable in $\mathbb{F}_{\text{act}}$ (hence no rounding error occurs there), we have $\varepsilon_0 = 0$. Now assume condition (4) holds for all layers $\ell' \leq \ell$. Then iterating the recurrence $\varepsilon_{\ell'} \leq A\varepsilon_{\ell'-1} + B$ for $\ell' = \ell, \ell-1, \ldots, 1$ yields

$$\varepsilon_\ell \leq \sum_{j=0}^{\ell-1} A^j B.$$

Since $A \geq 2$, we have $\sum_{j=0}^{\ell-1} A^j \leq 2A^{\ell-1}$ and hence

$$\varepsilon_\ell \leq 2BA^{\ell-1} = 16 L_{\text{ff}} n e^{-\frac{1}{2}\frac{c^2}{\sqrt{d_k}}} (6L_{\text{ff}})^{\ell-1} \leq 16 L_{\text{ff}} n e^{-\frac{1}{2}\frac{c^2}{\sqrt{d_k}}} (6L_{\text{ff}})^{L-1}$$

Now if we choose $c$ sufficiently large such that

$$16 L_{\text{ff}} n e^{-\frac{1}{2}\frac{c^2}{\sqrt{d_k}}} (6L_{\text{ff}})^{L-1} \leq \min(\frac{1}{4d}, \frac{1}{20d_k}). \tag{5}$$

then $\varepsilon_\ell \leq \min(\frac{1}{4d}, \frac{1}{20d_k})$. Hence, if $c$ satisfies (5), then condition (4) holds for all $\ell \leq L$ and hence we obtain

$$\varepsilon_L \leq 16 L_{\text{ff}} n e^{-\frac{1}{2}\frac{c^2}{\sqrt{d_k}}} (6L_{\text{ff}})^{L-1} \leq \frac{1}{4d}.$$

Finally, let $v^* = \arg\max_{v \in \mathcal{V}} \langle x_{n-1}^{(L)}, \text{unemb}_v \rangle$ be the output token of $T_c$ (equivalently $T$) on $x$. For any $v \in \mathcal{V} \setminus \{v^*\}$,

$$\begin{aligned}
\langle \tilde{x}_{n-1}^{(L)}, \text{unemb}_{v^*} \rangle - \langle \tilde{x}_{n-1}^{(L)}, \text{unemb}_v \rangle &= \langle \tilde{x}_{n-1}^{(L)}, \text{unemb}_{v^*} - \text{unemb}_v \rangle \\
&= \langle x_{n-1}^{(L)}, \text{unemb}_{v^*} - \text{unemb}_v \rangle + \langle \tilde{x}_{n-1}^{(L)} - x_{n-1}^{(L)}, \text{unemb}_{v^*} - \text{unemb}_v \rangle \\
&\geq 1 - \|\tilde{x}_{n-1}^{(L)} - x_{n-1}^{(L)}\|_\infty \|\text{unemb}_{v^*} - \text{unemb}_v\|_1 \\
&\geq 1 - 2d\varepsilon_L \geq \frac{1}{2} > 0,
\end{aligned}$$

where we used Item 7 and $\|\text{unemb}_{v^*} - \text{unemb}_v\|_1 \leq 2d$ from Item 4. Hence $\arg\max_{v \in \mathcal{V}} \langle \tilde{x}_{n-1}^{(L)}, \text{unemb}_v \rangle = v^*$, so $\tilde{T}(x) = T(x)$. As $x \in \mathcal{X}$ was arbitrary, this shows $\tilde{T}|_{\mathcal{X}} = T|_{\mathcal{X}}$.

The condition (5) is equivalent to

$$c^2 \geq 2\sqrt{d_k} \log\left(\frac{16}{6} \max\{4d, 20d_k\} N (6L_{\text{ff}})^L\right). \tag{6}$$

showing the statement. $\square$

### D.4.1. PROOF OF THEOREM 4.1

We obtain Theorem 4.1 from Theorem D.16 by choosing $N$ as a uniform upper bound on the length of any context that can occur in the corresponding hardmax construction.

*Proof of Theorem 4.1.* Fix one of the hardmax constructions $T$ from Proposition 3.1 and Theorems 3.2 and 3.3 for the corresponding bound parameter $\hat{l} \in \{\hat{n}, \hat{t}, \hat{s}\}$. Let $\mathcal{X}$ be the set of all contexts that can occur when running $T$ in the respective setting (including intermediate prefixes for CoT/SCoT, as in Lemma D.15). By Lemmas C.13 and C.21, we may choose a uniform length bound $N$ for $\mathcal{X}$, namely $N = \hat{n} + 1$ for Proposition 3.1, $N = 4 + 6\hat{t}$ for Theorem 3.2, and $N = 8(\hat{s} + 3)$ for Theorem 3.3; in particular $N = \mathcal{O}(\hat{l})$. By Lemma D.15, $T$ and $\mathcal{X}$ satisfy the assumptions of Theorem D.16.

Define $c_0(\hat{l})$ as the right-hand side of (3) for this choice of $N$. Then for all $c \geq c_0(\hat{l})$, Theorem D.16 yields that the corresponding $c$-scaled softmax transformer $\tilde{T}_c$ agrees with $T$ on all contexts in $\mathcal{X}$, hence generates the same tokens as $T$. Finally, using $d, d_k, d_{\text{ff}}, L = \mathcal{O}(\log \hat{l})$ and $N = \mathcal{O}(\hat{l})$ in (3) yields

$$c_0(\hat{l}) = \mathcal{O}\left((\log \hat{l})^{\frac{3}{4}} (\log \log \hat{l})^{\frac{1}{2}}\right),$$

as claimed. To ensure $c \in \mathbb{F}_{\text{act}} = \mathbb{F}_{b_m^{\text{act}}, b_e^{\text{act}}}$ it suffices that $c \leq (2 - 2^{-b_m^{\text{act}}})2^{E_{\max}}$ with $E_{\max} = 2^{b_e^{\text{act}}-1} - 1$ (Definition B.18). Equivalently, $b_e^{\text{act}} = \mathcal{O}(\log \log c) = \mathcal{O}(\log \log \log \hat{l})$ exponent bits suffice. $\square$

### D.5. Attention Weight Rounding

**Theorem D.17** (Approximating Hardmax transformer with Softmax transformer using Denoising). *Let* $T \in \mathcal{T}^{\mathrm{hard}}(d, d_{ff}, d_k, d_v, H, L, \mathcal{V})$ *be a hardmax transformer and let* $\mathcal{X} \subseteq \mathcal{V}^+$ *be a set of inputs of length at most* $N$ *such that the statements in Lemma D.15 hold for* $T$ *and* $\mathcal{X}$. *Let*

$$c \geq d_k^{\frac{1}{4}} (\log(96N))^{\frac{1}{2}}.$$

*Consider an* activation precision $\mathbb{F}_{act} = \mathbb{F}_{b_m^{act}, b_e^{act}}$ *and an* attention weight precision $\mathbb{F}_{att} = \mathbb{F}_{b_m^{att}, b_e^{att}}$ *satisfying*

$$b_m^{att} \geq 4, \qquad \frac{1}{N} \geq 2^{E_{\min}} \quad (\text{for } \mathbb{F}_{att}), \qquad b_m^{act} \geq 1, \qquad b_e^{act} \geq 3, \qquad c \in \mathbb{F}_{act}$$

*Then there exists a softmax transformer* $\tilde{T}$ *of depth* $2L$ *and feedforward dimension* $\max\{d_{ff}, 6d\}$ *whose weight matrices have entries in* $\{-c, -2, -1, 0, 1, 2, c\}$ *such that, when evaluating* $\tilde{T}$ *with rounding to* $\mathbb{F}_{act}$ *and* $\mathbb{F}_{att}$ *as in Definition D.4, it holds that* $\tilde{T}|_{\mathcal{X}} = T|_{\mathcal{X}}$.

The proof will work as follows. For each transformer layer in $T$ we add two transformer layers to $\tilde{T}$. The first one uses the attention to approximate the attention in the transformer layer of $T$ and uses the MLP to remove the errors by mapping all values back to $\{-1, 0, 1\}$, while the second one simply consists of an empty attention layer and the MLP from $T$.

The denoising uses the following MLP:

**Lemma D.18** (Denoising with MLP). *Let* $d \in \mathbb{N}$. *Then there exists an MLP* $f \in \mathcal{F}(d, 6d)$ *such that for inputs* $x \in \mathbb{R}^d$, *writing* $y = f(x) + x$, *it holds that*

$$y_i = \begin{cases} -1, & x_i \in [-5/4, -3/4], \\ 0, & x_i \in [-1/4, 1/4], \\ 1, & x_i \in [3/4, 5/4] \end{cases} \qquad . \tag{7}$$

*Furthermore, this still holds for inputs* $x \in \mathbb{F}_{b_m, b_e}^d$ *when rounding hidden activations, the output* $f(x)$ *and* $f(x) + x$ *to* $\mathbb{F}_{b_m, b_e}^d$ *whenever* $b_e \geq 3$ *and* $b_m \geq 1$ *(so that* $\pm\frac{1}{4}, \pm\frac{3}{4} \in \mathbb{F}_{b_m, b_e}$*).*[7]

*Proof.* We consider the case $d = 1$, the general case works by simply combining the MLPs for each dimension (as everything will be coordinate-wise). The following function is the output of a 6 hidden neuron MLP with weights in $\{0, \pm 1, \pm 2\}$ and biases in $\{0, \pm\frac{1}{4}, \pm\frac{3}{4}\}$ (denoting $z^+ = \max(0, z) = \mathrm{ReLU}(z)$):

$$f(x) = (-x)^+ - (x)^+ + 2(x - \frac{1}{4})^+ - 2(x - \frac{3}{4})^+ - 2(-x - \frac{1}{4})^+ + 2(-x - \frac{3}{4})^+ . \tag{8}$$

Adding this function to the residual identity yields the desired values on the relevant ranges:

- If $x \in [-\frac{1}{4}, \frac{1}{4}]$, then only the first two terms in (8) can be non-zero and one checks that $f(x) = -x$, hence $f(x) + x = 0$.

- If $x \in [\frac{3}{4}, \frac{5}{4}]$, then only the terms involving $x$, $x - \frac{1}{4}$ and $x - \frac{3}{4}$ in (8) are non-zero, so

$$f(x) = -x + 2\left(x - \frac{1}{4}\right) - 2\left(x - \frac{3}{4}\right) = 1 - x,$$

  and therefore $f(x) + x = 1$.

- If $x \in [-\frac{5}{4}, -\frac{3}{4}]$, the claim follows by symmetry (or by the analogous computation), yielding $f(x) + x = -1$.

Now consider an input $x \in \mathbb{F}_{b_m, b_e}$ with $b_e \geq 3$ and $b_m \geq 1$. We show that (7) still holds under rounding to $\mathbb{F}_{b_m, b_e}$ by simply checking that all 6 terms in (8), the output $f(x)$ and $x + f(x)$ are also in $\mathbb{F}_{b_m, b_e}$.

**Case** $x \in [-1/4, 1/4]$: In this case, only the first two terms in (8) have non-zero contribution. As $x, -x \in \mathbb{F}_{b_m, b_e}$, no rounding occurs and the output is as in the non-rounding case.

---

[7]One can also work with $b_e = 2$ and $b_m \geq 2$, in which case $\pm\frac{1}{4}, \pm\frac{3}{4}$ are subnormals.

**Case** $x \in [3/4, 5/4]$: In this case, only the terms involving $x$, $x - \frac{1}{4}$ and $x - \frac{3}{4}$ appear. Since $x \in \mathbb{F}_{b_m, b_e}$ and $\pm\frac{1}{4}, \pm\frac{3}{4} \in \mathbb{F}_{b_m, b_e}$, it suffices to show that $x - \frac{1}{4}, x - \frac{3}{4} \in \mathbb{F}_{b_m, b_e}$ as well, so no rounding occurs anywhere in the computation.

If $b_m = 1$, then $\mathbb{F}_{b_m, b_e} \cap [\frac{3}{4}, \frac{5}{4}] = \{\frac{3}{4}, 1\}$ and one checks directly that in both cases $x - \frac{1}{4}, x - \frac{3}{4} \in \mathbb{F}_{b_m, b_e}$. Otherwise assume $b_m \geq 2$ and write $x = (1 + t2^{-b_m})2^\kappa$ with $\kappa \in \{-1, 0\}$.

- If $\kappa = 0$, then $x = 1 + t2^{-b_m}$ with $t \leq 2^{b_m - 2}$. If $t = 2^{b_m - 2}$ (i.e. $x = \frac{5}{4}$), then $x - \frac{1}{4} = 1$ and $x - \frac{3}{4} = \frac{1}{2}$, both of which are exactly representable in $\mathbb{F}_{b_m, b_e}$. Otherwise $t \leq 2^{b_m - 2} - 1$ and

$$x - \frac{1}{4} = \frac{3}{4} + t2^{-b_m} = \left(1 + (2^{b_m - 1} + 2t)2^{-b_m}\right)2^{-1} \in \mathbb{F}_{b_m, b_e},$$

  and

$$x - \frac{3}{4} = \frac{1}{4} + t2^{-b_m} = \left(1 + (4t)2^{-b_m}\right)2^{-2} \in \mathbb{F}_{b_m, b_e}.$$

- If $\kappa = -1$, then $x \in [\frac{3}{4}, 1)$ and $x - \frac{1}{4} \in [\frac{1}{2}, \frac{3}{4})$. Writing $x = (1 + t2^{-b_m})2^{-1}$, we have

$$x - \frac{1}{4} = \left(1 + (t - 2^{b_m - 1})2^{-b_m}\right)2^{-1} \in \mathbb{F}_{b_m, b_e},$$

  and

$$x - \frac{3}{4} \in [0, \tfrac{1}{4})$$

  is of the form $s2^{-(b_m + 1)}$ for some integer $s \in \{0, \ldots, 2^{b_m - 1} - 1\}$. If $x - \frac{3}{4} < 2^{E_{\min}}$, then setting $t' := s2^{-(1 + E_{\min})}$ (an integer since $E_{\min} \leq -2$) yields

$$x - \frac{3}{4} = (t'2^{-b_m})2^{E_{\min}},$$

  i.e. a (possibly zero) subnormal. Otherwise $2^{E_{\min}} \leq x - \frac{3}{4} < \frac{1}{4}$, so $x - \frac{3}{4}$ is a normal number with exponent in $\{E_{\min}, \ldots, -2\}$.

Therefore, the computation of (8) is identical to the real-arithmetic computation, and we again obtain $f(x) + x = 1$. Note that in each of the above cases we have $f(x) \in \{-x, 1 - x, -1 - x\} \subset \mathbb{F}_{b_m, b_e}$ and $x + f(x) \in \{-1, 0, 1\} \subset \mathbb{F}_{b_m, b_e}$, so rounding $f(x)$ and $x + f(x)$ has no effect.

The case $x \in [-\frac{5}{4}, -\frac{3}{4}]$ is analogous by symmetry. $\qquad\square$

*Proof of Theorem D.17.* As in the proof of Theorem D.16 we replace $T$ by the equivalent $T_c$. As in Theorem D.16, $T_c$ has separation at least

$$\beta := \frac{c^2}{\sqrt{d_k}} .$$

We construct $\tilde{T}$ from $T_c$ by replacing each layer $\ell \in \{1, \ldots, L\}$ with two layers corresponding to indices $2\ell - 1, 2\ell$.

Layer $2\ell - 1$ of $\tilde{T}$ replaces the $c$-scaled hardmax multihead attention from layer $\ell$ of $T_c$ by the softmax variant with rounding to $\mathbb{F}_{\text{act}}$ and $\mathbb{F}_{\text{att}}$ as in Definition D.4, and uses the denoising MLP from Lemma D.18 (which uses $b_m^{\text{act}} \geq 1, b_e^{\text{act}} \geq 3$) to remove the error introduced by this replacement. Layer $2\ell$ of $\tilde{T}$ sets all attention projections to zero (hence the multihead attention in layer $2\ell$ of $\tilde{T}$ does nothing), and uses the original MLP of layer $\ell$ in $T_c$. Let $x \in \mathcal{X}$ be some input, let $x^{(\ell)}$ ($\ell \in \{0, \ldots, L\}$) $\tilde{x}^{(\ell)}$ ($\ell \in \{0, \ldots, 2L\}$) be the hidden representations of $T_c$ and $\tilde{T}$ on $x$. Then we show that $T_c(x) = \tilde{T}(x)$ by showing that $x^{(\ell)} = \tilde{x}^{(2\ell)}$ for all $\ell \in \{0, \ldots, L\}$ by induction. For $\ell = 0$ this holds as $T_c$ and $\tilde{T}$ use the same embeddings, which are ternary (Item 4) and hence no rounding error occurs.

Assume $x^{(\ell)} = \tilde{x}^{(2\ell)}$ for some $\ell \in \{0, \ldots, L - 1\}$. We will show that the error introduced by replacing the hardmax with $c$-scaled softmax multihead attention is bounded as

$$\|x^{(\ell+0.5)} - \tilde{x}^{(2\ell+0.5)}\|_\infty \leq \frac{1}{4} . \tag{9}$$

Then Lemma D.18 removes this error to obtain

$$\tilde{x}^{(2\ell+1)} = x^{(\ell+0.5)} .$$

Since the attention in layer $2\ell + 2$ of $\tilde{T}$ does not change the representations, we have

$$\tilde{x}^{(2\ell+1.5)} = \tilde{x}^{(2\ell+1)} = x^{(\ell+0.5)} .$$

As all activations in the MLP in layer $\ell + 1$ of $T_c$ are ternary, rounding has no effect there and hence applying the original MLP yields

$$\tilde{x}^{(2\ell+2)} = x^{(\ell+1)} .$$

By construction, all weight matrices of $\tilde{T}$ have entries in $\{-c, -2, -1, 0, 1, 2, c\}$, where $\pm c$ occur in the query- and key-projections and $\pm 2$ in the denoising MLPs.

It thus remains to show that the assumptions of the theorem ensure (9). We apply Definition D.4 and the statements after it to layer $\ell + 1$ of $T_c$ and $2\ell + 1$ of $\tilde{T}$. Using that the input error is zero (by the induction hypothesis) and that inputs are ternary, Item 1 ensures that queries, keys and values of $T_c$ have entries in $\{0, \pm 1, \pm c\} \subset \mathbb{F}_{\text{act}}$, hence rounding has no effect on them and we obtain $\Delta_q = \Delta_k = \Delta_v = 0$ and hence also $\Delta_s = 0$. We apply Lemma D.8 to obtain

$$\bar{\Delta}_o \leq (1 + \delta_\alpha^{\text{round}})\Delta_v + \left(2ne^{-(\beta-2\Delta_s)} + \delta_\alpha^{\text{round}}\right)L_V M = 2ne^{-\beta} + \delta_\alpha^{\text{round}}$$

where we also used $L_V \leq 1$ and $M \leq 1$. We bound $\delta_\alpha^{\text{round}}$ using Lemma D.14 to obtain

$$\bar{\Delta}_o \leq 2ne^{-\beta} + 2^{-b_m^{\text{att}}-1} + ne^{-\beta}$$

and hence

$$\Delta_o \leq 2\bar{\Delta}_o \leq 6ne^{-\beta} + 2^{-b_m^{\text{att}}}$$

using Lemma D.12. As in the proof of Theorem D.16, Items 1 and 2 ensure that rounding after the output projections and summing has no effect, hence

$$\Delta_y = \bar{\Delta}_y \leq L_O\Delta_o = \Delta_o$$

. Adding to the residual stream can again at most double the rounding error, hence we obtain

$$\max_i \|x^{(\ell+0.5)} - \tilde{x}^{(2\ell+0.5)}\|_\infty \leq 12ne^{-\beta} + 2^{-b_m^{\text{att}}+1}$$

This term is $\leq \frac{1}{4}$ if both summands are $\leq \frac{1}{8}$. For the first summand this is precisely the condition

$$c^2 \geq \log(96N)\sqrt{d_k}$$

while for the second one it is the condition $b_m^{\text{att}} \geq 4$ . $\qquad\square$

### D.5.1. PROOF OF THEOREM 4.2

We obtain Theorem 4.2 from Theorem D.17 by choosing $N$ as a uniform upper bound on the length of any context that can occur in the corresponding hardmax construction.

*Proof of Theorem 4.2.* Fix one of the hardmax constructions $T$ from Proposition 3.1 and Theorems 3.2 and 3.3 for the corresponding bound parameter $\hat{l} \in \{\hat{n}, \hat{t}, \hat{s}\}$, and let $\mathcal{X}$ be the set of all contexts that can occur when running $T$ in the respective setting. As in the proof of Theorem 4.1, we may choose a uniform length bound $N$ for $\mathcal{X}$, namely $N = \hat{n} + 1$ for Proposition 3.1, $N = 4 + 6\hat{t}$ for Theorem 3.2, and $N = 8(\hat{s} + 3)$ for Theorem 3.3; in particular $N = \mathcal{O}(\hat{l})$. By Lemma D.15, $T$ and $\mathcal{X}$ satisfy the assumptions of Theorem D.17.

Define $c_0(\hat{l}) := d_k^{\frac{1}{4}}(\log(96N))^{\frac{1}{2}}$. Then for all $c \geq c_0(\hat{l})$, Theorem D.17 yields a depth-$2L$ softmax transformer $\tilde{T}_c$ with $\{0, \pm 1, \pm 2, \pm c\}$-valued weights that agrees with $T$ on all contexts in $\mathcal{X}$ when evaluated with rounding to $\mathbb{F}_{\text{act}}$ and $\mathbb{F}_{\text{att}}$ as in Definition D.4, and hence generates the same tokens as $T$. Since $d_k = \mathcal{O}(\log \hat{l})$ and $\log(96N) = \mathcal{O}(\log \hat{l})$, we have

$$c_0(\hat{l}) = d_k^{\frac{1}{4}}(\log(96N))^{\frac{1}{2}} = \mathcal{O}\big((\log \hat{l})^{\frac{3}{4}}\big).$$

Moreover, the attention-weight condition $\frac{1}{N} \geq 2^{E_{\min}}$ from Theorem D.17 holds whenever $b_e^{\text{att}} \geq 1 + \log_2(\log_2 N + 2) = \mathcal{O}(\log \log \hat{l})$, since $E_{\min} = 2 - 2^{b_e^{\text{att}}-1}$ for $\mathbb{F}_{\text{att}}$ (Definition B.18) and $N = \mathcal{O}(\hat{l})$. To ensure $c \in \mathbb{F}_{\text{act}} = \mathbb{F}_{b_m^{\text{act}}, b_e^{\text{act}}}$ it suffices that $c \leq (2 - 2^{-b_m^{\text{act}}})2^{E_{\max}}$ with $E_{\max} = 2^{b_e^{\text{act}}-1} - 1$ (Definition B.18), i.e. $b_e^{\text{act}} = \mathcal{O}(\log \log c) = \mathcal{O}(\log \log \log \hat{l})$. $\qquad\square$

```
<inp> (1,2) 9 (1,3) 4 (1,8) 7 (1,9) 1 ... </inp> (7,3) 6 (9,3) 9 (8,3) 2 (3,3) 3 (4,2)
4 6 (5,2) 1 6 (5,1) 3 6 (4,1) 6 9 (6,1) 9 (5,5) 2 6 (1,5) 3 6 (4,5) 7 (7,5) 1 (1,6) 6
8 (2,5) 9 (3,5) <none> (1,6) 8 (1,1) 2 5 (1,4) 6 (1,7) 5 (2,5) 9 (3,5) <none> (1,1) 5
(1,7) 2 (1,4) 6 (2,5) 9 (3,5) <none> (1,5) 6 (1,6) 3 8 ... <outp> (1,1) 6 (1,2) 9 (1,3)
4 (1,4) 2 ... </outp>
```

*Figure 7.* Example excerpt from a deterministic solver trace in our tokenization. Tokens are separated by spaces, with coordinate pairs and digits as separate tokens.

| $N$ | # puzzles | CoT tokens | # SCoT segments | SCoT tokens |
|---|---|---|---|---|
| $2^{10}$ | 1,003,062 | 549,886,480 | 1,216,586 | 597,636,422 |
| $2^{11}$ | 1,366,621 | 1,081,910,259 | 2,275,332 | 1,278,330,673 |
| $2^{12}$ | 1,744,525 | 2,215,629,831 | 4,514,399 | 2,812,386,367 |
| $2^{13}$ | 2,288,481 | 5,501,694,458 | 10,966,828 | 7,382,325,156 |
| $2^{14}$ | 2,978,536 | 13,676,729,210 | 26,968,686 | 18,916,908,506 |

*Table 3.* Training set sizes after filtering by solver CoT length $\leq N$.

# E. Additional Material for Section 5

## E.1. Datasets and Tokenization

We use the sudoku-extreme dataset from (Wang et al., 2025a) throughout. It contains 3,831,994 training puzzles and 422,786 test puzzles. Each puzzle is a standard $9 \times 9$ Sudoku with digits in $\{1, \ldots, 9\}$ and blanks.

**Solver** We generate traces with a deterministic depth-first backtracking solver. At each step the solver selects an unfilled cell with minimum remaining values (MRV), where remaining values are the digits consistent with the current partial assignment. Ties are broken lexicographically by the cell coordinates, which is equivalent to row-major order. The solver then tries the remaining values in increasing order and recurses. If a contradiction is reached, it backtracks to the most recent decision point that still has a remaining value to try.

**CoT Tokenization** Each puzzle yields one token sequence of the form

$$\texttt{<inp>} \text{ (givens) } \texttt{</inp>} \text{ (solver trace) } \texttt{<outp>} \text{ (solution) } \texttt{</outp>} .$$

The vocabulary contains 8 special tokens `<pad>`, `<inp>`, `</inp>`, `<summ>`, `</summ>`, `<outp>`, `</outp>`, and `<none>`, as well as 81 coordinate tokens $(r, c)$ for $r, c \in \{1, \ldots, 9\}$ and 9 digit tokens $1, \ldots, 9$, for a total vocabulary size of 98. The input block lists all givens as coordinate-value pairs. The trace is a concatenation of decision lines, where each line is a coordinate token followed by the remaining candidate digits for that cell. If the solver reaches a contradiction, it emits an empty candidate list, encoded by the token `<none>`. The output block lists the full solved grid as 81 coordinate-value pairs in row-major order. An example excerpt is shown in Figure 7.

We define the solver CoT length of a puzzle as the token length of this full sequence. We use it as a difficulty measure for this specific solver. For the CoT experiments we train only on puzzles with solver CoT length at most $N$ for $N \in \{2^{10}, 2^{11}, 2^{12}, 2^{13}, 2^{14}\}$. Dataset statistics for these cutoffs are given in Table 3.

**SCoT Tokenization** For SCoT we split the full CoT token sequence into segments and insert summary blocks. We insert a summary after approximately 512 non-summary trace tokens. Each summary has the form `<summ>` $\cdots$ `</summ>` and encodes the solver state as the concatenation of the givens and the current depth-first decision stack. The stack is represented by the most recent trace line for each cell on the active search path, using the same coordinate and candidate-list format as in the trace.

For a given cutoff $N$, the CoT training set contains all puzzles with solver CoT length at most $N$, while the SCoT training set contains all segments generated from these puzzles. Table 3 summarizes the resulting training set sizes. SCoT contains more total tokens because each segment repeats a summary of the current solver state.

| $d$ | $H$ | $L$ | parameters |
|---|---|---|---|
| 512 | 8 | 4 | 12,710,912 |
| 512 | 8 | 5 | 15,863,296 |
| 512 | 8 | 6 | 19,015,680 |
| 512 | 8 | 7 | 22,168,064 |
| 512 | 8 | 8 | 25,320,448 |
| 768 | 12 | 12 | 85,206,528 |

*Table 4.* Model sizes used in the Sudoku experiments.

### E.2. Models and Training

We use standard decoder-only transformers with RoPE positional encodings. All models use $d_{\text{ff}} = 4d$, no dropout, and a vocabulary of size 98. Table 4 summarizes the model sizes used in Section 5.

Training uses AdamW with weight decay 0.01 and a peak learning rate of $5 \cdot 10^{-4}$ with 2% of updates for warmup and cosine decay to $10^{-5}$. Each model is trained for $2 \cdot 10^{10}$ non-padding tokens. Batches are chosen to target roughly $10^5$ non-padding tokens per optimizer update, which corresponds to about $2 \cdot 10^5$ update steps for all runs. Since the trace format is deterministic given the input puzzle, training loss typically converges close to 0.

### E.3. Evaluation

**Generation** All evaluations use greedy decoding (temperature 0). For CoT, sequence lengths vary widely, which makes efficient batched generation difficult. We therefore evaluate CoT models using vLLM. For SCoT, the segments have similar lengths and we generate segment-wise with large batch sizes.

**Accuracy Estimates** We evaluate only whether the final output block matches the correct solution, which is unique for these puzzles. For each model, we estimate test accuracy on a random subsample of 10,000 test puzzles. This subsample is fixed across models by using a fixed random seed.

To estimate accuracy as a function of solver CoT length, we use target-based sampling. For a target length $n$, we select the 1000 test puzzles whose solver CoT length is closest to $n$ and compute accuracy on this set. The test set is large enough that these sampled lengths concentrate tightly around the target. For example, for $n = 2048$ the selected puzzles have solver CoT length between 2028 and 2068.

For the hardest 100 subset, we take the 100 test puzzles with largest solver CoT length, which ranges from 1,088,320 to 9,477,350 tokens in our test set. For CoT evaluation we cap generation at 32,512 new tokens for computational feasibility. If a CoT model does not solve the puzzle within this budget, it typically does not recover with longer generation, as generations become incoherent significantly beyond the context lengths seen in training.

For SCoT evaluation we cap the total generation length at 5 million new tokens for the standard test subset and at 15 million new tokens for the hardest 100 subset, across all segments. This corresponds to more than 10,000 generated segments for the hardest puzzles.

### E.4. Additional experiments

While we do not repeat all training runs multiple times, we feel that it is important to make sure that the SCoT model performing well at depth $L = 6$ and the CoT model failing at $N = 2^{14}$ for the same depth were not just by chance. We hence repeat the SCoT run 2 more times with different random seeds, and additionally compare the $L = 6$ CoT failure at $N = 2^{14}$ to several interventions.

**Success of SCoT at $L = 6$.** The SCoT models in Section 5.2 were trained on the data obtained from Sudokus with CoT length at most $N = 2^{14}$. The accuracies on the 10k-subsample of the test set for the original and 2 additional runs are as follows, with the first run being the one shown in the main part:

| Run | 1 | 2 | 3 |
|---|---|---|---|
| Acc(%) | 99.48 | 99.70 | 99.38 |

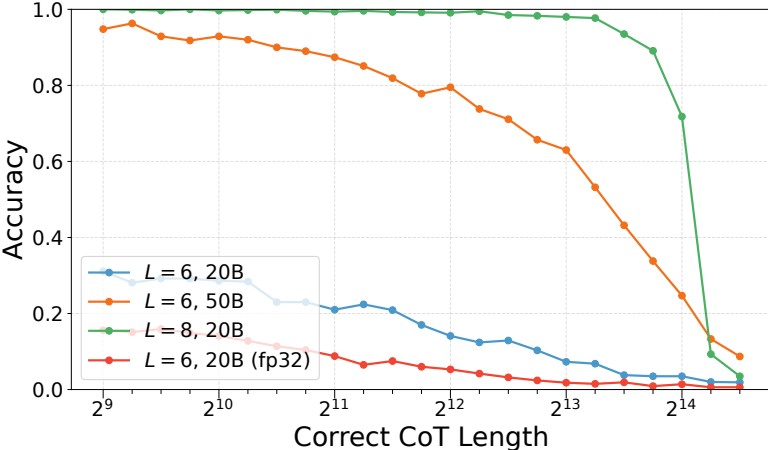

*Figure 8.* Accuracy vs. correct CoT length at $N = 2^{14}$ for the original $L = 6$ bfloat16 run, the same run with 50 billion training tokens, an $L = 8$ bfloat16 run with the standard 20 billion training tokens, and an $L = 6$ fp32 run with 20 billion training tokens.

| Training $N$ | Test Acc (%) | Hardest 100 Acc (%) |
|---|---|---|
| $2^{11}$ | 99.34 | 69 |
| $2^{12}$ | 99.96 | 92 |
| $2^{13}$ | 99.97 | 91 |
| $2^{14}$ | 99.95 | 94 |

*Table 5.* Performance of the larger SCoT models on random Sudokus and the 100 hardest ones from the test set.

While there is some variance in the performance, all runs perform well and it appears safe to conclude that the performance of SCoT at this model size was not an outlier.

**Interventions on the $L = 6$ failure at $N = 2^{14}$.** Figure 8 compares the $L = 6$ CoT failure at $N = 2^{14}$ to three interventions: training the same $L = 6$ model for 50 billion tokens instead of 20 billion, increasing the depth to $L = 8$ at the standard 20 billion tokens, and retraining $L = 6$ with full fp32 activations (TF32 disabled). Longer training substantially improves accuracy at short context lengths but degrades sharply for longer ones, suggesting only a partial mitigation of the failure. Switching to fp32 activations makes essentially no difference. In contrast, $L = 8$ at the same 20 billion training tokens achieves near-perfect accuracy up to the training context boundary, supporting our conclusion that increasing model size is the more effective lever. Nonetheless we recognize that longer runs with different hyperparameters are required to actually conclude that depth 6 is insufficient to learn this algorithm for $N = 2^{14}$.

### E.5. Constant-Space Generalization

A natural further question is whether the segment-length-only scaling of Theorem 3.3 transfers to learnability: once the summary update has been learned, can the model be rolled out to much longer reasoning chains than seen in training? We test this with a larger SCoT model ($d = 768, L = 12, H = 12$) trained on all segments obtained from Sudokus with CoT length at most $N \in \{2^{11}, 2^{12}, 2^{13}, 2^{14}\}$. All 4 models are evaluated on a random subsample of 10000 test Sudokus to see the general performance of SCoT on Sudoku, and again on the 100 hardest test puzzles (by solver CoT length) to test the generalization capabilities. These hard Sudokus have solver CoT lengths between 1 and 9 million tokens, i.e., for $N = 2^{12}$ longer by a factor of at least 250 than the ones seen in training. The results are shown in Table 5. With this algorithm, Sudokus of different total reasoning length require similar space and can hence use segments of similar length, which is what allows this *constant-space generalization*.

# F. Additional Considerations

## F.1. Concrete Instantiation

Here we instantiate the explicit constants from our results at realistic model sizes and precisions (as announced in the introduction), and also see how some prior constructions fail at moderate context size when using standard precision assumptions. The main goal here is to argue that the logarithmic growth of width and depth is much more compatible with transformers used in practice than the logarithmic precision growth in prior constructions. In particular, we analyze the capability predicted by the results for a GPT-3 sized model, i.e. $L = 96, H = 96, d = 12288, d_h = 128, d_{\text{ff}} = 4d$ (Brown et al., 2020). It should be noted that these are loose bounds that could be improved by compressing our constructions: in particular, in the hardmax constructions, one could reduce depth with more heads per layer allowing for better parallelism, while for the denoising MLP technique, the denoising could often be scheduled without doubling a layer (e.g. if the output of an attention layer is not used in the subsequent MLP).

### F.1.1. Hardmax Constructions

First, we consider the hardmax constructions from Theorems C.14 and C.22. As $L = 8 + \frac{5}{2}r$ in both constructions, the largest $r$ fitting depth 96 is $r = 34$. Similarly, as $d_k = 4r - 1$ in both constructions, the head dimension of 128 yields a very similar restriction bounding $r$ to 32. Hence, a sufficiently small Turing machine (small enough for $d$ and $d_{\text{ff}}$ to suffice as discussed shortly) can be simulated as long as the generated token sequence or segments have at most $2^{32}$ tokens; by Lemmas C.13 and C.21 this allows choosing $\hat{t}$ or $\hat{s}$ at roughly half a billion.

While the model dimension $d = 12288$ allows the simulated Turing machine to be quite large (see (2)), the feedforward dimension quickly becomes the bottleneck as it grows with $|Q||\Gamma|^K$ (see (1)). For example, a machine with $K = 3$ tapes, $|\Gamma| = 10$ tape symbols, and $|Q| = 49$ states barely fits the feedforward dimension. However, the growth of the feedforward dimension in the constructions from Theorems 3.2 and 3.3 comes from a single MLP which uses one neuron for each transition of the Turing machine to compute its transition function $\delta$. It suffices to allocate such a neuron only for each transition that can actually occur when running the Turing machine on admissible inputs. Moreover, since among these transition-neurons exactly one fires for any admissible configuration, the transition MLP effectively acts as a lookup table. For very large Turing machines with many transitions (loosely corresponding to stored information), a sparsely-routed Mixture-of-Experts layer (Shazeer et al., 2017; Fedus et al., 2022) could exploit this by routing (e.g. on the current state $q$) to an expert that implements only the relevant part of $\delta$, reducing the compute per token while allowing more total transition parameters.

### F.1.2. Softmax and Rounding

Next, we consider the softmax results from Appendix D and the standard bfloat16 floating point format, which uses $b_m = 7$ mantissa and $b_e = 8$ exponent bits. Without attention weight rounding (Theorem D.16), the only condition on the activation precision is that a sufficiently large $c$ is representable in the activation precision. Plugging the GPT-3 model sizes into the upper bound on the required magnitude of $c$ in (3) shows that bfloat16 easily suffices.

With attention weight rounding (Theorem D.17), the depth for a given $r$ doubles and hence depth 96 only suffices for $r = 16$, hence context sizes up to 65536 and $\hat{t}$ or $\hat{s}$ at least around 8000. This theorem also assumes that $\frac{1}{N}$ is in the normal range of the attention precision in order to control the error when rounding attention weights. For bfloat16, the smallest positive normal number is $2^{-126}$ and hence context lengths up to $N = 2^{126} > 10^{38}$ can be handled before rounding of attention weights becomes a problem. Hence, it is fair to say that bfloat16 suffices as activation and attention weight precision for all practical context sizes in these constructions.

Lastly, for this example where $d_k = 128$ and $N = 2^{16} = 65536$, the bound from Theorem D.17 yields $c \geq d_k^{\frac{1}{4}}\sqrt{\log(96N)} \approx 13.3$.

### F.1.3. Precision Bottlenecks in Prior Constructions

In contrast, prior constructions (Pérez et al., 2021; Merrill & Sabharwal, 2024; Yang et al., 2025) use fixed (small) depth and width, so precision quickly becomes the bottleneck when doing similar considerations for them. Considering (Merrill & Sabharwal, 2024) in particular, they use vectors $\phi_i = \frac{(i,1,-i,-1)}{\|(i,1,-i,-1)\|_2}$ to encode head and symbol positions and extract tape symbols in place of our binary representations. Their construction fundamentally relies on the fact that $\langle \phi_i, \phi_j \rangle$ is

maximal for $i = j$. This holds true at infinite precision (see their Lemma 2), yielding an elegant fixed-width position encoding which is used for keys and queries. However, this is not generally the case anymore when rounding $\phi_i$ to some finite precision. In particular, using the rounding notation from Definition B.18 with the standard precision formats $\text{bf16} = \mathbb{F}_{7,8}, \text{fp16} = \mathbb{F}_{10,5}, \text{fp32} = \mathbb{F}_{23,8}, \text{fp64} = \mathbb{F}_{52,11}$ (IEEE, 2019), it holds that

$$\langle [\phi_7]_{\text{bf16}}, [\phi_6]_{\text{bf16}} \rangle > \langle [\phi_7]_{\text{bf16}}, [\phi_7]_{\text{bf16}} \rangle.$$

Hence, when rounding activations to bf16, the symbol retrieval for head position 7 incorrectly extracts the symbol at position 6 instead of 7, potentially leading to errors when running the transformer with such activation rounding. The same problem appears for fp16, fp32 and fp64 at head position 8 (confused with 7), 61 (with 60) and 7875 (with 7874) respectively.

## F.2. Rotary Positional Encodings (RoPE)

Our constructions use binary positional encodings, which are not a standard positional encoding. They could be viewed as either a loose discrete analogue to absolute sinusoidal positional embeddings (as each bit of the binary representation of $n$ is periodic, with frequencies decreasing exponentially for higher bits) or simply as an instantiation of learned absolute positional embeddings. However, absolute positional embeddings are rarely used anymore in modern LLMs and raise the number of parameters to be at least linear in the context length, while the most used positional encodings appear to be relative *rotary positional encodings* (RoPE) (Su et al., 2024).

In this section we argue that the statements in the introduction still hold when using RoPE as positional encodings, albeit at the cost of requiring $\mathcal{O}(\log \log N)$ *mantissa* bits in the activation precision. In particular, we show that RoPE can be used to extract the binary positional encodings. This proceeds by first constructing a stack of hardmax transformer layers with RoPE which extract the binary positional encodings (and could hence be added in front of the other constructions) and have good score separation, and then arguing how a slight extension of the denoising conversion argument from Theorem D.17 could be used to convert to softmax attention. Note that this is likely not what RoPE does in real transformers and many parts of our constructions (like attending $k$ positions back) can be implemented directly with RoPE without resorting to binary arithmetic, but this path is simpler than adapting all constructions to use RoPE directly.

### F.2.1. DEFINING RoPE

We define RoPE attention heads, transformer layers, stacks and transformers by rotating the components of queries and keys at different frequencies before taking their inner products, while removing the positional embeddings added to the token embeddings.

**Definition F.1** (Rotary Positional Encodings (RoPE)). Consider a single attention head (hardmax or softmax) $A \in \mathcal{A}^{\text{hard}}(d, d_k, d_v)$ or $A \in \mathcal{A}^{\text{soft}}(d, d_k, d_v)$ parameterized by $W_Q, W_K, W_V, W_O$. Let $\omega \in (0, \infty)^l$ for some $l \in \mathbb{N}$ where $2l \leq d_k$. The $\omega_i$ are referred to as *frequencies* and $2l$ as the number of rotated dimensions of each head. We define the map $\text{rot}_\omega : \mathbb{R}^{d_k} \times \mathbb{N}_0 \to \mathbb{R}^{d_k}$ as follows:

$$\text{rot}_\omega(x, n) = \begin{pmatrix} \cos(n\omega_1)x_1 + \sin(n\omega_1)x_2 \\ -\sin(n\omega_1)x_1 + \cos(n\omega_1)x_2 \\ \cos(n\omega_2)x_3 + \sin(n\omega_2)x_4 \\ -\sin(n\omega_2)x_3 + \cos(n\omega_2)x_4 \\ \cdots \\ \cos(n\omega_l)x_{2l-1} + \sin(n\omega_l)x_{2l} \\ -\sin(n\omega_l)x_{2l-1} + \cos(n\omega_l)x_{2l} \\ x_{2l+1} \\ \cdots \\ x_{d_k} \end{pmatrix}$$

In words, $\text{rot}_\omega$ rotates the first two entries of $x$ by $n\omega_1$, the next two by $n\omega_2$ and so on. The rotary attention head $A_{\text{RoPE}(\omega)}$ is then defined exactly as the standard attention head (Definition B.10), except that the queries and keys are rotated before computing the scores $s_{ij}$:

$$s_{ij} = \frac{1}{\sqrt{d_k}} \langle \text{rot}_\omega(W_Q x_i, i), \text{rot}_\omega(W_K x_j, j) \rangle$$

Transformer layers and stacks are analogously defined, where one can apply the same or different frequencies to different

attention heads. Transformers using RoPE are then defined without the added positional embeddings, but with RoPE in (some of) the attention heads.

### F.2.2. EXTRACTING BINARY POSITIONAL ENCODINGS WITH ROPE AND HARDMAX

The following lemma says that RoPE can be used to extract the binary positional encodings. Note that the rotation frequencies decline exponentially, which is how they are typically used in practice (Su et al., 2024).

**Lemma F.2.** *Let $r \in \mathbb{N}$ and $n_{\max} := 2^r$. Then there exists a hardmax RoPE transformer stack*

$$TS \in \mathcal{TS}^{\text{hard}}(d, d_{ff}, d_k, d_v, H, L)$$

*with $d = 2r + 3$, $d_k = 2(r + 2) + 2$, $d_v = 1$, $H = 2$, $L = r + 1$, some $d_{ff} = \mathcal{O}(1)$, and angular frequencies*

$$\omega = \left( \frac{2\pi}{2}, \frac{2\pi}{4}, \dots, \frac{2\pi}{2^{r+2}} \right)$$

*such that for every $n \leq n_{\max}$ and every input $x = (x_0, \dots, x_{n-1}) \in (\mathbb{R}^d)^n$ with*

$$x_i[I_{const}] = 1, \qquad x_i[F_{firstpos}] = \mathbf{1}_{i=0},$$

*and all other coordinates zero, it holds for all $i \in \{0, \dots, n-1\}$ that*

$$TS(x)_i[I_{res}] = \text{bin}_r(i) .$$

*Moreover, every attention head used below has score separation at least $\frac{1}{2\sqrt{d_k}}$ for the treated inputs.*

*Proof.* We use the register/flag notation from Appendix C.1. The representation space is split into:

- a 1-dimensional register $I_{\text{const}}$ carrying the constant value 1,

- a flag $F_{\text{firstpos}}$ marking the first token,

- an $r$-dimensional output register $I_{\text{res}}$ (initialized to 0),

- flags $F_{\text{mod}}^1, \dots, F_{\text{mod}}^r$ (initialized to 0),

- and a scratch flag $F_{\text{ex}}$ (initialized to 0).

For the key-query space of each head, we use a 2-dimensional *unrotated* block $I_{\text{unrot}}$ and, for each $s \in \{1, \dots, r + 2\}$, a 2-dimensional *rotated* block $I_{\text{rot}}^s$ which is rotated with angular frequency $\omega_s = \frac{2\pi}{2^s}$. Recall that the attention scores are obtained by scaling the dot products $\langle \text{rot}_\omega(q_i, i), \text{rot}_\omega(k_j, j) \rangle$ by $\frac{1}{\sqrt{d_k}}$ (see Definition B.10 and the definition above). Since hardmax is invariant under scaling by a positive constant, it suffices to compare these dot products, and the score separations follow by dividing by $\sqrt{d_k}$.

**Cosine dot-product contribution.** We will repeatedly use the following observation: if, for some rotated block $I_{\text{rot}}^s$ with frequency $\omega_s$, we choose

$$q_i[I_{\text{rot}}^s] = (1, 0), \qquad k_j[I_{\text{rot}}^s] = (1, 0),$$

then after applying RoPE the contribution of this block to the dot product is

$$\langle \text{rot}_{\omega_s}(q_i[I_{\text{rot}}^s], i), \text{rot}_{\omega_s}(k_j[I_{\text{rot}}^s], j) \rangle = \cos(\omega_s(i - j)).$$

If we instead choose $q_i[I_{\text{rot}}^s] = (b, 0)$ and $k_j[I_{\text{rot}}^s] = (c, 0)$ with $b, c \in \{0, 1\}$, then this contribution is $bc \cos(\omega_s(i - j))$. In particular, for $\omega_1 = \pi$ we have $\cos(\pi(i - j)) \in \{1, -1\}$ depending on whether $i - j$ is even or odd.

**Marking positions divisible by** $2, 4, 8, \ldots$. To achieve the goal, we first introduce flags $F_{\text{mod}}^k$ for $k = 1, \ldots, r$, initialized to zero. $F_{\text{mod}}^k$ will be set to 1 for positions $j$ which satisfy $j \equiv 0 \mod 2^k$. This is done as follows. First, a head in layer 1 attends with queries $q_i[I_{\text{rot}}^1] = (1, 0), q_i[I_{\text{unrot}}] = (1, 0) = (x_i[I_{\text{const}}], 0)$ and the rest is filled with zeros, which is from now on implicitly meant, and keys $k_j[I_{\text{rot}}^1] = (1, 0), k_j[I_{\text{unrot}}] = (x_j[F_{\text{firstpos}}], 0)$. Using the cosine dot-product contribution above, this head has dot products

$$\langle \text{rot}_\omega(q_i, i), \text{rot}_\omega(k_j, j) \rangle = \cos(\pi(i - j)) + x_j[F_{\text{firstpos}}].$$

For even query positions $i$, the dot product to $j = 0$ equals 2, every $j > 0$ with even offset has dot product 1, and every $j$ with odd offset has dot product $-1$. Hence $j = 0$ is the unique maximizer with dot-product gap at least 1. For odd query positions $i$, the dot product to $j = 0$ equals 0, while every odd $j \in \{1, \ldots, i\}$ has dot product 1. Hence the maximizing indices are exactly the odd positions $j \leq i$; importantly, all these have $x_j[F_{\text{firstpos}}] = 0$. We set the values to $v_j := x_j[F_{\text{firstpos}}]$ and write the attention output to the flag $F_{\text{mod}}^1$. Therefore, $F_{\text{mod}}^1(i) = 1$ for even $i$ and $F_{\text{mod}}^1(i) = 0$ for odd $i$.

For $k = 2, \ldots, r$, we proceed analogously, but gate the query by the previously computed flag $F_{\text{mod}}^{k-1}$: in layer $k$ we add a head with frequency $\omega_k$ which uses

$$q_i[I_{\text{rot}}^k] = (x_i[F_{\text{mod}}^{k-1}], 0), \qquad k_j[I_{\text{rot}}^k] = (x_j[F_{\text{firstpos}}], 0),$$

together with unrotated queries and keys

$$q_i[I_{\text{unrot}}] = (x_i[F_{\text{mod}}^{k-1}], 1), \qquad k_j[I_{\text{unrot}}] = (x_j[F_{\text{firstpos}}], 1 - x_j[F_{\text{firstpos}}]),$$

and values $v_j := x_j[F_{\text{firstpos}}]$, writing the output to $F_{\text{mod}}^k$. The resulting dot products are 1 for all $j \in \{1, \ldots, i\}$ and

$$\langle \text{rot}_\omega(q_i, i), \text{rot}_\omega(k_0, 0) \rangle = x_i[F_{\text{mod}}^{k-1}]\left(1 + \cos(\omega_k i)\right).$$

If $x_i[F_{\text{mod}}^{k-1}] = 0$, then the dot product to $j = 0$ is $0 < 1$, so $i$ extracts 0. If $x_i[F_{\text{mod}}^{k-1}] = 1$, then $i$ is divisible by $2^{k-1}$, hence $\cos(\omega_k i) \in \{1, -1\}$ and the dot product to $j = 0$ lies in $\{2, 0\}$, with value 2 if and only if $i$ is divisible by $2^k$. Therefore, $i$ attends to $j = 0$ if and only if $i$ is divisible by $2^k$, extracting 1 in this case and 0 otherwise (with dot-product gap at least 1 in all cases).

**Acquiring binary positional encodings.** Next, we need to compute the bits of $\text{bin}_r(i)$ at each position $i$. Note that the $k$'th bit of the binary representation (corresponding to $2^k$) is 1 for positions that are $2^k, 2^k + 1, \ldots, 2^{k+1} - 1$ modulo $2^{k+1}$ and $-1$ (i.e. 0 in the standard binary representation, $-1$ in our $\pm 1$ encoding) for positions that are $0, \ldots, 2^k - 1$ modulo $2^{k+1}$. In the latter case, the closest position divisible by $2^k$ is hence also divisible by $2^{k+1}$, while in the first case it is not. Hence, for each $k$ we add an attention head which attends to the closest position where $F_{\text{mod}}^k$ is set and extracts $F_{\text{mod}}^{k+1}$. The subsequent MLP can then set the $k$'th bit of $I_{\text{res}}$ to 1 if the extracted value is 0 and to $-1$ otherwise. This can be done in layer $k + 2$, since $F_{\text{mod}}^k$ and $F_{\text{mod}}^{k+1}$ are computed in layers $k$ and $k + 1$. The difficult part here is to attend, for each position $i$, to the closest position $j$ which is divisible by $2^k$ with constant separation. This is done as follows.

For $k = 0$, the least significant bit is directly determined by $F_{\text{mod}}^1$: since $\text{bin}_r^0(i) = -1$ on even $i$ and 1 on odd $i$, the MLP in layer 2 writes $I_{\text{res}}[0] := 1 - 2F_{\text{mod}}^1$.

Now fix $k \in \{1, \ldots, r - 1\}$. We describe the head in layer $k + 2$ that finds the latest index divisible by $2^k$ and extracts $F_{\text{mod}}^{k+1}$ from it. We use the following pre-rotation queries and keys:

$$q_i[I_{\text{unrot}}] = (1, 1), \qquad q_i[I_{\text{rot}}^s] = (1, 0) \text{ for } s \in \{k + 2, \ldots, r + 2\},$$

and

$$k_j[I_{\text{unrot}}] = (x_j[F_{\text{mod}}^k], x_j[F_{\text{mod}}^k]), \qquad k_j[I_{\text{rot}}^s] = (1, 0) \text{ for } s \in \{k + 2, \ldots, r + 2\}.$$

We set the value to $v_j := x_j[F_{\text{mod}}^{k+1}]$ and write the attention output to the scratch flag $F_{\text{ex}}$.

We claim that, for every query position $i$, this head attends to

$$j_0 := \max\{j \leq i \mid j \equiv 0 \pmod{2^k}\}$$

with dot-product separation at least $\frac{1}{2}$ (hence score separation at least $\frac{1}{2\sqrt{d_k}}$).

Let $L := r - k + 1$ be the number of rotated frequencies used by this head, and write $d_0 := i - j_0 \in \{0, \ldots, 2^k - 1\}$. Since $F_{\mathrm{mod}}^k(j_0) = 1$, the dot product to $j_0$ is

$$\langle \mathrm{rot}_\omega(q_i, i), \mathrm{rot}_\omega(k_{j_0}, j_0) \rangle = 2 + \sum_{l=k+2}^{r+2} \cos\left(\frac{2\pi}{2^l}(i - j_0)\right) = 2 + L - \sum_{l=k+2}^{r+2}\left(1 - \cos\left(\frac{2\pi}{2^l}d_0\right)\right).$$

We upper bound the loss term. For $l = k + 2$ we have $0 \leq \frac{2\pi}{2^l}d_0 < \frac{\pi}{2}$, hence $1 - \cos(\cdot) \leq 1$. For $l = k + 3$ we have $0 \leq \frac{2\pi}{2^l}d_0 < \frac{\pi}{4}$, hence $1 - \cos(\cdot) \leq 1 - \cos(\pi/4) = 1 - \frac{\sqrt{2}}{2}$. For $l \geq k + 4$ we use $1 - \cos x \leq \frac{x^2}{2}$ and $d_0 \leq 2^k$ to get

$$\sum_{l=k+4}^\infty \left(1 - \cos\left(\frac{2\pi}{2^l}d_0\right)\right) \leq \frac{1}{2}\sum_{l=k+4}^\infty \left(\frac{2\pi}{2^l}2^k\right)^2 = \frac{\pi^2}{96}.$$

Therefore

$$\langle \mathrm{rot}_\omega(q_i, i), \mathrm{rot}_\omega(k_{j_0}, j_0) \rangle \geq 2 + L - \left(1 + \left(1 - \frac{\sqrt{2}}{2}\right) + \frac{\pi^2}{96}\right).$$

Using $\frac{\sqrt{2}}{2} > \frac{7}{10}$ and $\pi^2 < 10$, the bracket is strictly smaller than $\frac{3}{2}$, hence

$$\langle \mathrm{rot}_\omega(q_i, i), \mathrm{rot}_\omega(k_{j_0}, j_0) \rangle > L + \frac{1}{2}.$$

Next, we upper bound all other dot products. If $F_{\mathrm{mod}}^k(j) = 0$, then the unrotated contribution vanishes and the dot product to $j$ is at most $L$ since every cosine is at most 1. If $F_{\mathrm{mod}}^k(j) = 1$ but $j \leq j_0 - 2^{k+1}$, then $D := i - j \geq 2^{k+1}$. Choose $l^\star$ such that $2^{l^\star - 1} \leq D < 2^{l^\star}$. Then $l^\star \in \{k+2, \ldots, r\}$ and both indices $l^\star$ and $l^\star + 1$ lie in $\{k+2, \ldots, r+2\}$. Writing $\alpha := \frac{2\pi}{2^{l^\star}}D \in [\pi, 2\pi)$, we obtain

$$\cos \alpha + \cos(\alpha/2) \leq 0,$$

hence two consecutive cosine terms contribute at most 0 in total and all remaining $L - 2$ terms contribute at most 1. Therefore $\sum_{l=k+2}^{r+2} \cos(\frac{2\pi}{2^l}D) \leq L - 2$ and thus the dot product to $j$ is at most $2 + (L - 2) = L$. Finally, if $j_1 := j_0 - 2^k \geq 0$, then $F_{\mathrm{mod}}^k(j_1) = 1$ and we compare $j_0$ to $j_1$. For every $l \in \{k+2, \ldots, r+2\}$ we have

$$0 \leq \frac{2\pi}{2^l}d_0 \leq \frac{\pi}{2} \qquad \text{and} \qquad \frac{2\pi}{2^l}(d_0 + 2^k) = \frac{2\pi}{2^l}d_0 + \frac{2\pi}{2^{l-k}} \leq \pi,$$

so $\cos$ is decreasing on this interval and each summand in the cosine sum for $j_0$ is at least the corresponding summand for $j_1$. Moreover, for $l = k + 2$ the shift is exactly $\frac{\pi}{2}$, and with $\beta := \frac{2\pi}{2^{k+2}}d_0 \in [0, \frac{\pi}{2}]$,

$$\cos\beta - \cos(\beta + \pi/2) = \cos\beta + \sin\beta \geq 1.$$

Hence the dot product to $j_0$ exceeds the dot product to $j_1$ by at least 1.

In total, $j_0$ is the unique maximizer of the dot products and the dot-product gap to the second largest dot product is at least $\frac{1}{2}$, hence this head has score separation at least $\frac{1}{2\sqrt{d_k}}$.

The MLP in layer $k + 2$ then writes $I_{\mathrm{res}}[k] = 1$ if $F_{\mathrm{ex}} = 0$ and $I_{\mathrm{res}}[k] = -1$ if $F_{\mathrm{ex}} = 1$, and resets $F_{\mathrm{ex}}$ to 0 (again using Lemma C.1). By the discussion at the beginning of this paragraph, this yields exactly the $k$-th bit $\mathrm{bin}_r^k(i)$. □

### F.2.3. INTRODUCING ROUNDING

We briefly sketch how the hardmax construction above could be converted to softmax with rounding using the techniques from Appendix D. The end-to-end softmax conversion with activation rounding in Theorem D.16 relied on the fact that all relevant activations (in particular queries and keys) are ternary and hence exactly representable in the activation precision $\mathbb{F}_{\mathrm{act}}$, so that one can repeatedly apply Lemma D.12. When using RoPE, queries and keys should however be rounded after rotating them as well, as the rotated queries and keys are used in the computationally expensive attention score computation (see Remark D.5). The rotated queries and keys are not exactly representable anymore in the above construction, so Lemma D.12 does not apply to them and we need to control the induced score perturbation.

A natural way to do this is to adapt the denoising approach from Theorem D.17. In the notation of Definition D.4, the softmax error bounds depend on the effective separation $\beta - 2\Delta_s$ (see Lemma D.8), where $\beta$ is the separation of the

unperturbed (scaled) scores and $\Delta_s$ the maximal perturbation of the scores. Scaling query and key projections by $c$ as in Appendix D scales $\beta$ by $c^2$, and in our construction rotated queries and keys have entries bounded by $\mathcal{O}(c)$. With $b_m$ mantissa bits, rounding such entries incurs coordinatewise error $\mathcal{O}(c\,2^{-b_m})$ (by Lemma B.19 in the normal range), hence the induced perturbation of the (normalized) attention scores scales as $\Delta_s = \mathcal{O}(c^2\sqrt{d_k}\,2^{-b_m})$ (cf. Lemma D.7). On the other hand, by Lemma F.2 the unscaled RoPE heads have score separation at least $\frac{1}{2\sqrt{d_k}}$, so after $c$-scaling we have $\beta \geq \frac{c^2}{2\sqrt{d_k}}$. Therefore $\Delta_s/\beta = \mathcal{O}(d_k\,2^{-b_m})$, and choosing $b_m = \mathcal{O}(\log d_k) = \mathcal{O}(\log\log N)$ makes $\Delta_s$ a small constant fraction of $\beta$, so that e.g. $\beta - 2\Delta_s \geq \frac{1}{2}\beta$ can be ensured. With this modification, the denoising proof from Theorem D.17 can be adapted to this RoPE prefix (using the effective separation $\beta - 2\Delta_s$ from Lemma D.8 in place of $\beta$) and yields a rounded softmax RoPE position-extraction prefix that behaves like the hardmax one on contexts of length at most $N$. We omit the details and explicit constants.

It is noteworthy that problems with RoPE and low precision rounding have been observed empirically (Wang et al., 2025b).

### F.2.4. INCLUSION INTO CONSTRUCTIONS.

Each of the hardmax constructions could now be modified to use RoPE instead of the binary positional embeddings by prepending the layers from Lemma F.2 to extract the binary positional encodings into $I_{\text{pos}}$ before the actual construction. This assumes that attention heads for the prepended block have enough rotated and 2 unrotated dimensions as required by Lemma F.2, while later attention heads have enough unrotated dimensions as the hardmax constructions do not make use of the rotated dimensions.

### F.3. Computational Efficiency and Attention Sparsity

Running a CoT transformer with depth $L$, model width $d$ and $d_k = d_v = \frac{d}{H}$ (as is common) for $n$ steps generally uses $\mathcal{O}(nLd)$ memory and $\mathcal{O}(n^2Ld)$ compute operations[8]. Applied to the CoT Turing machine construction, simulating a Turing machine using $t$ steps and $s$ space costs $\mathcal{O}(t(\log N)^2)$ memory and $\mathcal{O}(t^2(\log N)^2)$ compute. SCoT brings that down to $\mathcal{O}(s(\log N)^2)$ memory and $\mathcal{O}(ts(\log N)^2)$ compute as the context size is bounded by $\mathcal{O}(s)$. Ignoring the $\log N$ factors, the memory usage of the transformer corresponds to that of the Turing machine, but the compute is still larger by a factor $s$.

The remaining factor $s$ comes entirely from the attention: for each generated token, the query has to be compared to all $\mathcal{O}(s)$ keys in context to identify the maximizing score. At the same time, in our constructions the heads are *effectively* 1-*sparse* in the sense discussed in Section 4: either the maximizer is unique, or all maximizers carry the same value $v_j$, so the head output is unchanged if we only retrieve the value from a single maximizer. Computing an attention head can hence be viewed as a *maximum inner product search* (MIPS) / nearest-neighbor problem.

For MIPS, efficient approximate methods exist, e.g. locality-sensitive hashing and graph-based indices such as HNSW (Indyk & Motwani, 1998; Malkov & Yashunin, 2020), which are often reported to have roughly logarithmic scaling in practice and could drop the extra factor $s$ to a polylogarithmic (in $N$) overhead in our SCoT setting. Unfortunately, these methods are inherently approximate: they are not guaranteed to return the true maximizer, and even their runtime can be poor in worst cases (Indyk & Xu, 2023). Nonetheless, it has been observed empirically that transformers can be run well when approximating attention via such retrieval-style sparsification (Liu et al., 2025; Kitaev et al., 2020), offering practical benefits despite being difficult to parallelize as effectively as standard attention. Our results hence give theoretical credibility to these observations, showing that even 1 retrieved token per query is enough for Turing completeness in our setting.

Sparse attention has also been shown to retain strong expressivity in settings where the sparsity pattern is fixed by design (Zaheer et al., 2020; Li & Wang, 2026). Conceptually, these results differ from retrieval-based sparse attention approximations (such as (Liu et al., 2025; Kitaev et al., 2020)), which aim to approximate the standard dot-product attention of a full transformer by dynamically retrieving a small set of tokens.

---

[8]Really the compute is $\mathcal{O}(n^2Ld + nLd^2)$, but the first term dominates in the considerations here.

