# OpenReview forum: "The Expressive Power of Low Precision Softmax Transformers with (Summarized) Chain-of-Thought"
_ICML.cc/2026/Conference — ICML 2026 regular_

### Official Review · Reviewer_k18p · 2026-02-14

**Soundness:** 3
**Presentation:** 3
**Significance:** 2
**Originality:** 3
**Overall Recommendation:** 4
**Confidence:** 3

**Summary:**

This paper studies the expressive power of Transformers with Chain-of-Thought (CoT) reasoning by formulating the problem as how efficiently a Transformer can simulate a Turing machine. In contrast to prior work that assumes high-precision activations (with precision scaling logarithmically in the number of reasoning steps) and hardmax attention, the paper replaces these assumptions with low-precision computation (while allowing model depth/width to scale logarithmically) and standard softmax attention. The authors first provide new theoretical constructions showing Turing-machine simulation under this low-precision setting. To address the growth of required model size, they further introduce summarized Chain-of-Thought (SCoT), arguing that since the theoretical model size depends on the maximum segment length rather than the total reasoning length, summarization offers a principled way to reduce scaling requirements. Finally, the paper presents experiments on Sudoku tasks to examine how small models generalize to longer reasoning chains under CoT and SCoT settings.

**Compliance With Llm Reviewing Policy:**

Affirmed.

**Key Questions For Authors:**

- Q1. Prior work assumes high precision largely because positional encodings require at least log $n$ bits to distinguish sequences of length $n$—a natural assumption given that such distinctions are necessary in practice. What is the motivation for studying the low-precision setting, and what theoretical or practical benefits does it offer?
- Q2. Does the extension from hardmax to softmax attention essentially rely only on scaling the attention scores so that softmax approximates hardmax, or are there additional technical challenges involved?
- Q3. Turing-machine simulations are primarily used to demonstrate theoretical universality. Is it appropriate to directly relate the resulting time- and space-complexity arguments to practical reasoning methods, or does this introduce a conceptual gap between the theoretical analysis and the empirical evaluation?

**Limitations:**

yes

**Strengths And Weaknesses:**

- *Soundness:* The theoretical arguments are mathematically rigorous, and the proof strategy is well structured.
- *Presentation:* The paper is logically organized, with a clear separation among the theoretical constructions, the softmax conversion analysis, and the empirical evaluation.
- *Significance:* The paper makes an important theoretical contribution by establishing the expressive power of quantized CoT reasoning. However, the empirical evaluation is limited to Sudoku tasks, and the practical impact of Summarized CoT remains somewhat limited at this stage.
- *Originality:* This work offers a novel theoretical contribution by eliminating the requirement for high-precision activations, instead leveraging low-precision rounding alongside new constructive proofs.

---

> ### Author Rebuttal · Authors · 2026-03-30
>
> Thank you for your thoughtful review and questions.
>
> First, we want to comment on the significance point. We believe the practical relevance of Summarized CoT is broader than the current version of the paper suggests: maintaining a compact summarized state instead of carrying the full reasoning trace is closely related to recent context compaction approaches for extended reasoning. In that sense, Summarized CoT is not merely a theoretical variant of CoT, but a formal abstraction of a practically relevant mechanism.
>
> Q1: You are correct about the use of logarithmic precision to encode $n$ distinct positions; in our setting we instead encode positions in binary using logarithmic width. The main motivation behind studying our setting was that practical transformers use softmax attention and precision that is too low to justify a log-precision assumption. In particular, while our constructions remain compatible with standard 16-bit formats for all practical context lengths, prior log-precision constructions break down quite quickly at realistic precision.
>
> Q2: Yes, this is essentially the main idea and has been put forward in prior work. When also rounding activations and in particular attention weights and trying to get concrete bounds on parameter growth and required precision, however, analyzing error propagation becomes technically involved.
>
> Q3: First of all, we agree that one should be careful not to directly conflate Turing-machine simulations and practical reasoning methods. We do however think that theoretical results for Turing machines, particularly under realistic model assumptions such as softmax attention and low-precision, can give insights into practical regimes. This can for instance be seen in the experiments in our paper: Theorems 3.2 and 3.3 suggest that CoT requires larger model size as reasoning length grows, while model size with SCoT should scale with segment length. This is consistent with the experiments in Section 5, which demonstrate a case (albeit a modest one) in which the theoretical results for Turing machines agree with practical reasoning for Sudoku.

---

> > ### Author Rebuttal · Reviewer_k18p · 2026-03-31
> >
> > The authors’ rebuttal clearly and directly addressed the concerns I raised in my review. While the paper is well-executed and technically sound, I still have reservations about its practical impact. Therefore, I will keep my original score.

---

### Official Review · Reviewer_dqjc · 2026-03-11

**Soundness:** 4
**Presentation:** 3
**Significance:** 3
**Originality:** 3
**Overall Recommendation:** 5
**Confidence:** 4

**Summary:**

This paper studies the expressive power of decoder-only transformers under assumptions that are closer to practice than prior expressivity papers. The main result is that standard transformers with softmax attention and doubly logarithmic precision rounding can simulate Turing machines when allowed logarithmic scaling in depth and width (with respect to the required context-window length). The paper gets there in two steps. First, the authors construct hardmax transformers with ternary activations and good attention-score separation ($1/\sqrt{d_k})$ for simulating DFAs and Turing machines up to a certain time/space bound. Then, they convert those constructions into softmax transformers while controlling the errors introduced by softmax and rounding (through appropriate scaling $c$, Theorem D.16).

A second main contribution of the paper is the analysis of summarized Chain-of-Thought (SCoT). The paper shows that SCoT is more efficient than standard CoT in the sense that the model size can scale with a space bound rather than a time bound. The Sudoku experiments are meant to support this picture empirically: Small SCoT models solve very hard instances and generalize to much longer reasoning traces than seen in training, while comparable CoT models do not.

**Compliance With Llm Reviewing Policy:**

Affirmed.

**Final Justification:**

I am maintaining my score of 5 (Accept), as this is a technically rigorous, well-motivated paper that makes valuable contributions to our understanding of transformer expressivity. Moreover, the paper provides a valuable summary of the related literature, and the overall presentation is great.

I gave high significance and originality scores because the paper proves Turing-completeness for softmax transformers under practical assumptions (logarithmic depth/width and doubly logarithmic precision). I also weigh the paper's significance highly because of the analysis of Summarized Chain-of-Thought (SCoT). Proving that model size can scale with a space bound rather than a time bound is an elegant conceptual contribution. In terms of clarity, the narrative is well-structured, though my initial review noted minor presentation concerns regarding the placement of Table 2 and the comparative framing with Li & Wang (2025b).

The authors provided a highly satisfactory rebuttal that completely addressed my questions and strongly reinforced my prior positive assessment. Overall, the rebuttal strengthened my confidence in the authors' rigorous theoretical grasp of the subject. I consider this an excellent addition to the expressivity literature and confidently recommend acceptance.

**Key Questions For Authors:**

1. Could you elaborate more on how the expressivity construction from (Li & Wang, 2025b) differs from yours and why one cannot supplement their construction with SCoT? In particular, can you explain your sentence on line 747: "Furthermore,
context size is not adaptive as in (Yang et al., 2025b) and our SCoT results, but fixed to $\hat{s}$."? Isn't their context window $O(\hat{s})$ already according to Table 2?

2. Can you comment further on your closing remarks (lines 425-427): "An obvious future avenue is to investigate, both theoretically and empirically, whether the logarithmic scaling of model size in these results is actually necessary."? Do you think one can achieve log width and depth but constant precision? Are you aware of any lower bounds on the minimal transformer size?

3. Would the transformer size need to grow substantially in order to accommodate LayerNorm into the expressivity proofs?

**Limitations:**

Yes.

**Strengths And Weaknesses:**

I think this is a strong and well-motivated paper. It addresses the limitations of prior expressivity results by proving that transformers can simulate Turing machines under reasonable assumptions. In particular, this is the first paper to prove Turing-completeness for softmax attention with doubly logarithmic precision and logarithmic depth/width with respect to the context length. Prior results either rely on much larger parameter precision (logarithmic or even linear) or use hardmax attention. Thus, the present work substitutes the constant parameter count and logarithmic precision of previous papers with polylogarithmic parameter count and loglog parameter precision -- a setting which I agree is closer to practice.

Moreover, this paper also proves that the sufficient depth, width, and precision of a softmax transformer with Summarized CoT (SCoT) need not depend on the total computation time of the underlying algorithm but on the required computation space. I found this SCoT result particularly interesting, and I consider it the strongest contribution of the paper. The shift from time-bounded to space-bounded scaling is a clean conceptual contribution, which shows that smaller models can generalize to longer problems when utilizing SCoT. I think this theoretical conclusion is nicely supported by the Sudoko experiments.

The paper is also generally well presented. The main narrative is easy to follow because it builds towards the softmax results by first going through the ternary hardmax constructions. The comparison to prior work is strong, especially the discussion of how this work differs from earlier Turing-completeness results for transformers. My main criticism of the presentation is that I would have liked to see Table 2 in the main part of the paper with a detailed comparison deferred to the appendix. Table 2 very clearly situates the contribution of the present work in the overall transformer expressivity literature. I get the impression that this wasn't done because the authors feel that such a comparison could hurt the paper by painting their results as incremental. For example, the table shows that (Wei et al., 2022a) have already achieved loglog precision for hardmax encoder-decoder transformers; and that (Jiang et al., 2025) have already proven Turing-completeness for softmax transformers. Moreover, the table cites (Li & Wang, 2025b) as achieving Turing-completeness with constant width, depth and precision.

I agree that reference (Li & Wang, 2025b) is very scary to have in the main part of the paper, and I recommend that the authors drop it from the comparison in Table 2. I believe this is removal is warranted because the transformer construction in (Li & Wang, 2025b) is very different in spirit from the constructions in the rest of the cited works since it uses ad-hoc positional information tailored to a specific Turing machine that cannot be discovered by gradient-based optimization. Hence, (Li & Wang, 2025b) operate in a different regime, and even though they achieve a much better expressivity result than the present paper, the comparison is not meaningful.

Overall, I acknowledge the difficulty in proving Turing-completeness for softmax transformers with loglog precision, and I consider this result as an important, albeit incremental, addition to the expressivity literature. I found Theorem 3.3 as the main conceptual contribution of the paper.

*Additional comments:*
1. On line 127, $x_0, \dots, x_i$ lack the superscript $(\ell-1)$ in the attention head $A_{\ell, h}$.

---

> ### Author Rebuttal · Authors · 2026-03-30
>
> Thank you for the positive and constructive feedback.
>
> We agree that Table 2 and the related comparison are an important aspect of the paper and we plan to highlight them more in the final version, potentially (depending on available space) moving a reduced version of the table into the main part as you suggested.
>
> Q1: Li & Wang [1] is indeed an interesting work as they achieve constant precision and model size at the same time. Fundamentally this works by not generating a sequence of tokens encoding one Turing machine step each, but instead simulating multi-queue machines. This avoids having to do things like searching for the last token with a matching head position, and instead allows reducing each attention head to only attend to a specific context-independent position at a pre-specified offset. This in turn is solved by simply using appropriate biases as positional encodings, hence requiring a simple but very much non-standard positional encoding. As tokens at a distance larger than the largest offset will never be attended to, they can simply be dropped from context, allowing for smaller context sizes when space usage is smaller. Crucially, the positional encodings and hence the context size depend on the space upper bound $\hat s$ which must be known a priori, and context size becomes $\mathcal O(\hat s)$ ($\hat s + 1$ to be precise) regardless of the input possibly requiring much less space. In order to get a single model that works for many inputs, one would like to choose $\hat s$ very large, but this negates the space-efficiency for inputs where the Turing machine's space usage is small. The dynamic, content-dependent context drop used in PENCIL [2] and in our SCoT constructions, on the other hand, allows choosing $\hat s$ very large, incurring only logarithmic precision or model size growth respectively, while context size adapts dynamically to the input. It is unclear to us whether the general construction idea in Li & Wang could be modified to achieve such an input-dependent context drop, and they themselves note this as an open problem.
>
> Q2: As detailed in our answer to question 1 from reviewer 1, we do believe that logarithmic width and depth are required when trying to simulate a Turing machine with one token per Turing machine step, a standard architecture, and the precision assumptions of our constructions. We are not aware of a lower bound that directly covers this setting; existing transformer lower bounds typically work under rather different assumptions and therefore do not resolve this question cleanly. Regarding your question on achieving log width & depth with constant precision: likely no when looking for softmax transformers with the standard $\frac{1}{\sqrt{d_k}}$ attention score scaling. As with constant precision attention scores will be bounded as $\mathcal{O}(d_k)$ (where $d_k = \mathcal{O}(\log n)$ with context size $n$), one can upper bound each attention weight by a term converging to zero as $n$ tends to infinity. Hence, as context size grows, attention will ultimately become too spread out to precisely attend to anything and rounding of attention weights will even make them all zero. Thus, at least some additional scaling of attention scores is required, for example by scaling queries and keys, which requires (very slowly) growing activation precision.
>
> Q3: No, a constant increase in width with no change in depth should suffice. In particular, in the ternary hardmax constructions, all activation entries $-1,0,1$ could be replaced e.g. as $(-1,1,0,0), (-1,0,1,0), (-1,0,0,1)$ respectively, quadrupling representation width. All the MLP operations and attention heads would need to be adapted to perform the same operations on these new representations, possibly increasing MLP and attention dimensions by constant factors. In the resulting transformer, all representations then have mean 0 and variance $1/2$ and hence (pre- or post-) LayerNorm with appropriate parameters has no effect, analogous to the invariance argument in [3]. Softmax conversion with rounding and error control works analogously to how it is done originally. Excluding LayerNorm was done mainly to avoid these technicalities.
>
> [1] Li & Wang (2025b). Efficient Turing Machine Simulation with Transformers.
> [2] Yang et al. (2025b). PENCIL: Long Thoughts with Short Memory.
> [3] Li et al. (2024). Chain of Thought Empowers Transformers to Solve Inherently Serial Problems.

---

> > ### Author Rebuttal · Reviewer_dqjc · 2026-04-04
> >
> > Thank you for your detailed rebuttal. My questions are fully resolved, and I will be maintaining my score of 5.

---

### Official Review · Reviewer_JKmW · 2026-03-12

**Soundness:** 3
**Presentation:** 3
**Significance:** 2
**Originality:** 2
**Overall Recommendation:** 4
**Confidence:** 4

**Summary:**

The paper presents several expressivity results about a new transformer model with ternary activations and parameters, non-uniform parameters with log depth and width, and various forms of CoT. First, the authors essentially recover known results for other transformer models in this model: regular language recognition without CoT (but with log depth), simulating a Turing machine with CoT, and some possibly new results with summarized CoT, showing that CoT length can track space rather than runtime. They then show that the averaging-hard-attention constructions in this modern can be converted to softmax attention with growing floating point precision (an idea essentially known in past work).

**Compliance With Llm Reviewing Policy:**

Affirmed.

**Key Questions For Authors:**

Clarify "Furthermore" part of Theorem 4.1

> exponent bits are double logarithmic for attention weights and triple-logarithmic for activations

In a rational-valued model, this would mean log precision attention and loglog precision activations. Log precision attention is probably the least that can be expected. For activations, it's interesting that you can do better, in line with the comments about mixed precision above (though this presumably relies on having log width; without that, you would probably need log precision activations as well).

**Limitations:**

yes

**Strengths And Weaknesses:**

## Strengths

The paper clearly introduces a new transformer model and presents various results in different CoT regimes. Technical arguments are generally clear and seem correct. I don't consider the model here to necessarily be the best model of practical transformers (more on that below), but the results may help the community relate existing transformer models as it moves towards consolidating different existing results.
## Weaknesses

This paper already proves uniform softmax transformers (fixed depth) are Turing-complete: https://arxiv.org/abs/2511.20038. Thus, the headline result that this is attainable in a more complicated model with log precision is weaker. There could still be opportunity to compare your results: for example, theirs seems like it might incur exponential runtime relative to a TM, whereas yours might get away with less runtime.

A central part of the paper seems to be the proposal for a new transformer model (ternary activations with log width), an argument this is a useful formal model, and then proofs that you can recover standard results about various dynamic depth transformers in the log-precision uniform model in this model (log-depth, CoT, SCoT). With this in mind, I am not convinced the model analyzed here is a better match to practice than existing models. In general, the authors should be much more precise and careful about what they think the "fundamental mismatch" between current results and standard transformers is. I elaborate on this extensively below.
### Precision

First, regarding precision, I have some objections to the arguments made in the paper:

> In real transformers, attention weights αij are rounded to low precision as well before being multiplied by the values vj

Not true! While activations use 8 or 16 bits, practical models tend use 32 bits to compute attention (cf. Olmo reports), sometimes called mixed-precision training, as performance takes a catastrophic hit without this. Thus, in practice, context length typically almost always satisfies n <= 2^p, which really justifies log precision as a practical model in my view. There is room for disagreement about whether to model the other components of the network as log precision or fixed precision (cf. C-RASP, which accounts for mixed-precision transformers), and it of course may be interesting to study fixed-precision transformers out of general curiosity. But I do not immediately buy that fixed-precision attention is the right practical model, and the authors should be more precise in how they motivate this.
### Layer Norm

In my mind, log precision and AHAT are fine assumptions. in fact, your results don't get around log precision. The thing that is the most non-standard in past work is probably the assumptions about layer-norm placement (although this also varies in practice, in addition to theory). Your ternary constructions with log width probably don't require any differences in layer-norm, right? Can you comment on this and potentially compare the two models more generally?
### Uniformity

"effectively relocating the non-uniformity to activation precision": allowing precision to change with n is not non-uniform in the conventional sense because the change in precision has a simple computational description. there's a simple function we can write down by which the model is fully specified.

In contrast, allowing parameters to change with n is non-uniform because there is no compact description of how to compute the parameters as n changes. This adds degenerate computational power to the model: e.g., with 0 CoT, a transformer that is non-uniform in this way can solve undecidable problems. This property makes the model undesirable from a theoretical perspective.
### Depth and Width

> as powerful practical models use large depth and width but typically no increased precision

Scaling depth and width increases performance, but this is not typically with respect to context length. Rather, the question is how loss changes as model size and parameter count scale, and depth or width are two different ways to increase model size, rather than focusing on the relationship to sequence length.

### Experiments

The relevance of the experiments in Section 5 is currently unclear. I encourage the authors to better describe the theoretical hypotheses that are being tested by the experiments and what the implications of the results are.

---

> ### Author Rebuttal · Authors · 2026-03-30
>
> Thank you for your constructive critique and comments.
>
> The paper you mention [1] is discussed in Appendix A.1, and we will extend that discussion to highlight the comparison more clearly. In particular, under unbounded activation precision, their work is the only one to achieve uniform Turing machine simulation with a softmax transformer, as all other works would require context-size dependent parameter scaling when converted to soft attention. As you correctly pointed out, the exponential overhead of counter machines when simulating Turing machines means their logarithmic-in-context-size precision requirement is actually precision linear in the number of simulated Turing machine steps. Thus, Jiang et al. are stronger on uniformity, while our result targets a different axis: efficient Turing machine simulation under precision assumptions that remain close to those used in practice.
>
> (Precision) You are right that in typical implementations the attention scores $\langle q_i, k_j \rangle$ and the softmax itself are kept in 32-bit for numerical stability. Our point was not that the whole attention computation is carried out in low-precision. Rather, in common fused implementations, the attention weights are downcast to match the precision of the values before the multiplication with them, so the weighted sum is formed from low-precision weights and low-precision values. As concrete evidence, the FlashAttention Triton kernel [2] contains:
> ```
> p = p.to(v.dtype)
> acc_o += tl.dot(p, v)
> ```
> This is exactly the attention weight rounding step in our rounding model. More importantly, several prior works also need logarithmic precision in other activations that are actually rounded to 16 or 8 bit in practice, and Appendix F.1 argues that these requirements become restrictive at much smaller context sizes than in our constructions.
>
> (Layer Norm) We chose not to include LayerNorm for simplicity and do not rely on any special placement. We expect the same invariance argument as in [3] to handle pre- and post-LayerNorm, see the answer to question 3 from reviewer 3 for some details. As you point out, this is in contrast to works such as [4], whose construction relies on a special non-standard form of LayerNorm.
>
> (Uniformity) This is a good point and we will weaken the phrasing here. We agree that from a mathematical standpoint, growing activation precision with input size is still uniform, while changing the actual model is not. Our intended point was only practical: having to grow the required precision format with context length is also undesirable if one wants a single realistic deployment format such as bf16. We will separate these two notions more carefully instead of calling the former non-uniform.
>
> (Depth and Width) We agree that the phrasing in the introduction is misleading and will be changed: we did not want to claim that in practice depth and width are scaled with context size directly. Instead, the argument is rather that if a model's capacity is insufficient for a task, practitioners typically increase model size rather than arithmetic precision.
>
> (Experiments) We agree that the motivation for Section 5 should be stated more clearly. The experiments are meant to test the qualitative prediction of Theorems 3.2 and 3.3 that for algorithmic tasks with small working space, the model size required by SCoT should scale mainly with segment length rather than total reasoning length, whereas CoT becomes harder as the full context grows. On Sudoku this is exactly what we observe: at fixed model size, SCoT generalizes to much longer reasoning traces as summaries keep segments short, while CoT fails to fit the task for longer context lengths. In addition, increasing model size helps CoT substantially in this setting, whereas increasing precision does not; for more on this low-precision and trainability perspective, we refer to our answer to question 4 from reviewer 1.
>
> (Single vs. double logarithmic) First, to clarify Theorem 4.1: this only treats rounded activations with exact attention, so the triple-logarithmic requirement there concerns activation exponent bits needed to represent the softmax-scaling constant $c$. The double-logarithmic exponent bits for attention weights arise only in Theorem 4.2, where attention weights are rounded as well. You are also correct that with rational-valued representations this would become logarithmic attention precision and double-logarithmic activation precision. We used floating-point formats with rounding because they are closer to practice and make the required growing precision arise only through the range of representable numbers.
>
> [1] Jiang et al. (2025). Softmax Transformers are Turing-Complete.
> [2] https://github.com/Dao-AILab/flash-attention/blob/main/flash_attn/flash_attn_triton.py
> [3] Li et al. (2024). Chain of Thought Empowers Transformers to Solve Inherently Serial Problems.
> [4] Merrill and Sabharwal (2024). The Expressive Power of Transformers with Chain of Thought.

---

> > ### Author Rebuttal · Reviewer_JKmW · 2026-04-01
> >
> > I consider my concerns resolved, which were mainly about presentation and positioning the work relative to related work.
> >
> > Thanks for the clarification around [1]: I consider this concern addressed if appropriate discussion can be added to the main text.
> >
> > Regarding precision, thanks for the clarification. It would be good to discuss more explicitly what role attention precision and non-attention precision play in each construction (e.g., non-attention precision allows the propagation of pointers from one layer to the next). Your mixed-precision model is very related to the model in https://arxiv.org/abs/2506.16055, which is shown to be equivalent in expressivity to C-RASP with fixed-depth and without CoT. This is an interesting connection and I think it would be good to discuss this too
> >
> > Clarifying the other details you mentioned around layer norm, uniformity, and depth vs. width would also improve the paper in my view.

---

> > > ### Author Response · Authors · 2026-04-08
> > >
> > > We are happy that our clarifications around layer norm, uniformity etc. helped resolve your concerns and we will make sure to include them in the final version. Furthermore, as pointed out in the answer to reviewer 3, we will move the comparison table to the main text. We will also try to extend the discussion around related works (in particular [1]) in the main text as space permits. Thank you for pointing us to the C-RASP paper, which indeed uses a precision model very similar to ours without attention weight rounding and will hence be discussed appropriately.

---

### Official Review · Reviewer_eF2A · 2026-03-13

**Soundness:** 3
**Presentation:** 4
**Significance:** 2
**Originality:** 3
**Overall Recommendation:** 4
**Confidence:** 4

**Summary:**

**Summary**

The paper proposes that low precision and logarithmic width/depth is the right parameterization to model transformers in practice. Under these assumptions, the paper proves that transformers with CoT are Turing complete, matching the number of steps used by the Turing machine. Summarized CoT (an extension) allows the attainment of matching space bounds.

**Compliance With Llm Reviewing Policy:**

Affirmed.

**Final Justification:**

The theoretical results are very well-thought out and clearly positioned in relation to prior work. This is no small feat, in an area that is amassing a lot of different perspectives and subtle assumptions that may affect results. Of course there are limitations in scope, but this is still a notable theoretical contribution. The careful presentation alone is of value to the community. I commend the authors and highly rate the theoretical portion of the paper.

Presently my reservations are mostly about the experiments, which do not seem to confirm or falsify the theory in a conclusive manner.

**Key Questions For Authors:**

The main thesis of the paper seems to be that low-precision and log-depth/width is the correct way to analyze language models in practice, and I think proving this point should be more central to the writeup. The novelty of the constructions in the paper depends on the novelty of the low-precision assumption – otherwise they are fairly routine. The paper is quite well-written and I would like to see it accepted but I think the bar for significance at this point needs to be somewhat higher compared to previous work in the area.

If questions (1)-(3) are addressed satisfactorily then the significance, originality, and soundness of the paper will be improved and I will add +1 to my recommendation. Currently it’s unclear how the experiments are related to the low-precision framework, but if clarified in the answer to question (4) the presentation will be improved and I will also add +1 to my recommendation.

**Questions**
- Are both log depth and log width crucial for the results of this paper? Is there no way to use less?
- F.1 analyzes the parameters of GPT-3 to show that low-precision suffices to express the transformer computations over very large context windows. Does this apply to other language models as well? Curious about both larger and smaller ones.
- Historically it seems that assuming constant-depth log-precision was what enabled the proof that transformers were contained in the constant-depth circuit class TC^0. Because a log-depth transformer can simulate arbitrary automata, this allows them to break out of TC^0. This is a case where constant-precision log-depth transformers are more powerful than log-precision constant-depth transformers (which seems a bit incongruous since in practice transformers struggle to learn arbitrary automata). Do you think there are cases where log-precision becomes more powerful than low-precision?
- The experiments address how SCoT enables advantages over standard CoT. How does this relate to the expressivity of low-precision transformers? Aren’t the results consistent with both fixed-precision and log(n)-precision transformer expressivity?

**Limitations:**

yes

**Strengths And Weaknesses:**

**Strengths**
- Beautifully written with clean analysis
- Provides a novel perspective on what an appropriate model of transformer behaviour in practice should look like
- Tackling issues of precision and rounding are important due to the interest in quantizing large language models
- Experiments are interesting, well-motivated, and support the insights of the theory
- Appendix A gives a great overview of related work that is of independent value

**Weaknesses**
- Many of the results are straightforward reformulations of old ones
- The paper could use more evidence to back up the statement that low precision and log depth/width is the right perspective. This came off as a primary claim of the paper, but was not substantiated in the main body

---

> ### Author Rebuttal · Authors · 2026-03-30
>
> Thank you for your insightful review and questions.
>
> We agree with your general framing that the main thesis of our paper is that low precision with larger depth/width is a suitable regime to study practically relevant transformers, and that this should be clearly established. Before addressing your concrete questions below, we want to note that, based on your overall assessment, we plan to move some of the evidence from Appendix F.1 into the main paper. We believe this will more clearly highlight that the presented construction is the only one compatible with softmax attention and practical precision at realistic parameter magnitudes.
>
> Q1: Logarithmic width is required in our construction to encode distinct positions in the token sequence (for positional embeddings) and head positions in the Turing machine, analogous to how logarithmic precision is used in prior works when width is kept constant. Logarithmic depth is used mainly to perform a binary search for the latest token with the correct head position, which does not seem possible in constant depth. So when simulating a Turing machine directly (one Chain-of-Thought token per Turing machine step) with a standard decoder architecture, we do believe that both log width and depth are hard to avoid at low-precision with moderate parameter magnitudes. We do not claim a general lower bound, however. Works like [1] show that different methods are possible, but they are quite different from the route we analyze here: they simulate multi-queue machines rather than Turing machines directly and use non-standard positional encodings tailored to the known space bound. So our claim is only that, within a fairly standard decoder-style low-precision simulation of Turing machines, both logarithmic width and logarithmic depth seem hard to avoid. We discuss these caveats in more detail in our response to reviewer 3.
>
> Q2: Yes - the precision point is not specific to GPT-3. Appendix F.1 uses GPT-3 as a concrete example of a model with published depth and width to illustrate that realistic model sizes suffice to simulate a Turing machine with non-trivial time/space usage. The claim that the common bfloat16 precision suffices to control rounding errors for all practical context lengths is essentially model-independent. The main thing changing from model to model is how large a time/space bound the construction can fit into the available depth and head dimension. Similar computations can be made for both larger and smaller models, but for smaller models the explicit constants in our current theorems are too loose to give meaningful concrete bounds without making the construction more compact.
>
> Q3: While we cannot give conclusive evidence either way, we think that logarithmic precision with constant width is typically more powerful for one-layer retrieval and uniform attention tricks used in prior TM simulation constructions for obtaining constant depth, while logarithmic width with constant precision is more powerful for softmax conversion, as it allows well-separated attention scores and hence avoids extreme parameter magnitudes when sharpening scores. Overall, while we think such distinctions are important to study in the future, a precise exploration is beyond the scope of the current work.
>
> Q4: The experiments are indeed consistent with prior high-precision results as well. In particular, SCoT working across all inputs when context is kept bounded is also consistent with prior high-precision SCoT results [2]. Likewise, CoT failing at fixed model size on the harder Sudokus with longer contexts is consistent with both prior log-precision results and our work as a fixed-size limitation. The difference is in which scaling appears to matter for trainability: increasing model size helped substantially, as raising depth from 6 to 8 layers drastically improved CoT on longer contexts (Appendix E.4), while rerunning a failed CoT run in fp32 gave very similar loss and accuracy curves as bf16. Hence, in this specific setting, low-precision expressivity results seem to translate better into trainability predictions than prior high-precision ones, where increasing precision rather than model size is the analogous way to accommodate longer contexts. Nonetheless, our low-precision results still overestimate trainability, as you correctly point out with the difficulty of training transformers to simulate arbitrary automata. We will revise the final version to make this trainability perspective clearer in the paper and to include the fp32 comparison.
>
> [1] Li & Wang (2025b). Efficient Turing Machine Simulation with Transformers.
> [2] Yang et al. (2025b). PENCIL: Long Thoughts with Short Memory.

---

> > ### Author Rebuttal · Reviewer_eF2A · 2026-04-03
> >
> > Thank you for the very careful response!
> >
> > Q1) Thank you for the explanation about the dual need to encode positions and perform search. Maybe this could be relaxed if you didn't need a one-step for one-step construction, but this is of course not necessary to discuss here.
> >
> > Q2) Thank you for the clarification.
> >
> > Q3) I agree this would be interesting to study and potentially of notable significance to practice down the line
> >
> > Q4) Thank you for the honest explanation.
> >
> > I will increase my score to 4. I am still concerned about the significance of the experimental results and how they tie into the story of the theory, but I believe the theory is well-thought out and developed. The improved presentation will also be helpful to the reader. I commend the authors for a great piece of work.

---

### Decision · Program_Chairs · 2026-04-30

**Decision:**

Accept (regular)

**Comment:**

This paper studies the expressive power of softmax transformers under low-precision assumptions, proving Turing-completeness via Chain-of-Thought and showing that with Summarized CoT, model size scales logarithmically in a space bound rather than a time bound. All four reviewers acknowledged the technical rigor and clarity of the work, and gave the strongest endorsement, such as well-motivated approach, beautifully written with clean analysis. All four reviewers marked their concerns as fully resolved after rebuttal.

Given the strong consensus on technical soundness and novelty, I recommend acceptance.